# Efficient Uncertainty Quantification and Reduction for Over-Parameterized Neural Networks

**Ziyi Huang, Henry Lam, Haofeng Zhang** [*]
Columbia University
New York, NY, USA
`zh2354,khl2114,hz2553@columbia.edu`

## Abstract

Uncertainty quantification (UQ) is important for reliability assessment and enhancement of machine learning models. In deep learning, uncertainties arise not only from data, but also from the training procedure that often injects substantial noises and biases. These hinder the attainment of statistical guarantees and, moreover, impose computational challenges on UQ due to the need for repeated network retraining. Building upon the recent neural tangent kernel theory, we create statistically guaranteed schemes to principally *characterize*, and *remove*, the uncertainty of over-parameterized neural networks with very low computation effort. In particular, our approach, based on what we call a procedural-noise-correcting (PNC) predictor, removes the procedural uncertainty by using only *one* auxiliary network that is trained on a suitably labeled dataset, instead of many retrained networks employed in deep ensembles. Moreover, by combining our PNC predictor with suitable light-computation resampling methods, we build several approaches to construct asymptotically exact-coverage confidence intervals using as low as four trained networks without additional overheads.

## 1 Introduction

Uncertainty quantification (UQ) concerns the dissection and estimation of various sources of errors in a prediction model. It has growing importance in machine learning, as it helps assess and enhance the trustworthiness and deployment safety across many real-world tasks ranging from computer vision [67, 21] and natural language processing [113, 103] to autonomous driving [90, 91], as well as guiding exploration in sequential learning [9, 1, 119]. In the deep learning context, UQ encounters unique challenges on both the statistical and computational fronts. On a high level, these challenges arise from the over-parametrized and large-scale nature of neural networks so that, unlike classical statistical models, the prediction outcomes incur not only noises from the data, but also importantly the training procedure itself [73, 94]. This elicits a deviation from the classical statistical theory that hinders the attainment of established guarantees. Moreover, because of the sizes of these models, conventional procedures such as resampling [37, 30, 102] demand an amount of computation that could quickly become infeasible.

Our main goal of this paper is to propose a UQ framework for over-parametrized neural networks in regression that has simultaneous *statistical coverage guarantee*, in the sense of classical frequentist asymptotic exactness, and *low computation cost*, in the sense of requiring only few (as low as four) neural network trainings, without other extra overheads. A main driver of these strengths in our framework is a new implementable concept, which we call the *Procedural-Noise-Correcting (PNC)* predictor. It consists of an auxiliary network that is trained on a suitably artificially labeled dataset, with behavior mimicking the variability coming from the training procedure. To reach our goal, we

---

[*]Authors are listed alphabetically.

37th Conference on Neural Information Processing Systems (NeurIPS 2023).

synthesize and build on two recent lines of tools that appear largely segregated thus far. First is neural tangent kernel (NTK) theory [64, 80, 7], which provides explicit approximate formulas for well-trained infinitely wide neural networks. Importantly, NTK reveals how procedural variability enters into the network prediction outcomes through, in a sense, a functional shift in its corresponding kernel-based regression, which guides our PNC construction. Second is light-computation resampling methodology, including batching [47, 99, 100] and the so-called cheap bootstrap method [75], which allows valid confidence interval construction using as few as two model repetitions. We suitably enhance these methods to account for both data and procedural variabilities via the PNC incorporation.

We compare our framework with several major related lines of work. First, our work focuses on the quantification of epistemic uncertainty, which refers to the errors coming from the inadequacy of the model or data noises. This is different from aleatoric uncertainty, which refers to the intrinsic stochasticity of the problem [92, 96, 121, 62, 36], or predictive uncertainty which captures the sum of epistemic and aleatoric uncertainties (but not their dissection) [94, 97, 12, 3, 22]. Regarding epistemic uncertainty, a related line of study is deep ensemble that aggregates predictions from multiple independent training replications [81, 73, 41, 8, 94]. This approach, as we will make clear later, can reduce and potentially quantify procedural variability, but a naive use would require demanding retraining effort and does not address data variability. Another line is the Bayesian UQ approach on neural networks [44, 2]. This regards network weights as parameters subject to common priors such as Gaussian. Because of the computation difficulties in exact inference, an array of studies investigate efficient approximate inference approaches to estimate the posteriors [43, 49, 16, 33, 32, 95, 79, 56]. While powerful, these approaches nonetheless possess inference error that could be hard to quantify, and ultimately finding rigorous guarantees on the performance of these approximate posteriors remains open to our best knowledge.

## 2 Statistical Framework of Uncertainty

We first describe our learning setting and define uncertainties in our framework. Suppose the input-output pair $(X, Y)$ is a random vector following an unknown probability distribution $\pi$ on $\mathcal{X} \times \mathcal{Y}$, where $X \in \mathcal{X} \subset \mathbb{R}^d$ is an input and $Y \in \mathcal{Y} \subset \mathbb{R}$ is the response. Let the marginal distribution of $X$ be $\pi_X(x)$ and the conditional distribution of $Y$ given $X$ be $\pi_{Y|X}(y|x)$. Given a set of training data $\mathcal{D} = \{(x_1, y_1), (x_2, y_2), ..., (x_n, y_n)\}$ drawn independent and identically distributed (i.i.d.) from $\pi$ (we write $\boldsymbol{x} = (x_1, ..., x_n)^T$ and $\boldsymbol{y} = (y_1, ..., y_n)^T$ for short), we build a prediction model $h : \mathcal{X} \to \mathcal{Y}$ that best approximates $Y$ given $X$. Let $\pi_{\mathcal{D}}$ or $\pi_n$ denote the empirical distribution associated with the training data $\mathcal{D}$, where $\pi_n$ is used to emphasize the sample size dependence in asymptotic results.

To this end, provided a loss function $\mathcal{L} : \mathcal{Y} \times \mathbb{R} \to \mathbb{R}$, we denote the population risk $R_\pi(h) := \mathbb{E}_{(X,Y) \sim \pi}[\mathcal{L}(h(X), Y)]$. If $h$ is allowed to be any possible functions, the best prediction model is the *Bayes predictor* [61, 93]: $h_B^*(X) \in \mathrm{argmin}_{y \in \mathcal{Y}} \mathbb{E}_{Y \sim \pi_{Y|X}}[\mathcal{L}(y, Y)|X]$. With finite training data of size $n$, classical statistical learning suggests finding $\hat{h}_n \in \mathcal{H}$, where $\mathcal{H}$ denotes a hypothesis class that minimizes the empirical risk, i.e., $\hat{h}_n \in \mathrm{argmin}_{h \in \mathcal{H}} R_{\pi_{\mathcal{D}}}(h)$. This framework fits methods such as linear or kernel ridge regression (Appendix B). However, it is not feasible for deep learning due to non-convexity and non-identifiability. Instead, in practice, gradient-based optimization methods are used, giving rise to $\hat{h}_{n,\gamma}$ as a variant of $\hat{h}_n$, where the additional variable $\gamma$ represents the randomness in the training procedure. It is worth mentioning that this randomness generally depends on the empirical data $\mathcal{D}$, and thus we use $P_{\gamma|\mathcal{D}}$ to represent the distribution of $\gamma$ conditional on $\mathcal{D}$.

Furthermore, we consider $\hat{h}_n^* = \mathrm{aggregate}(\{\hat{h}_{n,\gamma} : \gamma \sim P_{\gamma|\mathcal{D}}\})$ where "aggregate" means an idealized aggregation approach to remove the training randomness in $\hat{h}_{n,\gamma}$ (known as ensemble learning [73, 18, 17, 87, 45]). A prime example in deep learning is deep ensemble [73] that will be detailed in the sequel. Finally, we denote $h^* = \lim_{n \to \infty} \hat{h}_n^*$ as the grand "limiting" predictor with infinite samples. The exact meaning of "lim" will be clear momentarily.

Under this framework, epistemic uncertainty, that is, errors coming from the inadequacy of the model or data noises, can be dissected into the following three sources, as illustrated in Figure 1. Additional discussions on other types of uncertainty are in the Appendix A for completeness.

**Model approximation error.** $\mathrm{UQ}_{AE} = h^* - h_B^*$. This discrepancy between $h_B^*$ and $h^*$ arises from the inadequacy of the hypothesis class $\mathcal{H}$. For an over-parameterized sufficiently wide neural network

$\mathcal{H}$, this error is usually negligible thanks to the universal approximation power of neural networks for any continuous functions [28, 58, 53] or Lebesgue-integrable functions [88].

**Data variability.** $UQ_{DV} = \hat{h}_n^* - h^*$. This measures the representativeness of the training dataset, which is the most standard epistemic uncertainty in classical statistics [110].

**Procedural variability.** $UQ_{PV} = \hat{h}_{n,\gamma} - \hat{h}_n^*$. This arises from the randomness in the training process for a single network $\hat{h}_{n,\gamma}$, which is present even with deterministic or infinite data. The randomness comes from the initialization of the network parameters, and also data ordering and possibly training time when running stochastic gradient descent with finite training epochs.

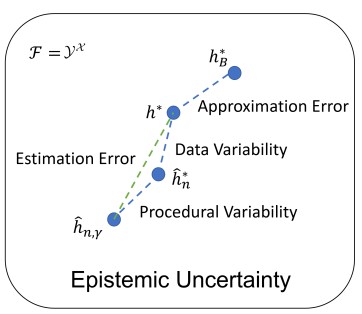

Figure 1: Three sources of epistemic uncertainty.

## 3 Quantifying Epistemic Uncertainty

We use a frequentist framework and, for a given $x$, we aim to construct a confidence interval for the "best" predictor $h^*(x)$. As discussed in Section 1, the over-parametrized and non-convex nature of neural networks defies conventional statistical techniques and moreover introduces procedural variability that makes inference difficult.

We focus on over-parameterized neural networks, that is, neural networks whose width is sufficiently large, while the depth can be arbitrary such as two [120, 107]. Over-parameterized neural networks give the following two theoretical advantages. First, this makes model approximation error negligible and a confidence interval for $h^*(x)$ also approximately applies to $\hat{h}_B^*(x)$. Second, the NTK theory [64] implies a phenomenon that the network evolves essentially as a "linear model" under gradient descent, and thus the resulting predictor behaves like a shifted kernel ridge regression whose kernel is the NTK [7, 80, 54, 59, 118] (detailed in Appendix C). However, to our knowledge, there is no off-the-shelf result that exactly matches our need for the epistemic uncertainty task, so we describe it below. Consider the following regression problem:

$$\hat{R}_n(f_\theta, \theta^b) = \frac{1}{n}\sum_{i=1}^n (f_\theta(x_i) - y_i)^2 + \lambda_n\|\theta - \theta^b\|_2^2. \qquad (1)$$

where $\theta$ is the network parameters to be trained, $\theta^b$ is the initial network parameters, and $\lambda_n > 0$ is the regularization hyper-parameter which may depend on the data size. We add regularization $\lambda_n$ to this problem, which is slightly different from previous work [80] without the regularization $\lambda_n$. This can guarantee stable computation of the inversion of the NTK Gram matrix and can be naturally linked to kernel ridge regression, which will be introduced shortly in Proposition 3.1. We assume the network adopts the NTK parametrization with parameters randomly initialized using He initialization [55], and it is sufficiently wide to ensure the linearized neural network assumption holds; See Appendix C for details. Moreover, we assume that the network is trained using the loss function in (1) via continuous-time gradient flow by feeding the entire training data and using sufficient training time $(t \to \infty)$. In this sense, the uncertainty of data ordering and training time vanishes, making the random initialization the only uncertainty in procedural variability. The above specifications are formally summarized in Assumption C.2. With the above specifications, we have:

**Proposition 3.1** (Proposition C.3)**.** *Suppose that Assumption C.2 holds. Then the final trained network, conditional on the initial network $s_{\theta^b}(x)$, is given by*

$$\hat{h}_{n,\theta^b}(x) = s_{\theta^b}(x) + \boldsymbol{K}(x,\boldsymbol{x})^T(\boldsymbol{K}(\boldsymbol{x},\boldsymbol{x}) + \lambda_n n\boldsymbol{I})^{-1}(\boldsymbol{y} - s_{\theta^b}(\boldsymbol{x})), \qquad (2)$$

*where in the subscript of $\hat{h}_{n,\theta^b}$, $n$ represents $n$ training data, $\theta^b$ represents an instance of the initial network parameters drawn from the standard Gaussian distribution $P_{\theta^b} = \mathcal{N}(0, \boldsymbol{I}_p)$ where the dimension of $\theta^b$ is $p$ (He initialization); $\boldsymbol{K}(\boldsymbol{x},\boldsymbol{x}) := (K(x_i,x_j))_{i,j=1,...,n}$ and $\boldsymbol{K}(x,\boldsymbol{x}) := (K(x,x_1),K(x,x_2),...,K(x,x_n))^T$ where $K$ is the (population) NTK. This implies that $\hat{h}_{n,\theta^b}$ is the solution to the following kernel ridge regression:*

$$s_{\theta^b} + \arg\min_{g\in\bar{\mathcal{H}}} \frac{1}{n}\sum_{i=1}^n (y_i - s_{\theta^b}(x_i) - g(x_i))^2 + \lambda_n\|g\|_{\bar{\mathcal{H}}}^2$$

*where $\bar{\mathcal{H}}$ is the reproducing kernel Hilbert space (RKHS) constructed from the NTK $K(x, x')$.*

Proposition 3.1 shows that the shifted kernel ridge regressor using NTK with a shift from an initial function $s_{\theta^b}$ is exactly the linearized neural network regressor that starts from the initial network $s_{\theta^b}$. It also reveals how procedural variability enters into the neural network prediction outcomes, highlighting the need to regard random initialization as an inevitable uncertainty component in neural networks. We provide details and deviation of Proposition 3.1 in Appendix C.

## 3.1 Challenges from the Interplay of Procedural and Data Variabilities

First, we will discuss existing challenges in quantifying and reducing uncertainty to motivate our main approach based on the PNC predictor under the NTK framework. To this end, deep ensemble [81, 73, 41, 8, 94] is arguably the most common ensemble approach in deep learning to reduce procedural variability. [8] shows that deep ensemble achieves the best performance compared with a wide range of other ensemble methods, and employing more networks in deep ensemble can lead to better performance. Specifically, the *deep ensemble predictor* [73] $\hat{h}_n^m(x)$ is defined as: $\hat{h}_n^m(x) := \frac{1}{m} \sum_{i=1}^m \hat{h}_{n,\theta_i^b}(x)$ where $m$ is the number of networks in the ensemble, $\hat{h}_{n,\theta_i^b}(x)$ is the independently trained network with initialization $\theta_i^b$ (with the same training data $\mathcal{D}$), and $\theta_1^b, ..., \theta_m^b$ are i.i.d. samples drawn from $P_{\theta^b}$. We also introduce $\hat{h}_n^*(x) := \mathbb{E}_{P_{\theta^b}}[\hat{h}_{n,\theta^b}(x)]$ as the expectation of $\hat{h}_{n,\theta^b}(x)$ with respect to $\theta^b \sim P_{\theta^b}$. Taking $m \to \infty$ and using the law of large numbers, we have $\lim_{m\to\infty} \hat{h}_n^m(x) = \hat{h}_n^*(x)$ *a.s.*. So $\hat{h}_n^*(x)$ behaves as an *idealized* deep ensemble predictor with infinitely many independent training procedures. Using Proposition 3.1 and the linearity of kernel ridge regressor with respect to data (Appendix B), we have

$$\hat{h}_n^m(x) = \frac{1}{m} \sum_{i=1}^m s_{\theta_i^b}(x) + \boldsymbol{K}(x, \boldsymbol{x})^T (\boldsymbol{K}(\boldsymbol{x}, \boldsymbol{x}) + \lambda_n n \boldsymbol{I})^{-1} \left( \boldsymbol{y} - \frac{1}{m} \sum_{i=1}^m s_{\theta_i^b}(\boldsymbol{x}) \right)$$

$$\hat{h}_n^*(x) = \mathbb{E}_{P_{\theta^b}}[\hat{h}_{n,\theta^b}(x)] = \bar{s}(x) + \boldsymbol{K}(x, \boldsymbol{x})^T (\boldsymbol{K}(\boldsymbol{x}, \boldsymbol{x}) + \lambda_n n \boldsymbol{I})^{-1} (\boldsymbol{y} - \bar{s}(\boldsymbol{x})) \qquad (3)$$

where $\bar{s}(x) = \mathbb{E}_{P_{\theta^b}}[s_{\theta^b}(x)]$ is the expectation of the initial network output $s_{\theta^b}(x)$ with respect to the the distribution of the initialization parameters $P_{\theta^b} = \mathcal{N}(0, \boldsymbol{I}_p)$. It is easy to see that $\mathbb{E}_{P_{\theta^b}}[\hat{h}_n^m(x)] = \hat{h}_n^*(x)$ and $\mathrm{Var}_{P_{\theta^b}}(\hat{h}_n^m(x)) = \frac{1}{m}\mathrm{Var}_{P_{\theta^b}}(\hat{h}_{n,\theta^b}(x))$ where $\mathrm{Var}_{P_{\theta^b}}$ is the variance with respect to the random initialization. As for the total variance:

**Proposition 3.2.** *We have*

$$Var(\hat{h}_n^m(x)) = Var(\hat{h}_n^*(x)) + \frac{1}{m}\mathbb{E}[Var(\hat{h}_{n,\theta^b}(x)|\mathcal{D})] \leq Var(\hat{h}_n^*(x)) + \mathbb{E}[Var(\hat{h}_{n,\theta^b}(x)|\mathcal{D})] = Var(\hat{h}_{n,\theta^b}(x)).$$

*where the variance is taken with respect to both the data $\mathcal{D}$ and the random initialization $P_{\theta^b}$.*

Proposition 3.2 shows that deep ensemble improves the statistical profile of a single model by reducing its procedural variability by a factor $\frac{1}{m}$ (but not the data variability), and achieving this reduction requires $m$ training times. To quantify the epistemic uncertainty that contains two variabilities from data and procedure, we may employ resampling approaches such as "bootstrap on a deep ensemble". This would involve two layers of sampling, the outer being the resampling of data, and the inner being the retraining of base networks with different initializations. In other words, this nested sampling amounts to a *multiplicative* amount of training effort in a large number of outer bootstrap resamples and the $m$ inner retaining per resample, leading to a huge computational burden. Moreover, the data variability and procedural variability in neural networks are dependent, making the above approach delicate. More discussion about this issue is presented in Section D.

In the following, we introduce our PNC framework that can bypass the above issues in that:

**Uncertainty reduction.** We train one single network and *one* additional auxiliary network to completely remove the procedural variability. This is in contrast to the deep ensemble approach that trains $m$ networks and only reduces the procedural variability by an $m$-factor.

**Uncertainty quantification.** We resolve the computational challenges in "bootstrap on a deep ensemble", by combining PNC predictors with low-budget inference tools that require only as low as *four* network trainings.

## 3.2 PNC Predictor and Procedural Variability Removal

We first develop a computationally efficient approach to obtain $\hat{h}_n^*(x)$, the idealized deep ensemble predictor that is free of procedural variability. We term our approach the procedural-noise-correcting (PNC) predictor, whose pseudo-code is given in Algorithm 1. This predictor consists of a difference between essentially two neural network outcomes, $\hat{h}_{n,\theta^b}(x)$ which is the original network trained using one initialization, and $\hat{\phi}'_{n,\theta^b}(x)$ that is trained on a suitably artificially labeled dataset. More precisely, this dataset applies label $\bar{s}(x_i)$ to $x_i$, where $\bar{s}$ is the expected output of an *untrained* network with random initialization. Also note that labeling this artificial dataset does not involve any training process and, compared with standard network training, the only additional running time in the PNC predictor is to train this single artificial-label-trained network.

---

**Algorithm 1** Procedural-Noise-Correcting (PNC) Predictor

**Input:** Training data $\mathcal{D} = \{(x_1, y_1), (x_2, y_2), ..., (x_n, y_n)\}$.
**Procedure: 1.** Draw $\theta^b \sim P_{\theta^b} = \mathcal{N}(0, \boldsymbol{I}_p)$ under NTK parameterization. Train a standard base network with data $\mathcal{D}$ and the initialization parameters $\theta^b$, which outputs $\hat{h}_{n,\theta^b}(\cdot)$ in (3).
**2.** Let $\bar{s}(x) = \mathbb{E}_{P_{\theta^b}}[s_{\theta^b}(x)]$. For each $x_i$ in $\mathcal{D}$, generate its "artificial" label $\bar{s}(x_i) = \mathbb{E}_{P_{\theta^b}}[s_{\theta^b}(x_i)]$. Train an auxiliary neural network with data $\{(x_1, \bar{s}(x_1)), (x_2, \bar{s}(x_2)), ..., (x_n, \bar{s}(x_n))\}$ and the initialization parameters $\theta^b$ (the same one as in Step 1.), which outputs $\hat{\phi}'_{n,\theta^b}(\cdot)$. Subtracting $\bar{s}(\cdot)$, we obtain $\hat{\phi}_{n,\theta^b}(\cdot) = \hat{\phi}'_{n,\theta^b}(\cdot) - \bar{s}(\cdot)$.
**Output:** At point $x$, output $\hat{h}_{n,\theta^b}(x) - \hat{\phi}_{n,\theta^b}(x)$.

---

The development of our PNC predictor is motivated by the challenge of computing $\hat{h}_n^*(x)$ (detailed below). To address this challenge, we characterize the procedural noise instead, which is given by

$$\hat{\phi}_{n,\theta^b}(\cdot) := \hat{h}_{n,\theta^b}(x) - \hat{h}_n^*(x) = s_{\theta^b}(x) - \bar{s}(x) + \boldsymbol{K}(x, \boldsymbol{x})^T (\boldsymbol{K}(\boldsymbol{x}, \boldsymbol{x}) + \lambda_n n \boldsymbol{I})^{-1} (\bar{s}(\boldsymbol{x}) - s_{\theta^b}(\boldsymbol{x})).$$
(4)

By Proposition 3.1, the closed-form expression of $\hat{\phi}_{n,\theta^b}$ in (4) corresponds exactly to the artificial-label-trained network in Step 2 of Algorithm 1. Our artificial-label-trained network is thereby used to quantify the procedural variability directly. This observation subsequently leads to:

**Theorem 3.3** (PNC). *Suppose that Assumption C.2 holds. Then the output of the PNC predictor (Algorithm 1) is exactly $\hat{h}_n^*(x)$ given in* (3).

We discuss two approaches to compute $\bar{s}(x)$ in Step 2 of Algorithm 1 and their implications: 1) Under He initialization, $s_{\theta^b}(x)$ is a zero-mean Gaussian process in the infinite width limit [80], and thus we may set $\bar{s} \equiv 0$ for simplicity. Note that even if we set $\bar{s} \equiv 0$, it does *not* imply that the artificial-label-trained neural network in Step 2 of Algorithm 1 will output a zero-constant network whose parameters are all zeros. In fact, neural networks are excessively non-convex and have many nearly global minima. Starting from an initialization parameter $\theta^b$, gradient descent on this artificial-label-trained neural network will find a global minimum that is close to the $\theta^b$ but not "zero" even if "zero" is indeed one of its nearly global minima ("nearly" in the sense of ignoring the negligible regularization term) [35, 118, 23]. This phenomenon can also be observed in Proposition 3.1 by plugging in $\boldsymbol{y} = \boldsymbol{0}$. Hence, the output depends on random initialization in addition to training data, and our artificial-label-trained network is designed to capture this procedural variability. 2) An alternative approach that does not require specific initialization is to use Monte Carlo integration: $\bar{s}(x) = \lim_{N \to \infty} \frac{1}{N} \sum_{i=1}^N s_{\theta_i^b}(x)$ where $\theta_i^b$ are i.i.d. from $P_{\theta^b}$. When $N$ is finite, it introduces procedural variance that vanishes at the order $N^{-1}$. Since this computation does not involve any training process and is conducted in a rapid manner practically, $N \gg n$ can be applied to guarantee that the procedural variance in computing $\bar{s}(x)$ is negligible compared with the data variance at the order $n^{-1}$. We practically observe that both approaches work similarly well.

Finally, we provide additional remarks on why $\hat{h}_n^*(x)$ cannot be computed easily except using our Algorithm 1. First, the following two candidate approaches encounter computational issues: 1) One may use deep ensemble with sufficiently many networks in the ensemble. However, this approach is time-consuming as $m$ networks in the ensemble mean $m$-fold training times. $m$ is typically as small as five in practice [73] so it cannot approximate $\hat{h}_n^*(x)$ well. 2) One may use the closed-

form expression of $\hat{h}_n^*$ in (3), which requires computing the NTK $K(x, x')$ and the inversion of the NTK Gram matrix $\boldsymbol{K}(\boldsymbol{x}, \boldsymbol{x})$. $K(x, x')$ is recursively defined and does not have a simple form for computation, which might be addressed by approximating it with the empirical NTK $\hat{K}_\theta(x, x')$ numerically (See Appendix C). However, the inversion of the NTK Gram matrix gives rise to a more serious computational issue: the dimension of the NTK Gram matrix is large on large datasets, making the matrix inversion very time-consuming. Another approach that seems plausible is that: 3) One may initialize one network that is equivalent to $\bar{s}(x)$ and then train it. However, this approach *cannot* obtain $\hat{h}_n^*(x)$ based on Proposition 3.1 because Proposition 3.1 requires random initialization. If one starts from a deterministic network initialization such as a zero-constant network, then the NTK theory underpinning the linearized training behavior of over-parameterized neural networks breaks down and the resulting network cannot be described by Proposition 3.1.

## 3.3 Constructing Confidence Intervals from PNC Predictors

We construct confidence intervals for $h^*(x)$ leveraging our PNC predictor in Algorithm 1. To handle data variability, two lines of works borrowed from classical statistics may be considered. First is an analytical approach using the delta method for asymptotic normality, which involves computing the influence function [51, 38] that acts as the functional gradient of the predictor with respect to the data distribution. It was introduced in modern machine learning for understanding a training point's effect on a model's prediction [72, 71, 13]. The second is to use resampling [37, 30, 102], such as the bootstrap or jackknife, to avoid explicit variance computation. The classical resampling method requires sufficiently large resample replications and thus incurs demanding resampling effort. For instance, the jackknife approach requires the number of training times to be the same as the training data size, which is very time-consuming and barely feasible for neural networks. Standard bootstrap requires a sufficient number of resampling and retraining to produce accurate resample quantiles. Given these computational bottlenecks, we consider utilizing light-computation resampling alternatives, including batching [47, 99, 100] and the so-called cheap bootstrap method [75, 76], which allows valid confidence interval construction using as few as two model repetitions.

**Large-sample asymptotics of the PNC predictor.** To derive our intervals, we first gain understanding on the large-sample properties of the PNC predictor. Proposition 3.1 shows that $\hat{h}_n^*(x)$ in (3) is the solution to the following empirical risk minimization problem:

$$\hat{h}_n^*(\cdot) = \bar{s}(\cdot) + \min_{g \in \mathcal{H}} \frac{1}{n} \sum_{i=1}^n [(y_i - \bar{s}(x_i) - g(x_i))^2] + \lambda_n \|g\|_{\mathcal{H}}^2. \tag{5}$$

Its corresponding population risk minimization problem (i.e., removing the data variability) is:

$$h^*(\cdot) = \bar{s}(\cdot) + \min_{g \in \mathcal{H}} \mathbb{E}_\pi [(Y - \bar{s}(X) - g(X))^2] + \lambda_0 \|g\|_{\mathcal{H}}^2 \tag{6}$$

where $\lambda_0 = \lim_{n \to \infty} \lambda_n$. To study the difference between the empirical and population risk minimization problems of kernel ridge regression, we introduce the following established result on the asymptotic normality of kernel ridge regression (See Appendix B for details):

**Proposition 3.4** (Asymptotic normality of kernel ridge regression [50])**.** *Let $\mathcal{H}$ be a generic RKHS. Suppose that Assumptions B.3 and B.4 hold. Let $g_{P,\lambda}$ be the solution to the following problem: $g_{P,\lambda} := \min_{g \in \mathcal{H}} \mathbb{E}_P [(Y - g(X))^2] + \lambda \|g\|_{\mathcal{H}}^2$ where $P = \pi_n$ or $\pi$. Then*

$$\sqrt{n}(g_{\pi_n, \lambda_n} - g_{\pi, \lambda_0}) \Rightarrow \mathbb{G} \text{ in } \bar{H}$$

*where $\mathbb{G}$ is a zero-mean Gaussian process and $\Rightarrow$ represents "converges weakly". Moreover, at point $x$, $\sqrt{n}(g_{\pi_n, \lambda_n}(x) - g_{\pi, \lambda_0}(x)) \Rightarrow \mathcal{N}(0, \xi^2(x))$ where $\xi^2(x) = \int_{z \in \mathcal{X} \times \mathcal{Y}} IF^2(z; g_{P,\lambda_0}, \pi)(x) d\pi(z)$ and $IF$ is the influence function of statistical functional $g_{P,\lambda_0}$.*

Next, we apply the above proposition to our problems about $\hat{h}_n^*$. Let $T_1(P)(x)$ be the solution of the following problem $\min_{g \in \bar{\mathcal{H}}} \mathbb{E}_P [(Y - \bar{s}(X) - g(X))^2] + \lambda_0 \|g\|_{\bar{\mathcal{H}}}^2$ that is evaluated at a point $x$. Then we have the following large-sample asymptotic of the PNC predictor, providing the theoretical foundation for our subsequent interval construction approaches.

**Theorem 3.5** (Large-sample asymptotics of the PNC predictor)**.** *Suppose that Assumption C.2 holds. Suppose that Assumptions B.3 and B.4 hold when $Y$ is replaced by $Y - \bar{s}(X)$. Input the training*

*data $\mathcal{D}$ into Algorithm 1 to obtain $\hat{h}_{n,\theta^b}(x) - \hat{\phi}_{n,\theta^b}(x)$. We have*

$$\sqrt{n}\left(\hat{h}_{n,\theta^b}(x) - \hat{\phi}_{n,\theta^b}(x) - h^*(x)\right) \Rightarrow \mathcal{N}(0, \sigma^2(x)), \tag{7}$$

*where*

$$\sigma^2(x) = \int_{z \in \mathcal{X} \times \mathcal{Y}} IF^2(z; T_1, \pi)(x) d\pi(z). \tag{8}$$

*Thus, an asymptotically (in the sense of $n \to \infty$) exact $(1 - \alpha)$-level confidence interval of $h^*(x)$ is $[\hat{h}_{n,\theta^b}(x) - \hat{\phi}_{n,\theta^b}(x) - \frac{\sigma(x)}{\sqrt{n}}q_{1-\frac{\alpha}{2}}, \hat{h}_{n,\theta^b}(x) - \hat{\phi}_{n,\theta^b}(x) + \frac{\sigma(x)}{\sqrt{n}}q_{1-\frac{\alpha}{2}}]$ where $q_\alpha$ is the $\alpha$-quantile of the standard Gaussian distribution $\mathcal{N}(0, 1)$.*

Theorem 3.5 does not indicate the value of $\sigma^2(x)$. In general, $\sigma^2(x)$ is unknown and needs to be estimated. It is common to approximate $IF^2(z; T_1, \pi)$ with $IF^2(z; T_1, \pi_n)$ and set $\hat{\sigma}^2(x) = \frac{1}{n}\sum_{z_i \in \mathcal{D}} IF^2(z_i; T_1, \pi_n)(x)$. This method is known as the infinitesimal jackknife variance estimator [102]. In Appendix B, we derive the exact closed-form expression of the infinitesimal jackknife variance estimation $\hat{\sigma}^2(x)$, and further prove its consistency as well as the statistical coverage guarantee of confidence intervals built upon it. These results could be of theoretical interest. Yet in practice, the computation of $\hat{\sigma}^2(x)$ requires the evaluation of the NTK Gram matrix and its inversion, which is computationally demanding for large $n$ and thus not recommended for practical implementation on large datasets.

In the following, we provide two efficient approaches that avoid explicit estimation of the asymptotic variance as in the infinitesimal jackknife approach, and the computation of the NTK Gram matrix inversion.

**PNC-enhanced batching.** We propose an approach for constructing a confidence interval that is particularly useful for large datasets, termed *PNC-enhanced batching*. The pseudo-code is given in Algorithm 2. Originating from simulation analysis [47, 99, 100], the key idea of batching is to construct a self-normalizing $t$-statistic that "cancels out" the unknown variance, leading to a valid confidence interval without explicitly needing to compute this variance. It can be used to conduct inference on serially dependent simulation outputs where the standard error is difficult to compute analytically. Previous studies have demonstrated the effectiveness of batching on the use of inference for Markov chain Monte Carlo [46, 40, 66] and also the so-called input uncertainty problem [48]. Its application in deep learning uncertainty quantification was not revealed in previous work, potentially due to the additional procedural variability. Integrating it with the PNC predictor, PNC-enhanced batching is very efficient and meanwhile possesses asymptotically exact coverage of its confidence interval, as stated below.

---
**Algorithm 2** PNC-Enhanced Batching

**Input:** Training dataset $\mathcal{D}$ of size $n$. The number of batches $m' \geq 2$.
**Procedure: 1.** Split the training data $\mathcal{D}$ into $m'$ batches and input each batch in Algorithm 1 to output $\hat{h}^j_{n',\theta^b}(x) - \hat{\phi}^j_{n',\theta^b}(x)$ for $j \in [m']$, where $n' = \frac{n}{m'}$.
**2.** Compute $\psi_B(x) = \frac{1}{m'}\sum_{j=1}^{m'}\left(\hat{h}^j_{n',\theta^b}(x) - \hat{\phi}^j_{n',\theta^b}(x)\right)$,
and $S_B(x)^2 = \frac{1}{m'-1}\sum_{j=1}^{m'}\left(\hat{h}^j_{n',\theta^b}(x) - \hat{\phi}^j_{n',\theta^b}(x) - \psi_B(x)\right)^2$.
**Output:** At point $x$, output $\psi_B(x)$ and $S_B(x)^2$.

---

**Theorem 3.6** (Exact coverage of PNC-enhanced batching confidence interval)**.** *Suppose that Assumption C.2 holds. Suppose that Assumptions B.3 and B.4 hold when $Y$ is replaced by $Y - \bar{s}(X)$. Choose any $m' \geq 2$ in Algorithm 2. Then an asymptotically exact $(1 - \alpha)$-level confidence interval of $h^*(x)$ is $[\psi_B(x) - \frac{S_B(x)}{\sqrt{m'}}q_{1-\frac{\alpha}{2}}, \psi_B(x) + \frac{S_B(x)}{\sqrt{m'}}q_{1-\frac{\alpha}{2}}]$, where $q_\alpha$ is the $\alpha$-quantile of the $t$ distribution $t_{m'-1}$ with degree of freedom $m' - 1$.*

**PNC-enhanced cheap bootstrap.** We propose an alternative approach for constructing a confidence interval that works for large datasets but is also suitable for smaller ones, termed *PNC-enhanced cheap bootstrap*. The pseudo-code is given in Algorithm 3. Cheap bootstrap [75, 76, 74] is a modified bootstrap procedure with substantially less retraining effort than conventional bootstrap methods, via leveraging the asymptotic independence between the original and resample estimators

and asymptotic normality. Note that our proposal is fundamentally different from the naive use of bootstrap or bagging when additional randomness appears [73, 81, 45], which mixes the procedural and data variabilities and does not directly provide confidence intervals. Like PNC-enhanced batching, PNC-enhanced cheap bootstrap also avoids the explicit estimation of the asymptotic variance. The difference between the above two approaches is that PNC-enhanced batching divides data into a small number of batches, and thus is suggested for large datasets while PNC-enhanced cheap bootstrap re-selects samples from the entire dataset, hence suited also for smaller datasets. On the other hand, in terms of running time, when $R = m' - 1$, PNC-enhanced cheap bootstrap and PNC-enhanced batching share the same number of network training, but since batching is trained on subsets of the data, the individual network training in PNC-enhanced batching is faster than PNC-enhanced cheap bootstrap. PNC-enhanced cheap bootstrap also enjoys asymptotically exact coverage of its confidence interval, as stated below.

---

**Algorithm 3** PNC-Enhanced Cheap Bootstrap

---

**Input:** Training dataset $\mathcal{D}$ of size $n$. The number of replications $R \geq 1$.
**Procedure: 1.** Input $\mathcal{D}$ in Algorithm 1 to output $\hat{h}_{n,\theta^b}(x) - \hat{\phi}_{n,\theta^b}(x)$.
**2.** For each replication $j \in [R]$, resample $\mathcal{D}$, i.e., independently and uniformly sample with replacement from $\{(x_1, y_1), ..., (x_n, y_n)\}$ $n$ times to obtain $\mathcal{D}^{*j} = \{(x_1^{*j}, y_1^{*j}), ..., (x_n^{*j}, y_n^{*j})\}$. Input $\mathcal{D}^{*j}$ in Algorithm 1 to output $\hat{h}_{n,\theta^b}^{*j}(x) - \hat{\phi}_{n,\theta^b}^{*j}(x)$.
**3.** Compute $\psi_C(x) = \hat{h}_{n,\theta^b}(x) - \hat{\phi}_{n,\theta^b}(x)$,
and $S_C(x)^2 = \frac{1}{R} \sum_{j=1}^{R} \left( \hat{h}_{n,\theta^b}^{*j}(x) - \hat{\phi}_{n,\theta^b}^{*j}(x) - \psi_C(x) \right)^2$.
**Output:** At point $x$, output $\psi_C(x)$ and $S_C(x)^2$.

---

**Theorem 3.7** (Exact coverage of PNC-enhanced cheap bootstrap confidence interval). *Suppose that Assumption C.2 holds. Suppose that Assumptions B.3 and B.4 hold when $Y$ is replaced by $Y - \bar{s}(X)$. Choose any $R \geq 1$ in Algorithm 3. Then an asymptotically exact $(1 - \alpha)$-level confidence interval of $h^*(x)$ is $[\psi_C(x) - S_C(x)q_{1-\frac{\alpha}{2}}, \psi_C(x) + S_C(x)q_{1-\frac{\alpha}{2}}]$ where $q_\alpha$ is the $\alpha$-quantile of the t distribution $t_R$ with degree of freedom $R$.*

## 4 Experiments

We conduct numerical experiments to demonstrate the effectiveness of our approaches.[2] Our proposed approaches are evaluated on the following two tasks: 1) construct confidence intervals and 2) reduce procedural variability to improve prediction. With a known ground-truth regression function, training data are regenerated from the underlying synthetic data generative process. According to the NTK parameterization in Section C, our base network is formed with two fully connected layers with $n \times 32$ neurons in each hidden layer to ensure the network is sufficiently wide and over-parameterized. Detailed optimization specifications are described in Proposition C.3. Our synthetic datasets #1 are generated with the following distributions: $X \sim \text{Unif}([0, 0.2]^d)$ and $Y \sim \sum_{i=1}^{d} \sin(X^{(i)}) + \mathcal{N}(0, 0.001^2)$. The training set $\mathcal{D} = \{(x_i, y_i) : i = 1, ..., n\}$ is formed by drawing i.i.d. samples of $(X, Y)$ from the above distribution with sample size $n$. We consider multiple dimension settings $d = 2, 4, 8, 16$ and data size settings $n = 128, 256, 512, 1024$ to study the effects on different dimensionalities and data sizes. Additional experimental results on more datasets are presented in Appendix F. The implementation details of our experiments are also provided in Appendix F.

**Constructing confidence intervals.** We use $x_0 = (0.1, ..., 0.1)$ as the fixed test point for confidence intervals construction. Let $y_0 = \sum_{i=1}^{d} \sin(0.1)$ be the ground-truth label for $x_0$ without aleatoric noise. Our goal is to construct a confidence interval at $x_0$ for $y_0$. To evaluate the performance of confidence intervals, we set the number of experiments $J = 100$. In each repetition $j \in [J]$, we generate a new training dataset from the same synthetic distribution and construct a new confidence interval $[L_j(x_0), U_j(x_0)]$ with 95% or 90% confidence level, and then check the coverage rate (CR): $\text{CR} = \frac{1}{J} \sum_{j=1}^{J} \mathbf{1}_{L_j(x_0) \leq y_0 \leq U_j(x_0)}$. The primary evaluation of the confidence interval is based on whether its coverage rate is equal to or larger than the desired confidence level. In addition to CR,

---

[2]The source code for experiments is available at `https://github.com/HZ0000/UQforNN`.

Table 1: Confidence interval construction on synthetic datasets #1 with different data sizes $n = 128, 256, 512, 1024$ and different data dimensions $d = 2, 4, 8, 16$. The CR results that attain the desired confidence level 95%/90% are in **bold**.

| | PNC-enhanced batching | | | PNC-enhanced cheap bootstrap | | | DropoutUQ | | |
|---|---|---|---|---|---|---|---|---|---|
| | 95%CI(CR/IW) | 90%CI(CR/IW) | MP | 95%CI(CR/IW) | 90%CI(CR/IW) | MP | 95%CI(CR/IW) | 90%CI(CR/IW) | MP |
| $(d = 2)$ | | | | | | | | | |
| $n = 128$ | **0.98**/0.0437 | **0.95**/0.0323 | 0.1998 | **0.98**/0.0571 | **0.95**/0.0438 | 0.1983 | 0.93/0.0564 | 0.87/0.0480 | 0.2119 |
| $n = 256$ | **0.99**/0.0301 | **0.96**/0.0222 | 0.2004 | **1.00**/0.0376 | **0.95**/0.0289 | 0.1997 | **0.99**/0.0390 | **0.96**/0.0327 | 0.2045 |
| $n = 512$ | **0.97**/0.0211 | **0.95**/0.0156 | 0.2004 | **0.98**/0.0294 | **0.96**/0.0226 | 0.1992 | **0.95**/0.0313 | **0.93**/0.0267 | 0.2049 |
| $n = 1024$ | **0.96**/0.0152 | **0.92**/0.0112 | 0.2003 | **0.97**/0.0205 | **0.94**/0.0157 | 0.2011 | **0.96**/0.0287 | 0.89/0.0244 | 0.2052 |
| $(d = 4)$ | | | | | | | | | |
| $n = 128$ | **0.98**/0.0622 | **0.94**/0.0460 | 0.4012 | **1.00**/0.0820 | **0.95**/0.0629 | 0.4013 | **0.96**/0.0868 | **0.90**/0.0742 | 0.4157 |
| $n = 256$ | **0.98**/0.0411 | **0.95**/0.0304 | 0.3991 | **0.99**/0.0569 | **0.96**/0.0437 | 0.3988 | **0.97**/0.0566 | **0.92**/0.0481 | 0.4071 |
| $n = 512$ | **0.97**/0.0295 | **0.94**/0.0218 | 0.3988 | **0.98**/0.0396 | **0.96**/0.0304 | 0.3989 | **0.96**/0.0375 | **0.90**/0.0316 | 0.4045 |
| $n = 1024$ | **0.97**/0.0213 | **0.95**/0.0158 | 0.3983 | **0.98**/0.0296 | **0.98**/0.0227 | 0.3988 | 0.93/0.0340 | 0.84/0.0290 | 0.4055 |
| $(d = 8)$ | | | | | | | | | |
| $n = 128$ | **0.98**/0.0865 | **0.95**/0.0639 | 0.7999 | **0.98**/0.1084 | **0.98**/0.0832 | 0.7980 | 0.91/0.1348 | 0.87/0.1142 | 0.8245 |
| $n = 256$ | **0.99**/0.0594 | **0.96**/0.0439 | 0.8014 | **1.00**/0.0796 | **0.99**/0.0611 | 0.7957 | 0.88/0.0791 | 0.81/0.0656 | 0.8152 |
| $n = 512$ | **0.98**/0.0393 | **0.94**/0.0290 | 0.7983 | **0.99**/0.0588 | **0.99**/0.0452 | 0.8012 | **0.99**/0.0632 | **0.97**/0.0538 | 0.7998 |
| $n = 1024$ | **0.95**/0.0270 | **0.92**/0.0200 | 0.7981 | **0.99**/0.0393 | **0.96**/0.0302 | 0.7997 | 0.88/0.0421 | 0.82/0.0356 | 0.8040 |
| $(d = 16)$ | | | | | | | | | |
| $n = 128$ | **0.95**/0.1068 | **0.94**/0.0790 | 1.5946 | **0.99**/0.1568 | **0.97**/0.1204 | 1.6057 | **0.99**/0.1565 | **0.96**/0.1313 | 1.6093 |
| $n = 256$ | **0.98**/0.0730 | **0.94**/0.0540 | 1.5966 | **1.00**/0.1137 | **0.98**/0.0873 | 1.5954 | 0.90/0.1072 | 0.86/0.0909 | 1.6177 |
| $n = 512$ | **0.98**/0.0543 | **0.93**/0.0401 | 1.5966 | **0.99**/0.0788 | **0.98**/0.0605 | 1.5972 | 0.86/0.0920 | 0.80/0.0790 | 1.6132 |
| $n = 1024$ | **0.95**/0.0388 | **0.92**/0.0287 | 1.5976 | **0.97**/0.0550 | **0.95**/0.0422 | 1.5980 | 0.87/0.0760 | 0.83/0.0647 | 1.6079 |

we also report the median point of the interval (MP): MP $= \frac{1}{J} \sum_{j=1}^{J} \frac{1}{2}(U_j(x_0) + L_j(x_0))$ and the interval width (IW): IW $= \frac{1}{J} \sum_{j=1}^{J}(U_j(x_0) - L_j(x_0))$. MP reflects the most likely point prediction for $x_0$, while IW reflects the conservativeness of the confidence interval, as a wide interval can easily achieve 95% coverage rate but is not practically preferable. In the confidence interval tasks, we use the dropout-based Bayesian approximate inference (DropoutUQ) [43] as the baseline for comparison since it is one of the most widely-used representatives in Bayesian uncertainty quantification.

Table 1 reports the CR, IW, and MP of 95% and 90% confidence intervals from our proposals and baselines. We have the following observations:

1) The CR values: For our two proposed approaches, in the majority of experiments, CR $\geq 95\%$ for 95% confidence intervals, and CR $\geq 90\%$ for 90% confidence intervals, thus satisfying the coverage requirements, while in very few experiments (in Appendix F), the performances are degraded. Nonetheless, this occasional degeneration appears insignificant and can be potentially explained by the fact that the statistical guarantee of confidence intervals generated by PNC-enhanced batching and cheap bootstrap is in the asymptotic sense. In contrast, DropoutUQ does not have such statistical guarantees. The intervals from DropoutUQ cannot normally maintain a satisfactory coverage rate when they are narrow, although they have a similar or larger interval width as PNC-enhanced batching or cheap bootstrap. Moreover, we notice that the dropout rate has a significant impact on the interval width of DropoutUQ, and thus, additional tuning efforts are demanded for this baseline, while our approaches do not need this level of tuning procedure. These observations demonstrate the robustness and effectiveness of our proposals.

2) The IW values: Overall, the IW values are relatively small for both of our approaches, indicating that our framework is able to successfully generate narrow confidence intervals with high coverage rates. Moreover, as shown, the IW values from both of our approaches decrease roughly at the rate $n^{-\frac{1}{2}}$ along with an increased training data size $n$, which corroborates with our theoretical results well (Section 3.3). In comparison with PNC-enhanced cheap bootstrap, we observe that PNC-enhanced batching tends to generate narrower intervals and has less training time per network, as only a subset of training data is involved in one training trial. In general, PNC-enhanced batching requires data splits, making its accuracy vulnerable to small sample sizes, and thus leading to less stable performance compared with PNC-enhanced cheap bootstrap. Therefore, we recommend employing PNC-enhanced cheap bootstrap on small datasets and applying PNC-enhanced batching on datasets with sufficient training samples.

3) The MP values: MP appears consistent with the ground-truth label $y_0$ in all experiments, showing that our confidence intervals can accurately capture the ground-truth label on average under multiple problem settings.

Table 2: Reducing procedural variability to improve prediction on synthetic datasets #1 with different data sizes $n = 128, 256, 512, 1024$ and different data dimensions $d = 2, 4, 8, 16$. The best MSE results are in **bold**.

| MSE | One base network | PNC predictor | Deep ensemble (5 networks) | Deep ensemble (2 networks) |
|---|---|---|---|---|
| $(d = 2)$ | | | | |
| $n = 128$ | $(6.68 \pm 2.74) \times 10^{-4}$ | $\mathbf{(3.68 \pm 1.84) \times 10^{-4}}$ | $(4.22 \pm 1.74) \times 10^{-4}$ | $(6.28 \pm 2.74) \times 10^{-4}$ |
| $n = 256$ | $(2.38 \pm 1.06) \times 10^{-4}$ | $\mathbf{(6.22 \pm 2.01) \times 10^{-5}}$ | $(8.86 \pm 4.94) \times 10^{-5}$ | $(1.25 \pm 0.67) \times 10^{-4}$ |
| $n = 512$ | $(1.03 \pm 0.98) \times 10^{-4}$ | $\mathbf{(2.06 \pm 0.72) \times 10^{-5}}$ | $(3.32 \pm 1.10) \times 10^{-5}$ | $(5.11 \pm 3.33) \times 10^{-5}$ |
| $n = 1024$ | $(6.98 \pm 4.77) \times 10^{-5}$ | $\mathbf{(8.92 \pm 5.69) \times 10^{-6}}$ | $(1.72 \pm 0.77) \times 10^{-5}$ | $(5.75 \pm 2.28) \times 10^{-5}$ |
| $(d = 4)$ | | | | |
| $n = 128$ | $(2.11 \pm 1.49) \times 10^{-3}$ | $\mathbf{(1.18 \pm 0.25) \times 10^{-3}}$ | $(1.53 \pm 0.60) \times 10^{-3}$ | $(1.83 \pm 0.80) \times 10^{-3}$ |
| $n = 256$ | $(8.82 \pm 3.26) \times 10^{-4}$ | $\mathbf{(4.09 \pm 0.85) \times 10^{-4}}$ | $(4.22 \pm 2.01) \times 10^{-4}$ | $(5.47 \pm 1.91) \times 10^{-4}$ |
| $n = 512$ | $(5.35 \pm 1.91) \times 10^{-4}$ | $\mathbf{(1.92 \pm 0.53) \times 10^{-4}}$ | $(2.88 \pm 1.80) \times 10^{-4}$ | $(3.99 \pm 1.87) \times 10^{-4}$ |
| $n = 1024$ | $(2.23 \pm 0.83) \times 10^{-4}$ | $\mathbf{(4.22 \pm 0.43) \times 10^{-5}}$ | $(8.50 \pm 2.39) \times 10^{-5}$ | $(1.65 \pm 0.57) \times 10^{-4}$ |
| $(d = 8)$ | | | | |
| $n = 128$ | $(4.07 \pm 1.04) \times 10^{-3}$ | $\mathbf{(2.54 \pm 0.29) \times 10^{-3}}$ | $(2.73 \pm 0.86) \times 10^{-3}$ | $(3.27 \pm 0.96) \times 10^{-3}$ |
| $n = 256$ | $(2.13 \pm 0.70) \times 10^{-3}$ | $\mathbf{(1.05 \pm 0.18) \times 10^{-3}}$ | $(1.34 \pm 0.33) \times 10^{-3}$ | $(1.48 \pm 0.38) \times 10^{-3}$ |
| $n = 512$ | $(1.36 \pm 0.34) \times 10^{-3}$ | $\mathbf{(5.04 \pm 0.70) \times 10^{-4}}$ | $(7.40 \pm 1.35) \times 10^{-4}$ | $(1.08 \pm 0.40) \times 10^{-3}$ |
| $n = 1024$ | $(8.54 \pm 2.23) \times 10^{-4}$ | $\mathbf{(2.02 \pm 0.24) \times 10^{-4}}$ | $(3.79 \pm 1.01) \times 10^{-4}$ | $(5.91 \pm 1.87) \times 10^{-4}$ |
| $(d = 16)$ | | | | |
| $n = 128$ | $(9.03 \pm 1.64) \times 10^{-3}$ | $\mathbf{(6.00 \pm 0.83) \times 10^{-3}}$ | $(6.37 \pm 0.67) \times 10^{-3}$ | $(7.47 \pm 1.29) \times 10^{-3}$ |
| $n = 256$ | $(6.76 \pm 1.79) \times 10^{-3}$ | $\mathbf{(4.19 \pm 0.94) \times 10^{-3}}$ | $(4.83 \pm 1.14) \times 10^{-3}$ | $(5.46 \pm 1.49) \times 10^{-3}$ |
| $n = 512$ | $(4.19 \pm 0.51) \times 10^{-3}$ | $\mathbf{(1.60 \pm 0.35) \times 10^{-3}}$ | $(2.05 \pm 0.30) \times 10^{-3}$ | $(2.87 \pm 0.34) \times 10^{-3}$ |
| $n = 1024$ | $(3.16 \pm 0.35) \times 10^{-3}$ | $\mathbf{(7.44 \pm 0.99) \times 10^{-4}}$ | $(1.20 \pm 0.13) \times 10^{-3}$ | $(1.91 \pm 0.20) \times 10^{-3}$ |

**Reduce procedural variability to improve prediction.** In this part, we illustrate that our proposed PNC preditor (Algorithm 1) can achieve better prediction by reducing procedural variability. We use empirical mean square error (MSE) to evaluate the prediction performance. Specifically, the empirical MSE is computed as $\text{MSE}(h) := \frac{1}{N_{te}} \sum_{i=1}^{N_{te}} (h(x_i') - y_i')^2$, where $(x_i', y_i'), i = 1, ..., N_{te}$ are i.i.d. test data independent of the training. We set $N_{te} = 2048$ for test performance evaluation in all experiments. We compare the prediction performance of our proposed PNC predictor with the following approaches: a base network, i.e., a standard network training with a single random initialization, and the deep ensemble approach with different numbers ($m$) of networks in the ensemble [73]. In deep ensemble, we consider $m = 2$ as it has a similar running time as our PNC (with one additional network training), and $m = 5$ as it is widely used in previous work [73, 94]. All networks share the same hyperparameters and specifications.

Table 2 reports the sample mean and sample standard deviation of MSE from our proposed PNC predictor and other baseline approaches with 10 experimental repetition times. As shown, the proposed PNC achieves notably smaller MSE than baselines with similar computational costs (the base network and deep ensemble with two networks) in all experiments. Moreover, it achieves better or analogous results compared with the deep ensemble with 5 networks, but the running time of the latter is 2.5 times as much as that of PNC. It also has relatively small sample standard deviations, showing its capacity to reduce variability in networks effectively. These results verify that our proposed PNC can successfully produce better prediction and simultaneously bears significantly reduced computational costs compared to the classical deep ensemble.

## 5  Concluding Remarks

In this study, we propose a systematic epistemic uncertainty assessment framework with simultaneous statistical coverage guarantees and low computation costs for over-parametrized neural networks in regression. Benefiting from our proposed PNC approach, our study shows promise to characterize and eliminate procedural uncertainty by introducing only one artificial-label-trained network. Integrated with suitable light-computation resampling methods, we provide two effective approaches to construct asymptotically exact-coverage confidence intervals using few model retrainings. Our evaluation results in different settings corroborate our theory and show that our approach can generate confidence intervals that are narrow and have satisfactory coverages simultaneously. In the future, we will extend our current framework to general loss functions that can potentially handle classification tasks to open up opportunities for broader applications.

## Acknowledgments and Disclosure of Funding

This work has been supported in part by the National Science Foundation under grants CAREER CMMI-1834710 and IIS-1849280, and the Cheung-Kong Innovation Doctoral Fellowship. The authors thank the anonymous reviewers for their constructive comments which have helped greatly improve the quality of our paper.

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

# Appendices

We provide further results and discussions in this supplementary material. Section A discusses aleatoric uncertainty and predictive uncertainty. Section B discusses statistical inference for kernel-based regression and, in particular, the asymptotic normality of kernel-based regression. Section C discusses the training of the linearized neural networks based on the NTK theory. In particular, we show in Proposition 3.1 that the shifted kernel ridge regressor using NTK with a shift from an initial function $s_{\theta^b}$ is exactly the linearized neural network regressor that starts from the initial network $s_{\theta^b}$. Section D provides additional remarks to explain some of the details in the main paper. Section E presents the proofs for results in the paper. Section F presents experimental details and more experimental results.

## Appendix A    Other Types of Uncertainty

Back in 1930, [39] first introduced a formal distinction between aleatory and epistemic uncertainty in statistics [52], while in modern machine learning, their distinction and connection were investigated in [101] and further extended to deep learning models [67, 33]. In [94], the differences between procedural and data variabilities in epistemic uncertainty were intuitively described; however, no rigorous definition or estimation method was provided. Our previous draft [60] discussed intuitive approaches for quantifying procedural and data variabilities, which some of the ideas in this work originate from. In the following, we present an additional discussion on other types of uncertainty for the sake of completeness. Section A.1 presents aleatoric uncertainty and Section A.2 presents predictive uncertainty.

### A.1    Aleatoric Uncertainty

We note that if we could remove all the epistemic uncertainty, i.e., letting $\mathrm{UQ}_{EU} = 0$, the best predictor we could get is the Bayes predictor $h_B^*$. However, $h_B^*$, as a point estimator, is not able to capture the randomness in $\pi_{Y|X}$ if it is not a point mass distribution.

The easiest way to think of aleatoric uncertainty is that it is captured by $\pi_{Y|X}$. At the level of the realized response or label value, this uncertainty is represented by

$$\mathrm{UQ}_{AU} = y - h_B^*(x)$$

If the connection between $X$ and $Y$ is non-deterministic, the description of a new prediction problem involves a conditional probability distribution $\pi_{Y|X}$. Standard neural network predictors can only provide a single output $y$. Thus, even given full information of the distribution $\pi$, the uncertainty in the prediction of a single output $y$ remains. This uncertainty cannot be removed by better modeling or more data.

There are multiple work aiming to estimate the aleatoric uncertainty, i.e., to learn $\pi_{Y|X}$. For instance, conditional quantile regression aims to learn each quantile of the distribution $\pi_{Y|X}$ [70, 89, 108]. Conditional density estimation aims to approximately describe the density of $\pi_{Y|X}$ [57, 36, 62, 29, 42, 63]. However, we remark that these approaches also face their own epistemic uncertainty in the estimation. See [26, 108, 10] for recent studies on the epistemic bias in quantile regression.

### A.2    Predictive Uncertainty

Existing work on uncertainty measurement in deep learning models mainly focuses on *prediction sets* (*predictive uncertainty*), which captures the sum of epistemic and aleatoric uncertainties [94, 97, 12, 3, 22].

In certain scenarios, it is not necessary to estimate each uncertainty separately. A distinction between aleatoric and epistemic uncertainties might appear less significant. The user may only concern about the overall uncertainty related to the prediction, which is called predictive uncertainty and can be thought as the summation of the epistemic and aleatoric uncertainties.

The most common way to quantify predictive uncertainty is the *prediction set*: We aim to find a map $\hat{C} : \mathcal{X} \to 2^{\mathcal{Y}}$ which maps an input to a subset of the output space so that for each test point

$x_0 \in \mathcal{X}$, the prediction set $\hat{C}(x_0)$ is likely to cover the true outcome $y_0$ [111, 11, 12, 84, 83, 82]. This prediction set is also called a prediction interval in the case of regression [68, 69]. The prediction set communicates predictive uncertainty via a statistical guarantee on the marginal coverage, i.e.,

$$\mathbb{P}(y_0 \in \hat{C}(x_0)) \geq 1 - \delta$$

for a small threshold $\delta > 0$ where the probability is taken with respect to both training data $\mathcal{D}_{tr}$ (epistemic uncertainty) for learning $\hat{C}$ and the test data $(x_0, y_0)$ (aleatoric uncertainty). It is more tempting to obtain a statistical guarantee with only aleatoric uncertainty by considering the probablity conditional on the prediction set, i.e.,

$$\mathbb{P}(y_0 \in \hat{C}(x_0)|\hat{C}) \geq 1 - \delta. \tag{9}$$

However, this guarantee is in general very hard to achieve in the finite-sample case. Even asymptotically, (9) is not easy to achieve unless we have a very simple structure on the data distribution [98, 116]. A recent study show that (9) could hold in the finite-sample sense if we could leverage a set of validation data [22].

## Appendix B  Statistical Inference for Kernel-Based Regression

### B.1  Classical Statistical Learning Framework and Kernel Ridge Regression

Following Section 2, assuming that the input-output pair $(X, Y)$ is a random vector following an unknown probability distribution $\pi$ on $\mathcal{X} \times \mathcal{Y}$ where $X \in \mathcal{X} \subset \mathbb{R}^d$ is an input and $Y \in \mathcal{Y} \subset \mathbb{R}$ is the corresponding output. Let the marginal distribution of $X$ be $\pi_X(x)$ and the conditional distribution of $Y$ given $X$ be $\pi_{Y|X}(y|x)$.

Suppose a learner has access to a set of training data $\mathcal{D} = \{(x_1, y_1), (x_2, y_2), ..., (x_n, y_n)\}$, which is assumed to be independent and identically distributed (i.i.d.) according to the data distribution $\pi$. Let $\boldsymbol{x} = (x_1, ..., x_n)^T$ and $\boldsymbol{y} = (y_1, ..., y_n)^T$ for short. Let $\pi_{\mathcal{D}}$ or $\pi_n$ denote the empirical distribution associated with the training data $\mathcal{D}$, where $\pi_n$ is used to emphasize the sample size dependence in asymptotic results:

$$\pi_{\mathcal{D}} = \pi_n = \frac{1}{n} \sum_{i=1}^{n} \delta_{(x_i, y_i)}.$$

The goal of the learner is to obtain a function $h : \mathcal{X} \to \mathcal{Y}$ such that $h(x)$ is a "good" predictor for the response $y$ if $X = x$ is observed for any $x \in \mathcal{X}$. This is typically done by minimizing the following (ground-truth) *population risk*

$$R_\pi(h) := \mathbb{E}_{(X,Y) \sim \pi}[\mathcal{L}(h(X), Y)]$$

where $\mathcal{L} : \mathcal{Y} \times \mathbb{R} \to \mathbb{R}$ is the *loss function*. For instance, $\mathcal{L}$ can be the square error in regression or cross-entropy loss in classification. If $h$ is allowed to be any possible functions, the best predictions in the sense of minimizing the risk are described by the *Bayes predictor* $h_B^*$ [61, 93]:

$$h_B^*(X) := \arg \min_{\hat{y} \in \mathcal{Y}} \mathbb{E}_{Y \sim \pi_{Y|X}}[\mathcal{L}(\hat{y}, Y)|X].$$

Note that $h_B^*$ cannot be obtained in practice, since the conditional distribution $\pi_{Y|X}(y|x)$ is unknown. In particular, the least squares loss $\mathcal{L}(h(X), Y) = (h(X) - Y)^2$ yields the *(ground-truth) regression function* defined by

$$g_\pi^*(x) := \mathbb{E}_{(X,Y) \sim \pi}[Y|X = x].$$

As the ground-truth distribution $\pi$ is unknown, we can neither compute nor minimize the population risk $R_\pi(h)$ directly. We define the risk under a general distribution $P$,

$$R_P(h) := \mathbb{E}_{(X,Y) \sim P}[\mathcal{L}(h(X), Y)].$$

In particular, for $P = \pi_n$ (the empirical distribution associated with the training data $\mathcal{D}$), an *empirical risk* is derived:

$$R_{\pi_n}(h) := \mathbb{E}_{(X,Y) \sim \pi_n}[\mathcal{L}(h(X), Y)] = \frac{1}{n} \sum_{i=1}^{n} \mathcal{L}(h(x_i), y_i)$$

In practice, minimizing the risk over $h$ is restricted to a certain function class. Let $\mathcal{H}$ be a hypothesis class, which is a set of functions $\{h : \mathcal{X} \to \mathcal{Y} | h \in \mathcal{H}\}$. For instance, 1) $\mathcal{H}$ could be a nonparametric class such as a *reproducing kernel Hilbert space (RKHS)*. The resulting method is known as *kernel-based regression*; See below. 2) $\mathcal{H}$ could be a parametric class $\{h_\theta : \mathcal{X} \to \mathcal{Y} | \theta \in \Theta\}$ where $\theta$ is the parameter, and $\Theta$ is the set of all possible parameters, e.g., the linear coefficients in a linear regression model, or the set of all network parameters in a neural network model.

In classical statistical learning, one is interested in finding a hypothesis $g_\pi \in \mathcal{H}$ that minimizes the population risk

$$g_\pi := \arg\min_{h \in \mathcal{H}} R_\pi(h). \tag{10}$$

which is called the true risk minimizer. We remark that $h_\pi$ is the best choice in the sense of minimizing the risk within the hypothesis set $\mathcal{H}$ and different choices of $\mathcal{H}$ may lead to different $h_\pi$. As $\pi$ is unknown, the learner may consider minimizing the empirical risk:

$$g_{\pi_n} := \arg\min_{h \in \mathcal{H}} R_{\pi_n}(h) \tag{11}$$

which is called the empirical risk minimizer. More generally, to avoid overfitting in the finite sample regime, the learner may consider a regularized empirical risk minimization problem:

$$g_{\pi_n, \lambda_n} := \arg\min_{h \in \mathcal{H}} R_{\pi_n}(h) + \lambda_n \|h\|_{\mathcal{H}}^2 \tag{12}$$

and its corresponding population problem

$$g_{\pi, \lambda_0} := \arg\min_{h \in \mathcal{H}} R_\pi(h) + \lambda_0 \|h\|_{\mathcal{H}}^2 \tag{13}$$

which are called the true regularized risk minimizer and the empirical regularized risk minimizer, respectively. Here, $\lambda_n \geq 0$ is the regularization parameter, which may depend on the data size $n$, and we assume it has a limit $\lambda_0 = \lim_{n \to \infty} \lambda_n$. In general, the target $g_\pi$ is not equal to $g_{\pi_n, \lambda_n}$ or $g_{\pi_n}$. We omit $0$ in the subscript if $\lambda = 0$ and write $g_{\pi_n} := g_{\pi_n, 0}$ and $g_\pi = g_{\pi, 0}$ for simplicity.

The above framework fits classical machine learning approaches such as linear regression or kernel-based convex regression (such as kernel ridge regression), since (12) as well as (13) has a unique solution in this setting. However, this framework is not suitable for deep learning: the empirical (regularized) risk minimizer $g_{\pi_n, \lambda_n}$ is not unique and cannot be precisely obtained due to the non-convex nature of neural networks. Therefore, the gradient-based methods in deep learning can only find an approximate solution $g_{\pi_n, \lambda_n}$, subjected to procedural variability.

**Kernel ridge regression.** Next, we apply the above framework to kernel ridge regression and review some basic existing results about kernel ridge regression, which can be found, e.g., in [14, 78, 104–106, 27, 109].

The kernel-based regression means that the hypothesis class $\mathcal{H}$ in statistical learning is chosen to be an RKHS. Formally, a Hilbert space $\mathcal{H}$ consisting of functions $h : \mathcal{X} \to \mathbb{R}$ with an inner product $\langle \cdot, \cdot \rangle : \mathcal{H} \times \mathcal{H} \to \mathbb{R}$ is a RKHS if there exists a symmetric positive definite function $k : \mathcal{X} \times \mathcal{X} \to \mathbb{R}$, called a (reproducing) kernel, such that for all $x \in \mathcal{X}$, we have $k(\cdot, x) \in \mathcal{H}$ and for all $x \in \mathcal{X}$ and $h \in \mathcal{H}$, we have $h(x) = \langle h(\cdot), k(\cdot, x) \rangle$. We use the above $k$ to denote the kernel associated with $\mathcal{H}$. Note that for any symmetric positive definite function $k$, we can naturally construct, from $k$, an RKHS $\mathcal{H}$ whose kernel is exactly $k$ [14].

When $\mathcal{H}$ is a RKHS, $g_{\pi_n, \lambda_n}$ in (12) can be computed for a number of convex loss functions $\mathcal{L}$. In particular, for the least squares loss $\mathcal{L}(h(X), Y) = (h(X) - Y)^2$, the resulting problem (12) is well known as the kernel ridge regression, and there exists a closed-form expression for both $g_{\pi_n, \lambda_n}$ and $g_{\pi, \lambda_0}$, as we discuss below.

Formally, the kernel ridge regression problem is given by

$$g_{\pi_n, \lambda_n}(x) := \arg\min_{g \in \mathcal{H}} \left\{ \frac{1}{n} \sum_{j=1}^n (y_j - g(x_j))^2 + \lambda_n \|g\|_{\mathcal{H}}^2 \right\}$$

where $\lambda_n > 0$ is a regularization hyperparameter that may depend on the cardinality of the training data set. There is a closed-form formula for its solution, as stated below.

**Proposition B.1.** *Let*
$$\boldsymbol{k}(\boldsymbol{x}, \boldsymbol{x}) = (k(x_i, x_j))_{n \times n} \in \mathbb{R}^{n \times n}$$
*be the kernel Gram matrix and*
$$\boldsymbol{k}(x, \boldsymbol{x}) = (k(x, x_1), \cdots, k(x, x_n))^T.$$
*Then the (unique) kernel ridge regression solution is given as*
$$g_{\pi_n, \lambda_n}(x) = \boldsymbol{k}(x, \boldsymbol{x})^T (\boldsymbol{k}(\boldsymbol{x}, \boldsymbol{x}) + \lambda_n n \boldsymbol{I})^{-1} \boldsymbol{y}$$

It is worth mentioning the linearity of kernel ridge regression: When we have two outputs $\boldsymbol{y} = (y_1, ..., y_n)^T$ and $\boldsymbol{y}' = (y_1', ..., y_n')^T$, we have
$$\boldsymbol{k}(x, \boldsymbol{x})^T (\boldsymbol{k}(\boldsymbol{x}, \boldsymbol{x}) + \lambda_n n \boldsymbol{I})^{-1} (\boldsymbol{y} + \boldsymbol{y}')$$
$$= \boldsymbol{k}(x, \boldsymbol{x})^T (\boldsymbol{k}(\boldsymbol{x}, \boldsymbol{x}) + \lambda_n n \boldsymbol{I})^{-1} \boldsymbol{y} + \boldsymbol{k}(x, \boldsymbol{x})^T (\boldsymbol{k}(\boldsymbol{x}, \boldsymbol{x}) + \lambda_n n \boldsymbol{I})^{-1} \boldsymbol{y}$$

which means the solution to
$$\arg\min_{g \in \mathcal{H}} \left\{ \frac{1}{n} \sum_{j=1}^n (y_j + y_j' - g(x_j))^2 + \lambda_n \|g\|_{\mathcal{H}}^2 \right\}$$

is the summation of the solution to
$$\arg\min_{g \in \mathcal{H}} \left\{ \frac{1}{n} \sum_{j=1}^n (y_j - g(x_j))^2 + \lambda_n \|g\|_{\mathcal{H}}^2 \right\}$$

and the solution to
$$\arg\min_{g \in \mathcal{H}} \left\{ \frac{1}{n} \sum_{j=1}^n (y_j' - g(x_j))^2 + \lambda_n \|g\|_{\mathcal{H}}^2 \right\}.$$

Define $\mathcal{O}_k : L^2(\pi_X) \to L^2(\pi_X)$ as the kernel integral operator
$$(\mathcal{O}_k g)(x) := \int_{\mathcal{X}} k(x, x') g(x') \pi_X(x') dx', \ x \in \mathcal{X}, \ g \in L^2(\pi_X).$$
where $L^2(\pi_X) := \{f : \int_{\mathcal{X}} f(x)^2 \pi_X(x) dx < \infty\}$.

[109] shows that $\mathcal{O}_k$ is a compact and positive self-adjoint linear operator on $L^2(\pi_X)$. Note that since $L_k$ is positive: $\mathcal{O}_k \geq 0$, we have that $\mathcal{O}_k + \lambda_0 I$ is strictly positive for any $\lambda_0 > 0$: $\mathcal{O}_k + \lambda_0 I > 0$, and thus its inverse operator $(\mathcal{O}_k + \lambda_0 I)^{-1}$ exists and is unique. Note that $(\mathcal{O}_k + \lambda_0 I)^{-1}$ is also a linear operator on $L^2(\pi_X)$.

Next, consider the population risk minimization problem corresponding to $g_{\pi_n, \lambda_n}$ as follows:
$$g_{\pi, \lambda_0} := \arg\min_{g \in \mathcal{H}} \left\{ \mathbb{E}_\pi[(Y - g(X))^2] + \lambda_0 \|g\|_{\mathcal{H}}^2 \right\}.$$

It is easy to see that this problem is equivalent to
$$g_{\pi, \lambda_0} := \arg\min_{g \in \mathcal{H}} \left\{ \mathbb{E}_\pi[(g_\pi^*(X) - g(X))^2] + \lambda_0 \|g\|_{\mathcal{H}}^2 \right\}.$$

where $g_\pi^*(x) = \mathbb{E}_{(X,Y) \sim \pi}[Y|X = x]$ is the (ground-truth) regression function. This problem has the following explicit closed-form expression of the solution (a proof can be found in [27]):

**Proposition B.2.** *The solution of $g_{\pi, \lambda_0}$ is given as $g_{\pi, \lambda_0} = (\mathcal{O}_k + \lambda_0 I)^{-1} \mathcal{O}_k g_\pi^*$.*

The linearity of kernel ridge regression also holds for the population-level solution:
$$(g + g')_{\pi, \lambda_0} = (\mathcal{O}_k + \lambda_0 I)^{-1} \mathcal{O}_k (g + g')_\pi^*$$
$$= (\mathcal{O}_k + \lambda_0 I)^{-1} \mathcal{O}_k (g_\pi^* + g_\pi'^*)$$
$$= (\mathcal{O}_k + \lambda_0 I)^{-1} \mathcal{O}_k g_\pi^* + (\mathcal{O}_k + \lambda_0 I)^{-1} \mathcal{O}_k g_\pi'^*$$
$$= g_{\pi, \lambda_0} + g_{\pi, \lambda_0}'$$

for any two functions $g, g' \in L^2(\pi_X)$ since both $(\mathcal{O}_k + \lambda_0 I)^{-1}$ and $\mathcal{O}_k$ are linear operators on $L^2(\pi_X)$.

## B.2 Asymptotic Normality of Kernel-Based Regression

In this section, we review existing results on the asymptotic normality of kernel-based regression, which is established using the influence function concept [25, 24, 50].

Let $\mathcal{H}$ be a generic RKHS with the associated kernel $k(x, x')$. Let $\|\cdot\|_{\mathcal{H}}$ denote the norm on $\mathcal{H}$. Note that the feature map $k_x = k(x, \cdot)$ is a function in $\mathcal{H}$. We consider the connection between the true regularized risk minimizer $g_{\pi, \lambda_0}$ and the empirical regularized risk minimizer $g_{\pi_n, \lambda_n}$ introduced in (11) and (12) when $\mathcal{H}$ is a RKHS.

The asymptotic normality of kernel-based regression roughly states that under some mild conditions, $\sqrt{n}(g_{\pi_n, \lambda_n} - g_{\pi, \lambda_0})$ is asymptotically normal as $n \to \infty$. This result provides the theoretical foundations to conduct asymptotic statistical inference for kernel-based regression, in particular, building asymptotic confidence interval for the ground-truth $g_{\pi, \lambda_0}$.

To be rigorous, we first list the following assumptions introduced by [50]:

**Assumption B.3.** Let $(\Omega, \mathcal{A}, \mathcal{Q})$ be a probability space. Suppose that $\mathcal{X} \subset \mathbb{R}^d$ is closed and bounded with Borel-$\sigma$-algebra $\mathcal{B}(\mathcal{X})$, and $\mathcal{Y} \subset \mathbb{R}$ is closed with Borel-$\sigma$-algebra $\mathcal{B}(\mathcal{Y})$. The Borel-$\sigma$-algebra of $\mathcal{X} \times \mathcal{Y}$ is denoted by $\mathcal{B}(\mathcal{X} \times \mathcal{Y})$. Let $\mathcal{H}$ be an RKHS with kernel $k$ and let $\pi$ be a probability measure on $(\mathcal{X} \times \mathcal{Y}, \mathcal{B}(\mathcal{X} \times \mathcal{Y}))$. Assume that the kernel of $\mathcal{H}$, $k : \mathcal{X} \times \mathcal{X} \to \mathbb{R}$ is the restriction of an $m$-times continuously differentiable kernel $\tilde{k} : \mathbb{R}^d \times \mathbb{R}^d \to \mathbb{R}$ such that $m > d/2$ and $k \neq 0$. Let $\lambda_0 \in (0, +\infty)$ be any positive number. Suppose that the sequence $\lambda_n$ satisfy that $\lambda_n - \lambda_0 = o(\frac{1}{\sqrt{n}})$.

**Assumption B.4.** Let $\mathcal{L} : \mathcal{Y} \times \mathbb{R} \to [0, +\infty)$ be a loss function satisfying the following conditions:

- $\mathcal{L}$ is a convex loss, i.e., $z \mapsto L(y, z)$ is convex in $z$ for every $y \in \mathcal{Y}$.

- The partial derivatives

$$\mathcal{L}'(y, z) = \frac{\partial}{\partial z}\mathcal{L}(y, z), \quad \mathcal{L}''(y, z) = \frac{\partial^2}{\partial^2 z}\mathcal{L}(y, z)$$

  exist for every $(y, z) \in \mathcal{Y} \times \mathbb{R}$.

- The maps

$$(y, z) \mapsto \mathcal{L}'(y, z), \quad (y, z) \mapsto \mathcal{L}''(y, z)$$

  are continuous.

- There is a $b \in L^2(\pi_Y)$, and for every $a \in (0, +\infty)$, there is a $b'_a \in L^2(\pi_Y)$ and a constant $b''_a \in (0, +\infty)$ such that, for every $y \in \mathcal{Y}$,

$$|\mathcal{L}(y, z)| \leq b(y) + |z|^p \ \forall z, \quad \sup_{z \in [-a, a]} |\mathcal{L}'(y, z)| \leq b'_a(y), \quad \sup_{z \in [-a, a]} |\mathcal{L}''(y, z)| \leq b''_a$$

  where $p \geq 1$ is a constant.

Note that the conditions on the loss function $\mathcal{L}$ in Assumption B.4 are satisfied, e.g., for the logistic loss for classification or least-square loss for regression with $\pi$ such that $\mathbb{E}_{\pi_Y}[Y^4] < \infty$. Therefore, Assumption B.4 is reasonable for the kernel ridge regression problem we consider in this work.

**Theorem B.5** (Theorem 3.1 in [50]). *Suppose that Assumptions B.3 and B.4 hold. Then there is a tight, Borel-measurable Gaussian process $\mathbb{G} : \Omega \to \mathcal{H}, \omega \mapsto \mathbb{G}(\omega)$ such that*

$$\sqrt{n}(g_{\pi_n, \lambda_n} - g_{\pi, \lambda_0}) \Rightarrow \mathbb{G} \text{ in } \mathcal{H}$$

*where $\Rightarrow$ represents "converges weakly". The Gaussian process $\mathbb{G}$ is zero-mean; i.e., $\mathbb{E}[\langle g, \mathbb{G}\rangle_{\mathcal{H}}] = 0$ for every $g \in \mathcal{H}$.*

Note that Theorem B.5 implies that for any $g \in \mathcal{H}$, the random variable

$$\Omega \to \mathbb{R}, \quad \omega \mapsto \langle g, \mathbb{G}(\omega)\rangle_{\mathcal{H}}$$

has a zero-mean normal distribution. In particular, letting $g = k_x$, the reproducing property of $k$ implies that,

$$\sqrt{n}(g_{\pi_n, \lambda_n}(x) - g_{\pi, \lambda_0}(x)) \Rightarrow \mathbb{G}(x)$$

To obtain the variance of this limiting distribution, denoted as $\xi^2(x)$, we review the tool of influence functions as follows.

Let $P$ be a general distribution on the domain $\mathcal{X} \times \mathcal{Y}$. Let $z = (z_x, z_y) \in \mathcal{X} \times \mathcal{Y}$ be a general point. Let $T$ be a general statistical functional that maps a distribution $P$ to a real value, that is, $T : P \to T(P) \in \mathbb{R}$. The Gâteaux derivative of $T(P)$, $T'_P(\cdot)$, is defined as

$$T'_P(Q) := \lim_{\varepsilon \to 0} \frac{T(P + \varepsilon Q) - T(P)}{\varepsilon}$$

Let

$$P_{\varepsilon,z} := (1 - \varepsilon)P + \varepsilon \delta_z = P + \varepsilon(\delta_z - P)$$

where $\delta_z$ denotes the point mass distribution in $z$. Then the influence function of $T(P)$ at the point $z$ is a special Gâteaux derivative defined as

$$IF(z; T, P) := \lim_{\varepsilon \to 0} \frac{T(P_{\varepsilon,z}) - T(P)}{\varepsilon} = T'_P(\delta_z - P).$$

Under some mild conditions (e.g., $T$ is Hadamard differentiable [38, 110]), the central limit theorem for the statistical functional $T$ holds:

$$\sqrt{n}(T(\pi_n) - T(\pi)) \Rightarrow \mathcal{N}\left(0, \int_z IF^2(z; T, \pi)d\pi(z)\right)$$

as $n \to \infty$ where $\pi_n$ is the empirical distribution associated with $n$ samples i.i.d. drawn from $\pi$.

Applying the above concepts to the kernel-based regression, we let $T(P)$ be the solution to the following problem:

$$\min_{g \in \mathcal{H}} \mathbb{E}_P[(Y - g(X))^2] + \lambda_0 \|g\|_{\mathcal{H}}^2.$$

Note that [50] has shown that $\sqrt{n}(g_{\pi_n, \lambda_n}(x) - g_{\pi, \lambda_0}(x))$ has the same limiting behavior as $\sqrt{n}(g_{\pi_n, \lambda_0}(x) - g_{\pi, \lambda_0}(x))$ so only the limiting regularization hyper-parameter $\lambda_0$ is used in $T(P)$. The proof of Theorem 3.1 in [50] provides the closed-formed expression of the Gâteaux derivative $T'_P(Q)$ as follows:

$$T'_P(Q) = -S_P^{-1}(\mathbb{E}_Q[\mathcal{L}'(Y, g_{P,\lambda_0}(X))\Phi(X)])$$

for $Q \in \operatorname{lin}(B_S)$ where $\operatorname{lin}(B_S)$ corresponds to a subset of finite measures on $(\mathcal{X} \times \mathcal{Y}, \mathcal{B}(\mathcal{X} \times \mathcal{Y}))$ (See Proposition A.8. in [50]) and $S_P : \mathcal{H} \to \mathcal{H}$ is defined by

$$S_P(f) = 2\lambda_0 f + \mathbb{E}_P[\mathcal{L}''(Y, g_{P,\lambda_0}(X))f(X)\Phi(X)]$$

where $\Phi$ is the feature map of $\mathcal{H}$ (e.g., we can simply take $\Phi(x)(\cdot) = k(x, \cdot)$). Note that $S_P$ is a continuous linear operator that is invertible (See Proposition A.5. in [50]). In particular, letting $Q = \delta_z - P$, we obtain that

$$T'_P(\delta_z - P) = S_P^{-1}(\mathbb{E}_P[\mathcal{L}'(Y, g_{P,\lambda_0}(X))\Phi(X)]) - \mathcal{L}'(z_y, g_{P,\lambda_0}(z_x))S_P^{-1}\Phi(z_x). \quad (14)$$

Note that the above special Gâteaux derivative (influence function) of $T(P)$, $T'_P(\delta_z - P)$, had been derived initially in [25].

The proof of Theorem 3.1 in [50] shows that $T(P)$ is Hadamard differentiable, and the asymptotic normality holds. Using the above expression of the influence function, we conclude that:

$$\sqrt{n}(g_{\pi_n, \lambda_n}(x) - g_{\pi, \lambda_0}(x)) \Rightarrow \mathcal{N}(0, \xi^2(x))$$

where

$$\xi^2(x) = \int_{z \in \mathcal{X} \times \mathcal{Y}} IF^2(z; T, \pi)(x)d\pi(z)$$

and $IF(z; T, \pi)$ is given by (14):

$$IF(z; T, \pi) = T'_\pi(\delta_z - \pi) = S_\pi^{-1}(\mathbb{E}_\pi[\mathcal{L}'(Y, g_{\pi, \lambda_0}(X))\Phi(X)]) - \mathcal{L}'(z_y, g_{\pi, \lambda_0}(z_x))S_\pi^{-1}\Phi(z_x).$$

Summarizing the above discussion, Proposition 3.4 follows.

## B.3 Infinitesimal Jackknife for Kernel-Based Regression

In this section, we follow up on our discussion in Section 3.3 on the infinitesimal jackknife variance estimation. We derive the closed-formed expression of the infinitesimal jackknife variance estimation for $\xi^2(x)$ in Section B.2 in the kernel ridge regression. We also show the consistency of infinitesimal jackknife variance estimation in general kernel-based regression. Our consistency result appears new in the literature.

First, we estimate $\xi^2(x_0)$ as follows:

$$\hat{\xi}^2(x_0) = \frac{1}{n} \sum_{z_i \in \mathcal{D}} IF^2(z_i; T, \pi_n)(x_0)$$

where $\mathcal{D} = \{(x_1, y_1), (x_2, y_2), ..., (x_n, y_n)\}$ and $\pi_n$ is the empirical distribution associated with the data $\mathcal{D}$. This estimator is known as the infinitesimal jackknife variance estimator [65]. In this section, we use $x_0$ instead of $x$ in the variance or influence function at a test point $x_0$ to avoid confusion between $x_0$ and $z_x$. The latter $z_x$ is referred to the $x$-componenet of $z$.

Based on [25, 31, 50], we derive the closed-form expression of the variance estimation $\hat{\xi}^2(x_0)$ as follows.

**Theorem B.6** (Expression of infinitesimal jackknife for kernel ridge regression)**.** *Suppose that Assumptions B.3 and B.4 hold. For the least squares loss $\mathcal{L}(h(X), Y) = (h(X) - Y)^2$ in kernel ridge regression, $IF(z; T, \pi_n)$ is given by*

$$IF(z; T, \pi_n)(x_0) = \boldsymbol{k}(x_0, \boldsymbol{x})^T (\boldsymbol{k}(\boldsymbol{x}, \boldsymbol{x}) + \lambda_0 n \boldsymbol{I})^{-1} M_z(\boldsymbol{x}) - M_z(x_0) \tag{15}$$

*where $M_z(\boldsymbol{x}) := (M_z(x_1), ..., M_z(x_n))^T$ and*

$$M_z(x_i) := g_{\pi_n, \lambda_0}(x_i) - \frac{1}{\lambda_0}(z_y - g_{\pi_n, \lambda_0}(z_x))k(z_x, x_i)$$

*for $x_i = x_0, x_1, \cdots, x_n$.*

Note that although Hadamard differentiability is able to guarantee the central limit theorem of $T$, the consistency of variance estimation generally requires more than Hadamard differentiability. In general, we need some additional continuity condition (such as continuously Gâteaux differentiability) to guarantee that $IF^2(z; T, \pi_n)$ is indeed "close" to $IF^2(z; T, \pi)$ and $\frac{1}{n} \sum_{z_i \in \mathcal{D}} IF^2(z_i; T, \pi_n)(x_0)$ is indeed "close" to $\int_{z \in \mathcal{X} \times \mathcal{Y}} IF^2(z; T, \pi)(x_0)d\pi(z)$. We show that this is achievable for the infinitesimal jackknife variance estimator for kernel ridge regression by only imposing a weak assumption.

**Theorem B.7** (Consistency of infinitesimal jackknife for kernel-based regression)**.** *Suppose that Assumptions B.3 and B.4 hold. Moreover, assume that $\mathcal{Y} \subset \mathbb{R}$ is bounded, and $b$ and $b'_a$ in Assumption B.4 are bounded on $\mathcal{Y}$. Then we have*

$$\hat{\xi}^2(x_0) = \frac{1}{n} \sum_{z_i \in \mathcal{D}} IF^2(z_i; T, \pi_n)(x_0) \to \int_{z \in \mathcal{X} \times \mathcal{Y}} IF^2(z; T, \pi)(x)d\pi(z) = \xi^2(x_0), \quad a.s.$$

*as $n \to \infty$. Hence, an asymptotically exact $(1 - \alpha)$-level confidence interval of $g_{\pi, \lambda_0}(x_0)$ is*

$$\left[ g_{\pi_n, \lambda_n}(x_0) - \frac{\hat{\xi}(x_0)}{n}q_{1-\frac{\alpha}{2}}, g_{\pi_n, \lambda_n}(x_0) + \frac{\hat{\xi}(x_0)}{n}q_{1-\frac{\alpha}{2}} \right]$$

*where $q_\alpha$ is the $\alpha$-quantile of the standard Gaussian distribution $\mathcal{N}(0, 1)$.*

The proof of Theorems B.6 and B.7 is given in Appendix E.

## B.4 PNC-Enhanced Infinitesimal Jackknife

In this section, we return to our problem setting in Section 3.3. We apply the general results in Section B.3 to our PNC predictor and develop the PNC-enhanced infinitesimal jackknife confidence interval for over-parameterized neural networks.

Recall that the statistical functional $T_1$ is associated with the following problem:

$$\hat{h}_{n,\theta^b}(\cdot) - \hat{\phi}_{n,\theta^b}(\cdot) - \bar{s}(\cdot) = \min_{g \in \mathcal{H}} \frac{1}{n} \sum_{i=1}^{n} [(y_i - \bar{s}(x_i) - g(x_i))^2] + \lambda_n \|g\|_{\mathcal{H}}^2$$

Consider the PNC-enhanced infinitesimal jackknife variance estimator:

$$\hat{\sigma}^2(x_0) = \frac{1}{n} \sum_{z_i \in \mathcal{D}} IF^2(z_i; T_1, \pi_n)(x_0).$$

We obtain that

**Theorem B.8** (Expression of PNC-enhanced infinitesimal jackknife). *Suppose that Assumption C.2 holds. Suppose that Assumptions B.3 and B.4 hold when $Y$ is replaced by $Y - \bar{s}(X)$. Then $IF(z; T_1, \pi_n)$ is given by*

$$IF(z; T_1, \pi_n)(x_0) = \boldsymbol{K}(x_0, \boldsymbol{x})^T (\boldsymbol{K}(\boldsymbol{x}, \boldsymbol{x}) + \lambda_0 n \boldsymbol{I})^{-1} M_z(\boldsymbol{x}) - M_z(x_0) \qquad (16)$$

*where $M_z(\boldsymbol{x}) := (M_z(x_1), ..., M_z(x_n))^T$ and*

$$M_z(x_i) := \hat{h}_{n,\theta^b}(x_i) - \hat{\phi}_{n,\theta^b}(x_i) - \bar{s}(x_i) - \frac{1}{\lambda_0}(z_y - (\hat{h}_{n,\theta^b}(z_x) - \hat{\phi}_{n,\theta^b}(z_x)))K(z_x, x_i)$$

*for $x_i = x_0, x_1, \cdots, x_n$. Hence*

$$\hat{\sigma}^2(x_0) = \frac{1}{n} \sum_{z_i \in \mathcal{D}} IF^2(z_i; T_1, \pi_n)(x_0)$$

$$= \frac{1}{n} \sum_{z_i \in \mathcal{D}} (\boldsymbol{K}(x_0, \boldsymbol{x})^T (\boldsymbol{K}(\boldsymbol{x}, \boldsymbol{x}) + \lambda_0 n \boldsymbol{I})^{-1} M_{z_i}(\boldsymbol{x}) - M_{z_i}(x_0))^2$$

Theorem B.8 immediately follows from Theorem B.6.

**Theorem B.9** (Exact coverage of PNC-enhanced infinitesimal jackknife confidence interval). *Suppose that Assumption C.2 holds. Suppose that Assumptions B.3 and B.4 hold when $Y$ is replaced by $Y - \bar{s}(X)$. Moreover, assume that $\mathcal{Y} \subset \mathbb{R}$ is bounded. Then we have*

$$\hat{\sigma}^2(x_0) = \frac{1}{n} \sum_{z_i \in \mathcal{D}} IF^2(z_i; T_1, \pi_n)(x_0) \to \int_{z \in \mathcal{X} \times \mathcal{Y}} IF^2(z; T_1, \pi)(x) d\pi(z) = \sigma(x_0), \quad a.s.$$

*as $n \to \infty$. Hence, an asymptotically exact $(1 - \alpha)$-level confidence interval of $h^*(x_0)$ is*

$$\left[ \hat{h}_{n,\theta^b}(x_0) - \hat{\phi}_{n,\theta^b}(x_0) - \frac{\hat{\sigma}(x_0)}{n} q_{1-\frac{\alpha}{2}}, \hat{h}_{n,\theta^b}(x_0) - \hat{\phi}_{n,\theta^b}(x_0) + \frac{\hat{\sigma}(x_0)}{n} q_{1-\frac{\alpha}{2}} \right]$$

*where $q_\alpha$ is the $\alpha$-quantile of the standard Gaussian distribution $\mathcal{N}(0, 1)$ and the computation of $\hat{\sigma}(x_0)$ is given by Theorem B.8.*

Note that in our problem setting, we can set $b(y) = y^2$ and $b'_a = a + |y|$ for kernel ridge regression. Since $\mathcal{Y}$ is bounded, $b$ and $b'_a$ are naturally bounded on $\mathcal{Y}$. Therefore, Theorem B.9 follows from Theorem B.7.

## Appendix C   Theory of Neural Tangent Kernel

In this section, we provide a brief review of the theory of neural tangent kernel (NTK). Then we discuss particularly one result employed by this paper, i.e.,

*The shifted kernel ridge regressor using NTK with a shift from an initial function $s_{\theta^b}$ is exactly the linearized neural network regressor that starts from the initial network $s_{\theta^b}$.*

NTK [64] has attracted a lot of attention since it provides a new perspective on training dynamics, generalization, and expressibility of over-parameterized neural networks. Recent papers show that when training an over-parametrized neural network, the weight matrix at each layer is close to its

initialization [85, 35]. An over-parameterized neural network can rapidly reduce training error to zero via gradient descent, and thus finds a global minimum despite the objective function being non-convex [34, 4–6, 122]. Moreover, the trained network also exhibits good generalization property [19, 20]. These observations are implicitly described by a notion, NTK, suggested by [64]. This kernel is able to characterize the training behavior of sufficiently wide fully-connected neural networks and build a new connection between neural networks and kernel methods. Another line of work is to extend NTK for fully-connected neural networks to CNTK for convolutional neural networks [7, 114, 86]. NTK is a fruitful and rapidly growing area.

We focus on the result that is used as the foundation of our epistemic uncertainty quantification: Since the gradient of over-parameterized neural networks is nearly constant and close to its initialization, training networks is very similar to a shifted kernel ridge regression where the feature map of the kernel is given by the gradient of networks [80]. Multiple previous works [7, 80, 54, 59, 118, 15] have established some kind of equivalences between the neural network regressor and the kernel ridge regressor based on a wide variety of assumptions or settings, such as the full-rank assumption of NTK, or zero-valued initialization, or introducing a small multiplier $\kappa$, or normalized training inputs $\|x_i\|_2 = 1$. These results are also of different forms. Since a uniform statement is not available, we do not intend to dig into those detailed assumptions or theorems as they are not the focus of this paper (Interested readers may refer to the above papers).

In the following, we provide some discussions on how to obtain the equivalence in Proposition 3.1 adopted in the main paper, based on a less (technically) rigorous assumption. This assumption is borrowed from the *linearized neural network* property introduced in [80], where they replace the outputs of the neural network with their first-order Taylor expansion, called the linearized neural network. They show that in the infinite width limit, the outputs of the neural network are the same as the linearized neural network. [80] does not add regularization in the loss. Instead, we introduce a regularization $\lambda_n > 0$ in the loss since regularization is common and useful in practice. Moreover, it can guarantee the stable computation of the inversion of the NTK Gram matrix and can be naturally linked to kernel ridge regression. In the following, we will review the linearized neural network assumption (Assumption C.1) as well as other network specifications. Based on them, we show that Proposition C.3 holds, which provides the starting point for our uncertainty quantification framework. We also provide some additional remarks in Section D.

**NTK parameterization.** We consider a fully-connected neural network with any depth defined formally as follows. Suppose the network consists of $L$ hidden layers. Denote $g^{(0)}(x) = x$ and $d_0 = d$. Let

$$f^{(l)}(x) = W^{(l)}g^{(l-1)}(x), \; g^{(l)}(x) = \sqrt{\frac{c_\sigma}{d_l}}\sigma(f^{(l)}(x))$$

where $W^{(l)} \in \mathbb{R}^{d_l \times d_{l-1}}$ is the weight matrix in the $l$-th layer, $\sigma$ is a coordinate-wise activation function, $c_\sigma = \mathbb{E}_{z \sim \mathcal{N}(0,1)}[\sigma^2(z)]^{-1}$, and $l \in [L]$. The output of the neural network is

$$f_\theta(x) := f^{(L+1)}(x) = W^{(L+1)}g^{(L)}(x)$$

where $W^{(L+1)} \in \mathbb{R}^{1 \times d_L}$, and $\theta = (W^{(1)}, ..., W^{(L+1)})$ represents all the parameters in the network. Note that here we use NTK parametrization with a width-dependent scaling factor, which is thus slightly different from the standard parametrization. Unlike the standard parameterization which only normalizes the forward dynamics of the network, the NTK parameterization also normalizes its backward dynamics.

**He Initialization.** The NTK theory depends on the random initialization of the network. Suppose the dimension of network parameters $\theta$ is $p$. We randomly initialize all the weights to be i.i.d. $\mathcal{N}(0, 1)$ random variables. In other words, we set $P_{\theta_b} = \mathcal{N}(0, \boldsymbol{I}_p)$ and let $\theta^b$ be an instantiation drawn from $P_{\theta_b}$. This initialization method is essentially known as the He initialization [55].

**NTK and Linearized neural network.** With the above NTK parameterization and He initialization, the population NTK expression is defined recursively as follows: [64, 7]: For $l \in [L]$,

$$\Sigma^{(0)}(x, x') = x^T x' \in \mathbb{R},$$

$$\Lambda^{(l)}(x, x') = \begin{pmatrix} \Sigma^{(l-1)}(x, x) & \Sigma^{(l-1)}(x, x') \\ \Sigma^{(l-1)}(x, x') & \Sigma^{(l-1)}(x', x') \end{pmatrix} \in \mathbb{R}^{2 \times 2},$$

$$\Sigma^{(l)}(x, x') = c_\sigma \mathbb{E}_{(u,v) \sim \mathcal{N}(0, \Lambda(l))}[\sigma(u)\sigma(v)] \in \mathbb{R}.$$

We also define a derivative covariance:

$$\Sigma^{(l)\prime}(x, x') = c_\sigma \mathbb{E}_{(u,v)\sim\mathcal{N}(0,\Lambda(l))}[\sigma'(u)\sigma'(v)] \in \mathbb{R},$$

The final population NTK is defined as

$$K(x, x') = \sum_{l=1}^{L+1} \left( \Sigma^{(l-1)}(x, x') \prod_{s=l}^{L+1} \Sigma^{(s)\prime}(x, x') \right)$$

where $\Sigma^{(L+1)\prime}(x, x') = 1$ for convenience. Let $\langle \cdot, \cdot \rangle$ be the standard inner product in $\mathbb{R}^p$.

Let $J(\theta; x) := \nabla_\theta f_\theta(x) \in \mathbb{R}^{1\times p}$ denote the gradient of the network. The empirical (since the initialization is random) NTK matrix is defined as

$$K_\theta(x, x') = \langle J(\theta; x); J(\theta; x') \rangle.$$

In practice, we may use the empirical NTK matrix to numerically compute the population NTK matrix.

One of the most important results in the NTK theory is that the empirical NTK matrix converges to the population NTK matrix as the width of the network increases [64, 114, 115, 7]:

$$K_\theta(x, x') = \langle J(\theta; x), J(\theta; x) \rangle \to K(x, x')$$

where $\to$ represents "converge in probability" [7] or "almost surely" [114].

The NTK theory shows that $J(\theta; x) = \nabla_\theta f_\theta(x)$ is approximately a constant independent of the network parameter $\theta$ (but still depends on $x$) when the network is sufficiently wide. Therefore, when we treat $J(\theta; x)$ as a constant independent of the network parameter $\theta$, we can obtain a model obtained from the first-order Taylor expansion of the network around its initial parameters, which is called a linearized neural network [80]. Our study on epistemic uncertainty is based on the linearized neural network.

To be rigorous, we adopt the following linearized neural network assumption based on the NTK theory:

**Assumption C.1.** [Linearized neural network assumption] Suppose that $J(\theta; x) = \nabla_\theta f_\theta(x) \in \mathbb{R}^{1\times p}$ is independent of $\theta$ during network training, which is thus denoted as $J(x)$. The population NTK is then given by $K(x, x') = \langle J(x), J(x) \rangle = J(x)J(x)^\top$.

We introduce the NTK vector/matrix as follows. For $\boldsymbol{x} = (x_1, ..., x_j)^\top$, we let

$$\boldsymbol{K}(\boldsymbol{x}, \boldsymbol{x}) := (K(x_i, x_j))_{i,j=1,...,n} = J(\boldsymbol{x})J(\boldsymbol{x})^\top \in \mathbb{R}^{n\times n}$$

be the NTK Gram matrix evaluated on data $\boldsymbol{x}$. For an arbitrary point $x \in \mathbb{R}^d$, we let $\boldsymbol{K}(x, \boldsymbol{x}) \in \mathbb{R}^n$ be the kernel value evaluated between the point $x$ and data $\boldsymbol{x}$, i.e.,

$$\boldsymbol{K}(x, \boldsymbol{x}) := (K(x, x_1), K(x, x_2), ..., K(x, x_n))^T = J(x)J(\boldsymbol{x})^\top \in \mathbb{R}^{1\times n}.$$

**Training dynamics of the linearized network.** Based on Assumption C.1, the training dynamics of the linearized network can be derived as follows.

We consider the regression problem with the following regularized empirical loss

$$\hat{R}(f_\theta) = \frac{1}{n} \sum_{i=1}^n \mathcal{L}(f_\theta(x_i), y_i) + \lambda_n \|\theta - \theta^b\|_2^2. \tag{17}$$

where $\theta^b = \theta(0)$ is the initialization of the network. In the following, we use $f_{\theta(t)}$ to represent the network with parameter $\theta(t)$ that evolves with the time $t$. Let $\eta$ be the learning rate. Suppose the network parameter is trained via continuous-time gradient flow. Then the evolution of the parameters $\theta(t)$ and $f_{\theta(t)}$ can be written as

$$\frac{d\theta(t)}{dt} = -\eta \left( \frac{1}{n} \nabla_\theta f_{\theta(t)}(\boldsymbol{x})^\top \nabla_{f_{\theta(t)}(\boldsymbol{x})} \mathcal{L} + 2\lambda_n(\theta(t) - \theta(0)) \right) \tag{18}$$

$$\frac{df_{\theta(t)}(\boldsymbol{x})}{dt} = \nabla_\theta f_{\theta(t)}(\boldsymbol{x}) \frac{d\theta(t)}{dt} \tag{19}$$

where $\theta(t)$ and $f_{\theta(t)}(x)$ are the network parameters and network output at time $t$.

Under Assumption C.1, we have that

$$f_{\theta(t)}(x) = f_{\theta(0)}(x) + J(x)(\theta(t) - \theta(0)),$$

which gives

$$\nabla_\theta f_{\theta(t)}(x) = J(x).$$

Suppose the loss function is an MSE loss, i.e., $\mathcal{L}(\tilde{y}, y) = (\tilde{y} - y)^2$. Then

$$\nabla_{f_{\theta(t)}(\boldsymbol{x})} \mathcal{L}(f_{\theta(t)}(\boldsymbol{x}), \boldsymbol{y}) = 2(f_{\theta(t)}(\boldsymbol{x}) - \boldsymbol{y})$$

Therefore, under Assumption C.1, (18) becomes

$$
\begin{aligned}
\frac{d\theta(t)}{dt} &= -\eta \left( \frac{1}{n} J(\boldsymbol{x})^\top \nabla_{f_{\theta(t)}(\boldsymbol{x})} \mathcal{L} + 2\lambda_n(\theta(t) - \theta(0)) \right) \\
&= -\eta \left( \frac{2}{n} J(\boldsymbol{x})^\top \left( f_{\theta(0)}(\boldsymbol{x}) + J(\boldsymbol{x})(\theta(t) - \theta(0)) - \boldsymbol{y} \right) + 2\lambda_n(\theta(t) - \theta(0)) \right) \\
&= -2\eta \left( \frac{1}{n} J(\boldsymbol{x})^\top \left( f_{\theta(0)}(\boldsymbol{x}) - J(\boldsymbol{x})\theta(0) - \boldsymbol{y} \right) - \lambda_n\theta(0) + \left( \frac{1}{n} J(\boldsymbol{x})^\top J(\boldsymbol{x}) + \lambda_n \boldsymbol{I} \right) \theta(t) \right)
\end{aligned}
$$

Note that $\frac{1}{n} J(\boldsymbol{x})^\top \left( f_{\theta(0)}(\boldsymbol{x}) - J(\boldsymbol{x})\theta(0) - \boldsymbol{y} \right) - \lambda_n\theta(0)$ and $\left( \frac{1}{n} J(\boldsymbol{x})^\top J(\boldsymbol{x}) + \lambda_n \boldsymbol{I} \right)$ are both independent of $t$. Solving this ordinary differential equation, we obtain

$$
\begin{aligned}
\theta(t) = &\theta(0) - \frac{1}{n} J(\boldsymbol{x})^\top \left( \frac{1}{n} J(\boldsymbol{x})^\top J(\boldsymbol{x}) + \lambda_n \boldsymbol{I} \right)^{-1} \\
&\left( \boldsymbol{I} - \exp\left( -2\eta t \left( \frac{1}{n} J(\boldsymbol{x})^\top J(\boldsymbol{x}) + \lambda_n \boldsymbol{I} \right) \right) \right) \left( f_{\theta(0)}(\boldsymbol{x}) - \boldsymbol{y} \right)
\end{aligned}
$$

Hence the network output at time $t$ becomes

$$
\begin{aligned}
&f_{\theta(t)}(x) - f_{\theta(0)}(x) \\
=&J(x)(\theta(t) - \theta(0)) \\
=& -\frac{1}{n} J(x)J(\boldsymbol{x})^\top \left( \frac{1}{n} J(\boldsymbol{x})^\top J(\boldsymbol{x}) + \lambda_n \boldsymbol{I} \right)^{-1} \\
&\left( \boldsymbol{I} - \exp\left( -2\eta t \left( \frac{1}{n} J(\boldsymbol{x})^\top J(\boldsymbol{x}) + \lambda_n \boldsymbol{I} \right) \right) \right) \left( f_{\theta(0)}(\boldsymbol{x}) - \boldsymbol{y} \right) \\
=& -\frac{1}{n} \boldsymbol{K}(x, \boldsymbol{x}) \left( \frac{1}{n} \boldsymbol{K}(\boldsymbol{x}, \boldsymbol{x}) + \lambda_n \boldsymbol{I} \right)^{-1} \left( \boldsymbol{I} - \exp\left( -2\eta t \left( \frac{1}{n} \boldsymbol{K}(\boldsymbol{x}, \boldsymbol{x}) + \lambda_n \boldsymbol{I} \right) \right) \right) \left( f_{\theta(0)}(\boldsymbol{x}) - \boldsymbol{y} \right)
\end{aligned}
$$

where the last inequality used the notation $\boldsymbol{K}(x, \boldsymbol{x}) = J(x)J(\boldsymbol{x})^\top$ and $\boldsymbol{K}(x_0, \boldsymbol{x}) = J(\boldsymbol{x})J(\boldsymbol{x})^\top$.

Therefore, with sufficient time of training ($t \to \infty$), the final trained network is

$$
\begin{aligned}
f_{\theta(\infty)}(x) &= \lim_{t \to \infty} f_{\theta(t)}(x) \\
&= f_{\theta(0)}(x) - \frac{1}{n} \boldsymbol{K}(x, \boldsymbol{x}) \left( \frac{1}{n} \boldsymbol{K}(\boldsymbol{x}, \boldsymbol{x}) + \lambda_n \boldsymbol{I} \right)^{-1} \left( f_{\theta(0)}(\boldsymbol{x}) - \boldsymbol{y} \right) \\
&= f_{\theta(0)}(x) + \boldsymbol{K}(x, \boldsymbol{x}) \left( \boldsymbol{K}(\boldsymbol{x}, \boldsymbol{x}) + n\lambda_n \boldsymbol{I} \right)^{-1} \left( \boldsymbol{y} - f_{\theta(0)}(\boldsymbol{x}) \right) \\
&= s_{\theta^b}(x) + \boldsymbol{K}(x, \boldsymbol{x}) \left( \boldsymbol{K}(\boldsymbol{x}, \boldsymbol{x}) + n\lambda_n \boldsymbol{I} \right)^{-1} \left( \boldsymbol{y} - s_{\theta^b}(\boldsymbol{x}) \right)
\end{aligned}
$$

where in the last inequality, we use $s_{\theta^b}$ to represent the network initialized with the parameter $\theta^b$ as in the main paper. Summarizing the above discussion, we conclude that

**Assumption C.2.** Suppose that the network training is specified as follows:

1. The network adopts the NTK parametrization and its parameters are randomly initialized using He initialization.

2. The network is sufficiently (infinitely) wide so that the linearized neural network assumption (Assumption C.1) holds.

3. The network is trained using the loss function in (1) via continuous-time gradient flow by feeding the entire training data and using sufficient training time ($t \to \infty$).

**Proposition C.3.** *Suppose that Assumption C.2 holds. Then the final trained network is given by*

$$\hat{h}_{n,\theta^b}(x) = s_{\theta^b}(x) + \boldsymbol{K}(x, \boldsymbol{x}) \left(\boldsymbol{K}(\boldsymbol{x}, \boldsymbol{x}) + \lambda_n n \boldsymbol{I}\right)^{-1} \left(\boldsymbol{y} - s_{\theta^b}(\boldsymbol{x})\right). \tag{20}$$

*For the rest part of Proposition 3.1, please refer to Lemma C.4 below.*

**Shifted kernel ridge regression.** On the other hand, we build a shifted kernel ridge regression based on the NTK as follows.

Let $\bar{\mathcal{H}}$ be the reproducing kernel Hilbert space constructed from the kernel function $K(x, x')$ (the population NTK); See Appendix B. Consider the following kernel ridge regression problem on the RKHS $\bar{\mathcal{H}}$:

$$s_{\theta^b} + \arg\min_{g \in \bar{\mathcal{H}}} \frac{1}{n} \sum_{i=1}^{n} (y_i - s_{\theta^b}(x_i) - g(x_i))^2 + \lambda_n \|g\|_{\bar{\mathcal{H}}}^2 \tag{21}$$

where $\lambda_n$ is a regularization hyper-parameter and $s_{\theta^b}$ is a known given function.

**Lemma C.4.** (20) *is the solution to the following kernel ridge regression problem*

$$s_{\theta^b} + \arg\min_{g \in \bar{\mathcal{H}}} \frac{1}{n} \sum_{i=1}^{n} (y_i - s_{\theta^b}(x_i) - g(x_i))^2 + \lambda_n \|g\|_{\bar{\mathcal{H}}}^2. \tag{22}$$

*Proof of Lemma C.4.* We first let $\tilde{y}_i = y_i - s_{\theta^b}(x_i)$ be the shifted label. Now consider the kernel ridge regression problem

$$\arg\min_{g \in \bar{\mathcal{H}}} \frac{1}{n} \sum_{i=1}^{n} (\tilde{y}_i - g(x_i))^2 + \lambda_n \|g\|_{\bar{\mathcal{H}}}^2.$$

Standard theory in kernel ridge regression (Proposition B.1) shows that the closed-form solution to the above problem is given by:

$$\boldsymbol{K}(x, \boldsymbol{x})^T (\boldsymbol{K}(\boldsymbol{x}, \boldsymbol{x}) + \lambda_n n \boldsymbol{I})^{-1} \tilde{\boldsymbol{y}} = \boldsymbol{K}(x, \boldsymbol{x})^T (\boldsymbol{K}(\boldsymbol{x}, \boldsymbol{x}) + \lambda_n n \boldsymbol{I})^{-1} (\boldsymbol{y} - s_{\theta^b}(\boldsymbol{x}))$$

as $K$ is the kernel function of $\bar{\mathcal{H}}$. This equation corresponds to the definition of $\hat{h}_{n,\theta^b}(x) - s_{\theta^b}(x)$ in (20). $\qquad\square$

From Proposition C.3 and Lemma C.4, we immediately conclude that

*The shifted kernel ridge regressor using NTK with a shift from an initial function $s_{\theta^b}$ is exactly the linearized neural network regressor that starts from the initial network $s_{\theta^b}$.*

## Appendix D  Additional Remarks

We make the following additional remarks to explain some of the details in the main paper:

1. **Regarding Proposition 3.1 (Proposition C.3) in Section 3.** Proposition 3.1 is the theoretical foundation of this paper to develop our framework for efficient uncertainty quantification and reduction for neural networks. We recognize the limitation of this proposition: The linearized neural network assumption (Assumption C.1) must hold to guarantee that the exact equality (2) holds in Proposition 3.1. In reality, the network can never be infinitely wide, and thus an additional error will appear in (2). Unfortunately, with finite network width, this error might be roughly estimated up to a certain order but still involves some unknown constants that hide beneath the data and network [34, 7, 80], which is extremely difficult to quantify precisely in practice. This work does not deal with such an error from finite network width. Instead, we assume that the linearized neural network assumption readily holds as the starting point for developing our uncertainty quantification framework. Equivalently, one may view our work as uncertainty quantification for linearized neural networks.

2. **Regarding the regularization parameters in Section 3.** We introduce the regularization parameters $\lambda_n > 0$ with the regularization term centering at the randomly initialized network parameter in (1) throughout this paper. This form of regularized loss is also adopted in [119, 117]. It has some advantages: 1) A common assumption in NTK theory is to assume that the smallest eigenvalue of the NTK Gram matrix is bounded away from zero [34, 7, 80]. With $\lambda_n > 0$ is introduced, there is an additional term $\lambda_n n \boldsymbol{I}$ in the NTK Gram matrix so that this assumption is always valid naturally. 2) It helps to stabilize the inversion of the NTK Gram matrix if any computation is needed. 3) It can be naturally linked to the well-known kernel ridge regression, which we leverage in this work to develop uncertainty quantification approaches.

On the other hand, we note that with $\lambda_0 = \lim_{n \to \infty} \lambda_n > 0$, the "best" predictor $h^*$ in (6) given in our framework is not exactly the same as the ground-truth regression function $g_\pi^* := \mathbb{E}_{(X,Y) \sim \pi}[Y|X = x]$. In fact, Proposition B.2 shows that $h^* = \bar{s} + (\mathcal{O}_K + \lambda_0 I)^{-1} \mathcal{O}_K(g_\pi^* - \bar{s}) = (\mathcal{O}_K + \lambda_0 I)^{-1} \mathcal{O}_K(g_\pi^*) \neq g_\pi^*$. However, previous work [109, 106, 112] shows that with some conditions, $h^* - g_\pi^* \to 0$ in some metrics when $\lambda_0 \to 0$. Therefore, we may use a very small limit value $\lambda_0$ in practice to make the error $h^* - g_\pi^*$ negligible, as we did in our experiments.

3. **Regarding naive resampling approaches such as "bootstrap on a deep ensemble" mentioned in Section 3.1.** We show that bootstrap on the deep ensemble estimator $\hat{h}_n^m(x)$ face computational challenges in constructing confidence intervals. Since $\hat{h}_n^m(x)$ involves two randomnesses, data variability and procedural variability, we need to bring up two asymptotic normalities to handle them. One approach is first to consider procedural variability given data and then to consider data variability, as described as follows. Another approach is first to consider data variability given procedure and then to consider procedural variability. This will encounter similar challenges as the first approach, so we only discuss the first approach here.

We intuitively present the technical discussions to explain the computational challenges. First, conditional on the training data of size $n$, by the central limit theorem, we have $\sqrt{m}(\hat{h}_n^m(x) - \hat{h}_n^*(x)) \overset{d}{\approx} \mathcal{N}(0, \xi_n^2)$ as $m \to \infty$ where $\overset{d}{\approx}$ represents "approximately equal in distribution" and $\xi_n^2 = \mathrm{Var}_{P_{\theta^b}}(\hat{h}_{n,\theta^b}(x))$ which depends on the training data. Second, we assume that the central limit theorem holds for $\hat{h}_n^*(x)$ (which is rigorously justified in Section 3.3): $\sqrt{n}(\hat{h}_n^*(x) - h^*(x)) \Rightarrow \mathcal{N}(0, \zeta^2)$ as $n \to \infty$. Moreover, we assume for simplicity that we can add up the above two convergences in distribution to obtain

$$\sqrt{n}(\hat{h}_n^m(x) - h^*(x)) \overset{d}{\approx} \mathcal{N}(0, \zeta^2 + \frac{n}{m}\xi_n^2).$$

Note that $\hat{h}_n^m(x) - \hat{h}_n^*(x)$ and $\hat{h}_n^*(x) - h^*(x)$ are clearly dependent, both depending on the training data of size $n$. Therefore, the statistical correctness of the above summation of two convergences in distribution needs to be justified, e.g., using techniques in [48, 75, 77]. However, even if we assume that all the technical parts can be addressed and the asymptotic normality of $\hat{h}_n^m(x)$ is established, computation issues still arise in the "bootstrap on a deep ensemble" approach: 1) "$\frac{n}{m}\xi_n^2$" is meaningful in the limit if we have the limit $\frac{n}{m}\xi_n^2 \to \alpha$. This typically requires the ensemble size $m$ to be large, at least in the order $\omega(1)$, which increases the computational cost of the deep ensemble predictor. However, this is not the only computationally demanding part. 2) The estimation of the asymptotic variance "$\zeta^2 + \frac{n}{m}\xi_n^2$" requires additional computational effort. When using resampling techniques such as the bootstrap, we need to resample $R$ times to obtain $(\hat{h}_n^m(x))^{*j}$ where $j \in [R]$. This implies that we need $Rm$ network training times in total, and according to Point 1), it is equivalent to at least $R \times \omega(1)$ network training times, which is extremely computationally demanding. Therefore, naive use of the "bootstrap on a deep ensemble" approach is barely feasible in neural networks.

# Appendix E   Proofs

In this section, we provide technical proofs of the results in the paper.

*Proof of Theorem 3.2.* In terms of estimation bias, we note that

$$\mathbb{E}[\hat{h}_n^m(x)|\mathcal{D}] = \frac{1}{m}\sum_{i=1}^m \mathbb{E}[\hat{h}_{n,\theta_i^b}(x)|\mathcal{D}] = \mathbb{E}[\hat{h}_{n,\theta^b}(x)|\mathcal{D}] = \hat{h}_n^*(x).$$

as shown in (3). This implies that the deep ensemble predictor and the single network predictor have the same conditional procedural mean given the data. In terms of variance, note that $\hat{h}_{n,\theta_1^b}(x), ..., \hat{h}_{n,\theta_m^b}(x)$, derived in (2), are conditionally independent given $\mathcal{D}$. Therefore the deep ensemble predictor gives

$$\mathrm{Var}(\hat{h}_n^m(x)|\mathcal{D}) = \frac{1}{m^2}\sum_{i=1}^m \mathrm{Var}(\hat{h}_{n,\theta_i^b}(x)|\mathcal{D}) = \frac{1}{m}\mathrm{Var}(\hat{h}_{n,\theta^b}(x)|\mathcal{D})$$

while the single model prediction gives $\mathrm{Var}(\hat{h}_{n,\theta^b}(x)|\mathcal{D})$. Now using conditioning, we have

$$\begin{aligned}
&\mathrm{Var}(\hat{h}_n^m(x))\\
=&\mathrm{Var}(\mathbb{E}[\hat{h}_n^m(x)|\mathcal{D}]) + \mathbb{E}[\mathrm{Var}(\hat{h}_n^m(x)|\mathcal{D})]\\
=&\mathrm{Var}(\hat{h}_n^*(x)) + \frac{1}{m}\mathbb{E}[\mathrm{Var}(\hat{h}_{n,\theta^b}(x)|\mathcal{D})]\\
\leq&\mathrm{Var}(\hat{h}_n^*(x)) + \mathbb{E}[\mathrm{Var}(\hat{h}_{n,\theta^b}(x)|\mathcal{D})]\\
=&\mathrm{Var}(\hat{h}_{n,\theta^b}(x))
\end{aligned}$$

as desired. □

*Proof of Theorem 3.3.* Recall that $\hat{\phi}'_{n,\theta^b}(\cdot)$ is obtained by training an auxiliary neural network with data $\{(x_1, \bar{s}(x_1)), (x_2, \bar{s}(x_2)), ..., (x_n, \bar{s}(x_n))\}$ and the initialization parameters $\theta^b$. Then Proposition 3.1 immediately implies that

$$\hat{\phi}'_{n,\theta^b}(x) = s_{\theta^b}(x) + \boldsymbol{K}(x,\boldsymbol{x})^T(\boldsymbol{K}(\boldsymbol{x},\boldsymbol{x}) + \lambda_n n\boldsymbol{I})^{-1}(\bar{s}(\boldsymbol{x}) - s_{\theta^b}(\boldsymbol{x})).$$

Note that this auxiliary network starts from the same initialization $\theta^b$ as in $\hat{h}_{n,\theta^b}(x)$. Then we have

$$\begin{aligned}
\hat{\phi}_{n,\theta^b}(x) &= \hat{\phi}'_{n,\theta^b}(x) - \bar{s}(x)\\
&= s_{\theta^b}(x) - \bar{s}(x) + \boldsymbol{K}(x,\boldsymbol{x})^T(\boldsymbol{K}(\boldsymbol{x},\boldsymbol{x}) + \lambda_n n\boldsymbol{I})^{-1}(\bar{s}(\boldsymbol{x}) - s_{\theta^b}(\boldsymbol{x}))\\
&= \hat{h}_{n,\theta^b}(x) - \hat{h}_n^*(x)
\end{aligned}$$

by (2) and (3). Hence,

$$\hat{h}_n^*(x) = \hat{h}_{n,\theta^b}(x) - (\hat{\phi}'_{n,\theta^b}(x) - \bar{s}(x)) = \hat{h}_{n,\theta^b}(x) - \hat{\phi}_{n,\theta^b}(x)$$

as desired. □

*Proof of Theorem 3.5.* Theorem 3.3 implies that

$$\hat{h}_{n,\theta^b}(x) - \hat{\phi}_{n,\theta^b}(x) = \hat{h}_n^*(x)$$

which is the solution to the following problem

$$\hat{h}_n^*(\cdot) = \bar{s}(\cdot) + \min_{g\in\bar{\mathcal{H}}}\frac{1}{n}\sum_{i=1}^n[(y_i - \bar{s}(x_i) - g(x_i))^2] + \lambda_n\|g\|_{\bar{\mathcal{H}}}^2$$

Therefore, its corresponding population risk minimization problem is:

$$h^*(\cdot) = \bar{s}(\cdot) + \min_{g\in\bar{\mathcal{H}}}\mathbb{E}_\pi[(Y - \bar{s}(X) - g(X))^2] + \lambda_0\|g\|_{\bar{\mathcal{H}}}^2$$

Applying Proposition 3.4 to $\hat{h}_n^*$ and $h^*$, we have that

$$\sqrt{n}\left((\hat{h}_n^*(x) - \bar{s}(x)) - (h^*(x) - \bar{s}(x))\right) \Rightarrow \mathcal{N}(0, \sigma^2(x))$$

In other words,

$$\sqrt{n}\left(\hat{h}_{n,\theta^b}(x) - \hat{\phi}_{n,\theta^b}(x) - h^*(x)\right) \Rightarrow \mathcal{N}(0, \sigma^2(x))$$

This shows that asymptotically as $n \to \infty$ we have

$$\mathbb{P}\left(-q_{1-\frac{\alpha}{2}} \leq \frac{\hat{h}_{n,\theta^b}(x) - \hat{\phi}_{n,\theta^b}(x) - h^*(x)}{\sigma(x)/\sqrt{n}} \leq q_{1-\frac{\alpha}{2}}\right) \to 1 - \alpha$$

where $q_\alpha$ is the $\alpha$-quantile of standard Gaussian distribution $\mathcal{N}(0,1)$. Hence

$$\left[\hat{h}_{n,\theta^b}(x) - \hat{\phi}_{n,\theta^b}(x) - \frac{\sigma(x)}{\sqrt{n}}q_{1-\frac{\alpha}{2}}, \hat{h}_{n,\theta^b}(x) - \hat{\phi}_{n,\theta^b}(x) + \frac{\sigma(x)}{\sqrt{n}}q_{1-\frac{\alpha}{2}}\right]$$

is an asymptotically exact $(1 - \alpha)$-level confidence interval of $h^*(x)$. $\qquad\square$

*Proof of Theorem 3.6.* Note that the statistics

$$\hat{h}^j_{n',\theta^b}(x) - \hat{\phi}^j_{n',\theta^b}(x)$$

from the $j$-th batch is i.i.d. for $j \in [m']$. Moreover, by Theorem 3.5, we have the asymptotic normality:

$$\sqrt{n'}\left(\hat{h}_{n',\theta^b}(x) - \hat{\phi}_{n',\theta^b}(x) - h^*(x)\right) \Rightarrow \mathcal{N}(0, \sigma^2(x)),$$

as $n \to \infty$ meaning $n' = n/m' \to \infty$. Therefore by the property of Gaussian distribution and the principle of batching, we have

$$\frac{\sqrt{n'}}{\sigma(x)}(\psi_B(x) - h^*(x)) \Rightarrow \frac{1}{m'}\sum_{i=1}^{m'} Z_i$$

and

$$\frac{\psi_B(x) - h^*(x)}{S_B(x)/\sqrt{m'}} \Rightarrow \frac{\frac{1}{m'}\sum_{i=1}^{m'} Z_i}{\sqrt{\frac{1}{m'(m'-1)}\sum_{j=1}^{m'}(Z_j - \frac{1}{m'}\sum_{i=1}^{m'} Z_i)^2}}$$

for i.i.d. $\mathcal{N}(0,1)$ random variables $Z_1, ..., Z_{m'}$, where we use the continuous mapping theorem to deduce the weak convergence. Note that

$$\frac{\frac{1}{m'}\sum_{i=1}^{m'} Z_i}{\sqrt{\frac{1}{m'(m'-1)}\sum_{j=1}^{m'}(Z_j - \frac{1}{m'}\sum_{i=1}^{m'} Z_i)^2}} \stackrel{\mathrm{d}}{=} t_{m'-1}$$

Hence, asymptotically as $n \to \infty$ we have

$$\mathbb{P}\left(-q_{1-\frac{\alpha}{2}} \leq \frac{\psi_B(x) - h^*(x)}{S_B(x)/\sqrt{m'}} \leq q_{1-\frac{\alpha}{2}}\right) \to 1 - \alpha$$

where $q_\alpha$ is the $\alpha$-quantile of the t distribution $t_{m'-1}$ with degree of freedom $m' - 1$. Hence

$$\left[\psi_B(x) - \frac{S_B(x)}{\sqrt{m'}}q_{1-\frac{\alpha}{2}}, \psi_B(x) + \frac{S_B(x)}{\sqrt{m'}}q_{1-\frac{\alpha}{2}}\right]$$

is an asymptotically exact $(1 - \alpha)$-level confidence interval of $h^*(x)$. $\qquad\square$

*Proof of Theorem 3.7.* Note that for $j \in [R]$, the statistics

$$\hat{h}^{*j}_{n,\theta^b}(x) - \hat{\phi}^{*j}_{n,\theta^b}(x)$$

from the $j$-th bootstrap replication are i.i.d. conditional on the dataset $\mathcal{D}$. By Theorem 3.5, we have the asymptotic normality:

$$\sqrt{n}\left(\hat{h}_{n,\theta^b}(x) - \hat{\phi}_{n,\theta^b}(x) - h^*(x)\right) \Rightarrow \mathcal{N}(0, \sigma^2(x)),$$

as $n \to \infty$. By Theorem 2 in [24], we have the following asymptotic normality: For $j \in [R]$,

$$\sqrt{n}\left(\hat{h}_{n,\theta^b}^{*j}(x) - \hat{\phi}_{n,\theta^b}^{*j}(x) - \left(\hat{h}_{n,\theta^b}(x) - \hat{\phi}_{n,\theta^b}(x)\right)\right) \Rightarrow \mathcal{N}(0, \sigma^2(x)),$$

as $n \to \infty$ conditional on the training data $\mathcal{D}$. Therefore Assumption 1 in [75] is satisfies and thus Theorem 1 in [75] holds, showing that asymptotically as $n \to \infty$, we have

$$\mathbb{P}\left(-q_{1-\frac{\alpha}{2}} \leq \frac{\psi_C(x) - h^*(x)}{S_C(x)} \leq q_{1-\frac{\alpha}{2}}\right) \to 1 - \alpha$$

where $q_\alpha$ is the $\alpha$-quantile of the t distribution $t_R$ with degree of freedom $R$. Hence

$$\left[\psi_C(x) - S_C(x)q_{1-\frac{\alpha}{2}}, \psi_C(x) + S_C(x)q_{1-\frac{\alpha}{2}}\right]$$

is an asymptotically exact $(1-\alpha)$-level confidence interval of $h^*(x)$. □

*Proof of Theorem B.6.* Recall that $\Phi$ is the feature map associated with the kernel $k$ of the RKHS $\mathcal{H}$. We apply the influence function formula in [25] (see also [31, 50]) to obtain the infinitesimal jackknife:

$$IF(z; T, \pi_n) = -S_{\pi_n}^{-1}(2\lambda_0 g_{\pi_n, \lambda_0}) + \mathcal{L}'(z_y - g_{\pi_n, \lambda_0}(z_x))S_{\pi_n}^{-1}\Phi(z_x)$$

where $S_P : \mathcal{H} \to \mathcal{H}$ is defined by

$$S_P(f) = 2\lambda_0 f + \mathbb{E}_P[\mathcal{L}''(Y, g_{P,\lambda_0}(X))f(X)\Phi(X)].$$

This can be seen from (14) by setting $P = \pi_n$ and using the closed-form solution of the kernel ridge regression.

To compute the exact formula, we need to obtain $S_{\pi_n}$ and $S_{\pi_n}^{-1}$. Since the loss function is $\mathcal{L}(\hat{y}, y) = (\hat{y} - y)^2$, we have

$$S_{\pi_n}(f) = 2\lambda_0 f + \mathbb{E}_{\pi_n}[\mathcal{L}''(Y, g_{\pi_n, \lambda_0}(X))f(X)\Phi(X)] = 2\lambda_0 f + \frac{2}{n}\sum_{j=1}^{n} f(x_j)\Phi(x_j).$$

Suppose $S_{\pi_n}^{-1}(2\lambda_0 g_{\pi_n, \lambda_0}) = \tilde{g}_1$. Then at $x_0$, we have

$$2\lambda_0 g_{\pi_n, \lambda_0}(x_0) = S_{\pi_n}(\tilde{g}_1(x_0)) = 2\lambda_0 \tilde{g}_1(x_0) + \frac{2}{n}\sum_{j=1}^{n} \tilde{g}_1(x_j)k(x_0, x_j).$$

Hence,

$$\tilde{g}_1(x_0) = g_{\pi_n, \lambda_0}(x_0) - \frac{1}{\lambda_0 n}\sum_{j=1}^{n} \tilde{g}_1(x_j)k(x_0, x_j).$$

This implies that we need to evaluate $\tilde{g}_1(x_j)$ on training data first, which is straightforward by letting $x_0 = x_1, ..., x_n$:

$$\tilde{g}_1(\boldsymbol{x}) = g_{\pi_n, \lambda_0}(\boldsymbol{x}) - \frac{1}{\lambda_0 n}\boldsymbol{k}(\boldsymbol{x}, \boldsymbol{x})\tilde{g}_1(\boldsymbol{x})$$

so

$$\tilde{g}_1(\boldsymbol{x}) = (\boldsymbol{k}(\boldsymbol{x}, \boldsymbol{x}) + \lambda_0 n\boldsymbol{I})^{-1}(\lambda_0 n\boldsymbol{I})g_{\pi_n, \lambda_0}(\boldsymbol{x})$$

and

$$\tilde{g}_1(x_0) = g_{\pi_n, \lambda_0}(x_0) - \frac{1}{\lambda_0 n}\boldsymbol{k}(x_0, \boldsymbol{x})^T\tilde{g}_1(\boldsymbol{x})$$

$$= g_{\pi_n, \lambda_0}(x_0) - \boldsymbol{k}(x_0, \boldsymbol{x})^T(\boldsymbol{k}(\boldsymbol{x}, \boldsymbol{x}) + \lambda_0 n\boldsymbol{I})^{-1}g_{\pi_n, \lambda_0}(\boldsymbol{x})$$

Next we compute $S_{\pi_n}^{-1}\Phi(z_x) = \tilde{g}_2$. At $x_0$, we have

$$k(z_x, x_0) = \Phi(z_x)(x_0) = S_{\pi_n}(\tilde{g}_2(x_0)) = 2\lambda_0 \tilde{g}_2(x_0) + \frac{2}{n}\sum_{j=1}^{n} \tilde{g}_2(x_j)k(x_0, x_j)$$

Hence

$$\tilde{g}_2(x_0) = \frac{1}{2\lambda_0} k(z_x, x_0) - \frac{1}{\lambda_0 n} \sum_{j=1}^{n} \tilde{g}_2(x_j) k(x_0, x_j)$$

This implies that we need to evaluate $\tilde{g}_2(x_j)$ on training data first, which is straightforward by letting $x_0 = x_1, ..., x_n$:

$$\tilde{g}_2(\boldsymbol{x}) = \frac{1}{2\lambda_0} \boldsymbol{k}(z_x, \boldsymbol{x}) - \frac{1}{\lambda_0 n} \boldsymbol{k}(\boldsymbol{x}, \boldsymbol{x}) \tilde{g}_2(\boldsymbol{x})$$

so

$$\tilde{g}_2(\boldsymbol{x}) = (\boldsymbol{k}(\boldsymbol{x}, \boldsymbol{x}) + \lambda_0 n \boldsymbol{I})^{-1} \left( \frac{n}{2} \boldsymbol{I} \right) \boldsymbol{k}(z_x, \boldsymbol{x})$$

and

$$
\begin{aligned}
\tilde{g}_2(x_0) &= \frac{1}{2\lambda_0} k(z_x, x_0) - \frac{1}{\lambda_0 n} \boldsymbol{k}(x_0, \boldsymbol{x})^T \tilde{g}_2(\boldsymbol{x}) \\
&= \frac{1}{2\lambda_0} k(z_x, x_0) - \frac{1}{2\lambda_0} \boldsymbol{k}(x_0, \boldsymbol{x})^T (\boldsymbol{k}(\boldsymbol{x}, \boldsymbol{x}) + \lambda_0 n \boldsymbol{I})^{-1} \boldsymbol{k}(z_x, \boldsymbol{x})
\end{aligned}
$$

Combing previous results, we obtain

$$
\begin{aligned}
&IF(z; T, \pi_n)(x_0) \\
=&- g_{\pi_n, \lambda_0}(x_0) + \boldsymbol{k}(x_0, \boldsymbol{x})^T (\boldsymbol{k}(\boldsymbol{x}, \boldsymbol{x}) + \lambda_0 n \boldsymbol{I})^{-1} g_{\pi_n, \lambda_0}(\boldsymbol{x}) \\
&+ 2(z_y - g_{\pi_n, \lambda_0}(z_x)) \left( \frac{1}{2\lambda_0} k(z_x, x_0) - \frac{1}{2\lambda_0} \boldsymbol{k}(x_0, \boldsymbol{x})^T (\boldsymbol{k}(\boldsymbol{x}, \boldsymbol{x}) + \lambda_0 n \boldsymbol{I})^{-1} \boldsymbol{k}(z_x, \boldsymbol{x}) \right) \\
=&\boldsymbol{k}(x_0, \boldsymbol{x})^T (\boldsymbol{k}(\boldsymbol{x}, \boldsymbol{x}) + \lambda_0 n \boldsymbol{I})^{-1} \left( g_{\pi_n, \lambda_0}(\boldsymbol{x}) - \frac{1}{\lambda_0}(z_y - g_{\pi_n, \lambda_0}(z_x)) \boldsymbol{k}(z_x, \boldsymbol{x}) \right) \\
&- g_{\pi_n, \lambda_0}(x_0) + \frac{1}{\lambda_0}(z_y - g_{\pi_n, \lambda_0}(z_x)) k(z_x, x_0) \\
=&\boldsymbol{k}(x_0, \boldsymbol{x})^T (\boldsymbol{k}(\boldsymbol{x}, \boldsymbol{x}) + \lambda_0 n \boldsymbol{I})^{-1} M_z(\boldsymbol{x}) - M_z(x_0)
\end{aligned}
$$

as desired. $\qquad\square$

*Proof of Theorem B.7.* Recall from Section B.2, we use the equivalence notations:

$$IF(z; T, P) = T'_P(\delta_z - P).$$

First, we prove the following Claim:

$$\sup_{z \in \mathcal{X} \times \mathcal{Y}} \|T'_\pi(\delta_z - \pi) - T'_{\pi_n}(\delta_z - \pi_n)\|_{\mathcal{H}} \to 0, \quad a.s. \tag{23}$$

From Lemma A.6. and Lemma A.7. in [50], we have

$$T'_P(Q) = -S_P^{-1}(W_P(Q)).$$

In this equation,

$$W_P : \text{lin}(B_S) \to \mathcal{H}, \quad Q \mapsto \mathbb{E}_Q[\mathcal{L}'(Y, g_{P, \lambda_0}(X)) \Phi(X)]$$

is a continuous linear operator on $\text{lin}(B_S)$ where the space $\text{lin}(B_S)$ is defined in [50]. Moreover, the operator norm of $W_P$ satisfies $\|W_P\| \leq 1$. In addition,

$$S_P : \mathcal{H} \to \mathcal{H}, \quad f \mapsto 2\lambda_0 f + \mathbb{E}_P[\mathcal{L}''(Y, g_{P, \lambda_0}(X)) f(X) \Phi(X)]$$

is an invertible continuous linear operator on $\mathcal{H}$ where $\Phi$ is the feature map of $\mathcal{H}$. This implies that $S_P^{-1} : \mathcal{H} \to \mathcal{H}$ is also a continuous linear operator on $\mathcal{H}$.

Next, we apply Lemma A.7 in [50] by considering the following two sequences: The first sequence is $\delta_z - \pi_n \in \text{lin}(B_S)$ which obviously satisfies

$$\|(\delta_z - \pi_n) - (\delta_z - \pi)\|_\infty = \|\pi - \pi_n\|_\infty \to 0, \quad a.s.$$

The second sequence is $\pi_n \in B_S$ which obviously satisfies

$$\|\pi - \pi_n\|_\infty \to 0, \quad a.s.$$

Following the proof of Lemma A.7 in [50], we have that

$$
\begin{aligned}
&\|T'_\pi(\delta_z - \pi) - T'_{\pi_n}(\delta_z - \pi_n)\|_{\mathcal{H}} \\
=&\|S_\pi^{-1}(W_\pi(\delta_z - \pi)) - S_{\pi_n}^{-1}(W_{\pi_n}(\delta_z - \pi_n))\|_{\mathcal{H}} \\
\leq&\|S_\pi^{-1}(W_\pi(\delta_z - \pi)) - S_\pi^{-1}(W_{\pi_n}(\delta_z - \pi))\|_{\mathcal{H}} + \|S_\pi^{-1}(W_{\pi_n}(\delta_z - \pi)) - S_\pi^{-1}(W_{\pi_n}(\delta_z - \pi_n))\|_{\mathcal{H}} \\
&+ \|S_\pi^{-1}(W_{\pi_n}(\delta_z - \pi_n)) - S_{\pi_n}^{-1}(W_{\pi_n}(\delta_z - \pi_n))\|_{\mathcal{H}} \\
\leq&\|S_\pi^{-1}\|\|W_\pi(\delta_z - \pi) - W_{\pi_n}(\delta_z - \pi)\|_{\mathcal{H}} + \|S_\pi^{-1}\|\|W_{\pi_n}\|\|(\delta_z - \pi) - (\delta_z - \pi_n)\|_\infty \\
&+ \|S_\pi^{-1} - S_{\pi_n}^{-1}\|\|W_{\pi_n}\|\|\delta_z - \pi_n\|_\infty \\
\leq&\|S_\pi^{-1}\|\|W_\pi(\pi) - W_{\pi_n}(\pi)\|_{\mathcal{H}} + \|S_\pi^{-1}\|\|W_{\pi_n}\|\|\pi - \pi_n\|_\infty \quad \text{(since } W_{\pi_n} \text{ is a linear operator)} \\
&+ \|S_\pi^{-1} - S_{\pi_n}^{-1}\|\|W_{\pi_n}\| (\|\delta_z - \pi\|_\infty + \|\pi - \pi_n\|_\infty)
\end{aligned}
$$

Therefore, we only need to study the three terms in the last equation. Note that the first two terms are independent of $z$, and thus it follows from Step 3 and Step 4 in the proof of Lemma A.7 in [50] that

$$
\begin{aligned}
&\sup_{z \in \mathcal{X} \times \mathcal{Y}} \|S_\pi^{-1}\|\|W_\pi(\pi) - W_{\pi_n}(\pi)\|_{\mathcal{H}} + \|S_\pi^{-1}\|\|W_{\pi_n}\|\|\pi - \pi_n\|_\infty \\
=&\|S_\pi^{-1}\|\|W_\pi(\pi) - W_{\pi_n}(\pi)\|_{\mathcal{H}} + \|S_\pi^{-1}\|\|W_{\pi_n}\|\|\pi - \pi_n\|_\infty \to 0, \quad a.s.
\end{aligned}
$$

To show that the third term also satisfies that

$$
\sup_{z \in \mathcal{X} \times \mathcal{Y}} \|S_\pi^{-1} - S_{\pi_n}^{-1}\|\|W_{\pi_n}\| (\|\delta_z - \pi\|_\infty + \|\pi - \pi_n\|_\infty) \to 0, \quad a.s.,
$$

we only need to note the following fact:
1) $\|S_\pi^{-1} - S_{\pi_n}^{-1}\| \to 0$, $a.s.$, by Step 2 in the proof of Lemma A.7 in [50]. Moreover, this equation is independent of $z$.
2) $\|W_{\pi_n}\| \leq 1$ by Step 1 in the proof of Lemma A.7 in [50]. Moreover, this equation is independent of $z$.
3) $\|\pi - \pi_n\|_\infty < +\infty$, $a.s.$, since $\|\pi - \pi_n\|_\infty \to 0$, $a.s.$. Moreover, this equation is independent of $z$.
4) $\sup_{z \in \mathcal{X} \times \mathcal{Y}} \|\delta_z - \pi\|_\infty < +\infty$. To see this, we note that by definition

$$
\sup_{z \in \mathcal{X} \times \mathcal{Y}} \|\delta_z - \pi\|_\infty = \sup_{z \in \mathcal{X} \times \mathcal{Y}} \sup_{g \in \mathcal{G}} \left| \int g \, d(\delta_z - \pi) \right| = \sup_{z \in \mathcal{X} \times \mathcal{Y}} \sup_{g \in \mathcal{G}} \left| g(z) - \int g \, d\pi \right| \leq 2 \sup_{g \in \mathcal{G}} \|g\|_\infty
$$

where the last term is independent of $z$ and the space $\mathcal{G} = \mathcal{G}_1 \cup \mathcal{G}_2 \cup \{b\}$ is defined in [50]. $\sup_{g \in \mathcal{G}_1} \|g\|_\infty \leq 1$ by the definition of $\mathcal{G}_1$. $\sup_{g \in \{b\}} \|g\|_\infty < +\infty$ by our additional assumption on $b$. Since both $\mathcal{X}$ and $\mathcal{Y}$ are bounded and closed, $\sup_{x \in \mathcal{X}} \sqrt{k(x, x)}$ is bounded above by, say $\kappa < \infty$. Thus, for every $h \in \mathcal{H}$ with $\|h\|_{\mathcal{H}} \leq C_1$, we have $\|h\|_\infty \leq C_1 \kappa$. By definition of $\mathcal{G}_2$, for any $g \in \mathcal{G}_2$, we can write $g(x, y) = \mathcal{L}'(y, f_0(x)) f_1(x)$ with $\|f_0\|_{\mathcal{H}} \leq c_0$ and $\|f_1\|_{\mathcal{H}} \leq 1$. Note that $\|f_1\|_{\mathcal{H}} \leq 1$ implies that $\|f_1\|_\infty \leq \kappa$ and $\|f_0\|_{\mathcal{H}} \leq c_0$ implies that $\|f_0\|_\infty \leq c_0 \kappa$. Thus Assumption B.4 shows that $\|\mathcal{L}'(y, f_0(x))\|_\infty \leq b'_{c_0\kappa}(y)$ which is bounded on $\mathcal{Y}$ by our additional assumption (uniformly for $f_0$). Hence we conclude that $\sup_{g \in \mathcal{G}_2} \|g\|_\infty \leq \kappa \sup_{y \in \mathcal{Y}} b'_{c_0\kappa}(y) < +\infty$. Summarizing the above discussion, we obtain that

$$
\sup_{z \in \mathcal{X} \times \mathcal{Y}} \|\delta_z - \pi\|_\infty \leq 2 \sup_{g \in \mathcal{G}} \|g\|_\infty < +\infty
$$

Combining the above points 1)-4), we conclude that

$$
\sup_{z \in \mathcal{X} \times \mathcal{Y}} \|S_\pi^{-1} - S_{\pi_n}^{-1}\|\|W_{\pi_n}\| (\|\delta_z - \pi\|_\infty + \|\pi - \pi_n\|_\infty) \to 0, \quad a.s.
$$

Hence, we obtain our Claim (23).

Applying $k_{x_0}$ to (23), the reproducing property of $k$ implies that

$$
\sup_{z \in \mathcal{X} \times \mathcal{Y}} \left| T'_\pi(\delta_z - \pi)(x_0) - T'_{\pi_n}(\delta_z - \pi_n)(x_0) \right| \to 0, \quad a.s.
$$

Note that since $\mathcal{X}$ and $\mathcal{Y}$ are bounded and closed by our assumptions, we have that

$$\left| \int_{z \in \mathcal{X} \times \mathcal{Y}} \left( T'_\pi (\delta_z - \pi)(x_0) \right)^2 d\pi(z) - \int_{z \in \mathcal{X} \times \mathcal{Y}} \left( T'_{\pi_n} (\delta_z - \pi_n)(x_0) \right)^2 d\pi(z) \right|$$

$$\leq |\pi(\mathcal{X} \times \mathcal{Y})| \times \sup_{z \in \mathcal{X} \times \mathcal{Y}} \left| T'_\pi (\delta_z - \pi)(x_0) - T'_{\pi_n} (\delta_z - \pi_n)(x_0) \right|$$

$$\times \left| \int_{z \in \mathcal{X} \times \mathcal{Y}} \left( T'_\pi (\delta_z - \pi)(x_0) + T'_{\pi_n} (\delta_z - \pi_n)(x_0) \right) d\pi(z) \right|$$

$$\to 0, \quad a.s.$$

where we use the fact that $\int_{z \in \mathcal{X} \times \mathcal{Y}} \left( T'_\pi (\delta_z - \pi)(x_0) \right)^2 d\pi(z) = \xi^2(x_0) < +\infty$. On the other hand, it follows from the strong law of large numbers that

$$\frac{1}{n} \sum_{z_i \in \mathcal{D}} \left( T'_{\pi_n} (\delta_{z_i} - \pi_n)(x_0) \right)^2 - \int_{z \in \mathcal{X} \times \mathcal{Y}} \left( T'_{\pi_n} (\delta_z - \pi_n)(x_0) \right)^2 d\pi(z) \to 0, \quad a.s.$$

Hence, we conclude that

$$\frac{1}{n} \sum_{z_i \in \mathcal{D}} \left( T'_{\pi_n} (\delta_{z_i} - \pi_n)(x_0) \right)^2 - \int_{z \in \mathcal{X} \times \mathcal{Y}} \left( T'_\pi (\delta_z - \pi)(x_0) \right)^2 d\pi(z) \to 0, \quad a.s.$$

In other words,

$$\hat{\xi}^2(x_0) = \frac{1}{n} \sum_{z_i \in \mathcal{D}} IF^2(z_i; T, \pi_n)(x_0) \to \int_{z \in \mathcal{X} \times \mathcal{Y}} IF^2(z; T, \pi)(x) d\pi(z) = \xi^2(x_0), \quad a.s.$$

For confidence intervals, the proof is straightforward. Theorem B.5 shows that as $n \to \infty$ we have

$$\frac{g_{\pi_n, \lambda_n}(x_0) - g_{\pi, \lambda_0}(x_0)}{\xi(x_0)/\sqrt{n}} \Rightarrow \mathcal{N}(0, 1)$$

By Slutsky's theorem and $\frac{\xi(x_0)}{\hat{\xi}(x_0)} \to 1$, $a.s.$, we have

$$\frac{g_{\pi_n, \lambda_n}(x_0) - g_{\pi, \lambda_0}(x_0)}{\hat{\xi}(x_0)/\sqrt{n}} \Rightarrow \mathcal{N}(0, 1)$$

This implies that asymptotically as $n \to \infty$, we have

$$\mathbb{P} \left( -q_{1-\frac{\alpha}{2}} \leq \frac{g_{\pi_n, \lambda_n}(x_0) - g_{\pi, \lambda_0}(x_0)}{\hat{\xi}(x_0)/\sqrt{n}} \leq q_{1-\frac{\alpha}{2}} \right) \to 1 - \alpha$$

where $q_\alpha$ is the $\alpha$-quantile of standard Gaussian distribution $\mathcal{N}(0, 1)$. Hence

$$\left[ g_{\pi_n, \lambda_n}(x_0) - \frac{\hat{\xi}(x_0)}{\sqrt{n}} q_{1-\frac{\alpha}{2}}, g_{\pi_n, \lambda_n}(x_0) + \frac{\hat{\xi}(x_0)}{\sqrt{n}} q_{1-\frac{\alpha}{2}} \right]$$

is an asymptotically exact $(1 - \alpha)$-level confidence interval of $g_{\pi, \lambda_0}(x)$. $\qquad \square$

# Appendix F   Experiments: Details and More Results

## F.1   Experimental Details

We provide more details about our experimental implementation in Section 4.

Throughout our experiments, we use a two-layer fully-connected neural network as the base predictor based on the NTK specifications in Section C (Proposition C.3). However, we need to resolve the conflicts between the theoretical assumptions therein (e.g., continuous-time gradient flow) and practical implementation (e.g., discrete-time gradient descent), and at the same time, guarantee that the training procedure indeed operates in the NTK regime so that the first-order Taylor expansion (linearized neural network assumption) works well [64, 34, 80, 118]. Therefore, we will use the following specifications in all experiments.

1. The network adopts the NTK parametrization, and its parameters are randomly initialized using He initialization. The ReLU activation function is used in the network.

2. The network has $32 \times n$ hidden neurons in its hidden layer where $n$ is the size of the entire training data. The network should be sufficiently wide so that the NTK theory holds.

3. The network is trained using the regularized square loss (1) with regularization hyper-parameter $\lambda_n \equiv 0.1^{10}$.

4. The network is trained using the (full) batch gradient descent (by feeding the whole dataset).

5. The learning rate and training epochs are properly tuned based on the specific dataset. The epochs should not be too small since the training needs to converge to a good solution, but the learning rate should also not be too large because we need to stipulate that the training procedure indeed operates in the NTK regime (an area around the initialization). Note that in practice, we cannot use the continuous-time gradient flow, and the network can never be infinitely wide. Therefore, with the fixed learning rate in gradient descent, we do not greatly increase the number of epochs so that the training procedure will likely find a solution that is not too far from the initialization.

6. We set $m' = 4$ in the PNC-enhanced batching approach and $R = 4$ in the PNC-enhanced cheap bootstrap approach.

7. In DropoutUQ, the dropout rate is a crucial hyper-parameter. We find that the dropout rate has a significant impact on the interval width of DropoutUQ: A large dropout rate produces a wide interval since the dropout rate is linked to the variance of the prior Gaussian distribution [43]. Therefore, to make fair comparisons between different approaches, we adjust the dropout rate in DropoutUQ so that they have a similar or larger interval width as PNC-enhanced batching or PNC-enhanced cheap bootstrap.

8. All experiments are conducted on a single GeForce RTX 2080 Ti GPU.

## F.2 Additional Experiments

In this section, we present additional experimental results on more datasets. These experimental results further support our observations and claims in Section 4, demonstrating the robustness and effectiveness of our proposals.

First, we consider additional synthetic datasets.

Synthetic Datasets #2: $X \sim \text{Unif}([0, 0.2]^d)$ and $Y \sim \sum_{i=1}^{d} X^{(i)} \sin(X^{(i)}) + \mathcal{N}(0, 0.001^2)$. The training set $\mathcal{D} = \{(x_i, y_i) : i = 1, ..., n\}$ is formed by i.i.d. samples of $(X, Y)$ with sample size $n$ from the above data generating process. We use $x_0 = (0.1, 0.1, ..., 0.1)$ and $y_0 = \sum_{i=1}^{d} 0.1 \sin(0.1)$ as the fixed test point in the confidence interval task. The rest of the setup is the same as Section 4. The implementation specifications are the same as in Section F.1. The results for constructing confidence intervals are displayed in Table 3, and the results for reducing procedural variability are displayed in Table 4.

Next, we consider real-world datasets.

It is worth mentioning the main reason for adopting synthetic datasets in our experiments. That is, precise evaluation of a confidence interval requires the following two critical components, which nevertheless are not available in real-world data: 1) One needs to know the ground-truth regression function to check the coverage of confidence intervals, while the label in real-world data typically contains some aleatoric noise. 2) To estimate the coverage rate of the confidence interval, we need to repeat the experiments multiple times by regenerating new independent datasets from the same data distribution. In practice, we cannot regenerate new real-world datasets.

However, to conduct simulative experiments on realistic real-world data, we provide a possible way to mimic an evaluation of the confidence interval on real-world datasets as follows.

Step 1. For a given real-world dataset, select a fixed test data point $(x_0, y_0)$ and a subset of the real-world data $(x_i, y_i)$ $(i \in \mathcal{I})$ that does not include $(x_0, y_0)$.

Step 2. For each experimental repetition $j \in [J]$, add certain artificial independent noise on the label to obtain a "simulated" real-world dataset $(x_i, y_i + \epsilon_{i,j})$ $(i \in \mathcal{I})$ where $\epsilon_{i,j}$ are all independent

Table 3: Confidence interval construction on synthetic datasets #2 with different data sizes $n = 128, 256, 512, 1024$ and different data dimensions $d = 2, 4, 8, 16$. The CR results that attain the desired confidence level $95\%/90\%$ are in **bold**.

| | PNC-enhanced batching | | | PNC-enhanced cheap bootstrap | | | DropoutUQ | | |
|---|---|---|---|---|---|---|---|---|---|
| | 95%CI(CR/IW) | 90%CI(CR/IW) | MP | 95%CI(CR/IW) | 90%CI(CR/IW) | MP | 95%CI(CR/IW) | 90%CI(CR/IW) | MP |
| $(d = 2)$ | | | | | | | | | |
| $n = 128$ | **0.98**/0.0075 | **0.91**/0.0056 | 0.0201 | **1.00**/0.0343 | **0.99**/0.0264 | 0.0184 | **0.99**/0.0392 | **0.97**/0.0330 | 0.0216 |
| $n = 256$ | **0.97**/0.0058 | **0.93**/0.0043 | 0.0197 | **0.97**/0.0215 | **0.93**/0.0165 | 0.0185 | **0.96**/0.0213 | **0.94**/0.0169 | 0.0214 |
| $n = 512$ | **0.96**/0.0050 | **0.95**/0.0037 | 0.0199 | 0.94/0.0148 | **0.90**/0.0114 | 0.0186 | **0.96**/0.0164 | **0.90**/0.0135 | 0.0216 |
| $n = 1024$ | **0.96**/0.0041 | **0.94**/0.0030 | 0.0199 | **0.95**/0.0115 | **0.93**/0.0088 | 0.0203 | 0.91/0.0124 | 0.86/0.0102 | 0.0210 |
| $(d = 4)$ | | | | | | | | | |
| $n = 128$ | **0.98**/0.0111 | **0.97**/0.0082 | 0.0395 | **1.00**/0.0443 | **0.98**/0.0340 | 0.0370 | **0.96**/0.0462 | **0.93**/0.0375 | 0.0441 |
| $n = 256$ | **0.98**/0.0088 | **0.92**/0.0065 | 0.0401 | **0.98**/0.0346 | **0.94**/0.0266 | 0.0374 | **0.98**/0.0344 | **0.94**/0.0282 | 0.0426 |
| $n = 512$ | 0.94/0.0073 | **0.91**/0.0054 | 0.0403 | **0.99**/0.0236 | **0.94**/0.0181 | 0.0373 | 0.94/0.0310 | 0.88/0.0260 | 0.0427 |
| $n = 1024$ | **1.00**/0.0055 | **0.91**/0.0040 | 0.0399 | 0.94/0.0167 | 0.87/0.0128 | 0.0373 | **0.95**/0.0241 | **0.90**/0.0203 | 0.0417 |
| $(d = 8)$ | | | | | | | | | |
| $n = 128$ | **0.97**/0.0145 | **0.93**/0.0107 | 0.0798 | **1.00**/0.0683 | **1.00**/0.0524 | 0.0849 | 0.94/0.0661 | 0.86/0.0548 | 0.0876 |
| $n = 256$ | **0.96**/0.0110 | **0.94**/0.0082 | 0.0801 | **0.99**/0.0540 | **0.97**/0.0414 | 0.0846 | 0.92/0.0538 | **0.91**/0.0446 | 0.0853 |
| $n = 512$ | **0.97**/0.0092 | **0.92**/0.0068 | 0.0793 | **0.99**/0.0393 | **0.96**/0.0302 | 0.0846 | 0.91/0.0427 | 0.85/0.0359 | 0.0862 |
| $n = 1024$ | **0.95**/0.0084 | **0.90**/0.0062 | 0.0792 | **0.97**/0.0284 | **0.95**/0.0218 | 0.0820 | 0.85/0.0295 | 0.79/0.0248 | 0.0809 |
| $(d = 16)$ | | | | | | | | | |
| $n = 128$ | **0.95**/0.0214 | **0.94**/0.0158 | 0.1608 | **1.00**/0.1004 | **0.99**/0.0771 | 0.1568 | 0.92/0.1124 | 0.87/0.0958 | 0.1814 |
| $n = 256$ | **0.97**/0.0174 | **0.91**/0.0129 | 0.1595 | **1.00**/0.0725 | **1.00**/0.0556 | 0.1578 | **0.95**/0.0800 | 0.88/0.0677 | 0.1720 |
| $n = 512$ | **0.96**/0.0140 | **0.92**/0.0104 | 0.1604 | **0.99**/0.0516 | **0.98**/0.0396 | 0.1569 | 0.88/0.0565 | 0.80/0.0475 | 0.1689 |
| $n = 1024$ | **0.95**/0.0114 | 0.88/0.0084 | 0.1599 | **0.98**/0.0350 | **0.95**/0.0269 | 0.1568 | 0.86/0.0584 | 0.83/0.0498 | 0.1685 |

Table 4: Reducing procedural variability to improve prediction on synthetic datasets #2 with different data sizes $n = 128, 256, 512, 1024$ and different data dimensions $d = 2, 4, 8, 16$. The best MSE results are in **bold**.

| MSE | One base network | PNC predictor | Deep ensemble (5 networks) | Deep ensemble (2 networks) |
|---|---|---|---|---|
| $(d = 2)$ | | | | |
| $n = 128$ | $(1.02 \pm 1.16) \times 10^{-4}$ | $\mathbf{(1.38 \pm 0.18) \times 10^{-5}}$ | $(2.90 \pm 1.72) \times 10^{-5}$ | $(5.39 \pm 3.01) \times 10^{-5}$ |
| $n = 256$ | $(7.36 \pm 4.84) \times 10^{-5}$ | $\mathbf{(1.28 \pm 0.10) \times 10^{-5}}$ | $(2.39 \pm 1.17) \times 10^{-5}$ | $(3.67 \pm 1.61) \times 10^{-5}$ |
| $n = 512$ | $(4.41 \pm 2.34) \times 10^{-5}$ | $\mathbf{(1.11 \pm 0.10) \times 10^{-5}}$ | $(1.79 \pm 5.19) \times 10^{-5}$ | $(2.52 \pm 1.55) \times 10^{-5}$ |
| $n = 1024$ | $(3.46 \pm 1.91) \times 10^{-5}$ | $\mathbf{(9.78 \pm 0.94) \times 10^{-6}}$ | $(1.03 \pm 0.39) \times 10^{-5}$ | $(1.82 \pm 1.27) \times 10^{-5}$ |
| $(d = 4)$ | | | | |
| $n = 128$ | $(1.89 \pm 1.23) \times 10^{-4}$ | $\mathbf{(1.98 \pm 0.25) \times 10^{-5}}$ | $(4.64 \pm 1.49) \times 10^{-5}$ | $(1.02 \pm 0.31) \times 10^{-4}$ |
| $n = 256$ | $(1.59 \pm 0.47) \times 10^{-4}$ | $\mathbf{(1.49 \pm 0.08) \times 10^{-5}}$ | $(4.55 \pm 1.58) \times 10^{-5}$ | $(1.02 \pm 0.37) \times 10^{-4}$ |
| $n = 512$ | $(1.72 \pm 0.65) \times 10^{-4}$ | $\mathbf{(1.29 \pm 0.05) \times 10^{-5}}$ | $(3.73 \pm 0.88) \times 10^{-5}$ | $(7.78 \pm 1.72) \times 10^{-5}$ |
| $n = 1024$ | $(9.75 \pm 3.49) \times 10^{-5}$ | $\mathbf{(1.05 \pm 0.03) \times 10^{-5}}$ | $(3.70 \pm 0.79) \times 10^{-5}$ | $(8.92 \pm 2.09) \times 10^{-5}$ |
| $(d = 8)$ | | | | |
| $n = 128$ | $(8.00 \pm 1.36) \times 10^{-4}$ | $\mathbf{(5.07 \pm 0.65) \times 10^{-5}}$ | $(2.18 \pm 0.59) \times 10^{-4}$ | $(4.77 \pm 1.32) \times 10^{-4}$ |
| $n = 256$ | $(4.50 \pm 0.30) \times 10^{-4}$ | $\mathbf{(2.30 \pm 0.24) \times 10^{-5}}$ | $(9.44 \pm 1.44) \times 10^{-5}$ | $(2.08 \pm 0.19) \times 10^{-4}$ |
| $n = 512$ | $(3.80 \pm 0.35) \times 10^{-4}$ | $\mathbf{(1.47 \pm 0.07) \times 10^{-5}}$ | $(8.44 \pm 1.11) \times 10^{-5}$ | $(2.02 \pm 0.23) \times 10^{-4}$ |
| $n = 1024$ | $(3.46 \pm 0.50) \times 10^{-4}$ | $\mathbf{(1.12 \pm 0.04) \times 10^{-5}}$ | $(8.11 \pm 1.27) \times 10^{-5}$ | $(1.71 \pm 0.23) \times 10^{-4}$ |
| $(d = 16)$ | | | | |
| $n = 128$ | $(2.32 \pm 0.24) \times 10^{-3}$ | $\mathbf{(1.39 \pm 0.14) \times 10^{-4}}$ | $(5.71 \pm 0.74) \times 10^{-4}$ | $(1.36 \pm 0.30) \times 10^{-3}$ |
| $n = 256$ | $(1.86 \pm 0.21) \times 10^{-3}$ | $\mathbf{(6.55 \pm 0.75) \times 10^{-5}}$ | $(3.96 \pm 0.34) \times 10^{-4}$ | $(9.49 \pm 0.67) \times 10^{-4}$ |
| $n = 512$ | $(1.42 \pm 0.08) \times 10^{-3}$ | $\mathbf{(3.32 \pm 0.29) \times 10^{-5}}$ | $(3.00 \pm 0.16) \times 10^{-4}$ | $(7.14 \pm 0.42) \times 10^{-4}$ |
| $n = 1024$ | $(1.18 \pm 0.06) \times 10^{-3}$ | $\mathbf{(2.58 \pm 0.24) \times 10^{-5}}$ | $(2.56 \pm 0.18) \times 10^{-4}$ | $(6.30 \pm 0.39) \times 10^{-4}$ |

Gaussian random variables. Construct a confidence interval based on this "regenerating" training data $(x_i, y_i + \epsilon_{i,j})$ $(i \in \mathcal{I})$ and evaluate it on $(x_0, y_0)$.

In the above setting, $y_0$ is treated as the true mean response of $x_0$ without aleatoric noise, and $\epsilon_{i,j}$ represents the only data variability. As discussed above, precise evaluation is impossible for real-world data; the above procedure, although not precise, is the best we can do to provide a resembling evaluation. Using this setting, we conduct experiments on real-world benchmark regression datasets from UCI datasets: Boston, Concrete, and Energy. These results are shown in Table 5.

From the results, we observe that our approaches also work well for these "simulated" real-world datasets. Under various problem settings, our proposed approach can robustly provide accurate confidence intervals that satisfy the coverage requirement of the confidence level. In contrast, DropoutUQ does not have such statistical guarantees. Overall, our approaches not only work for synthetic datasets but also is scalable to be applied potentially in benchmark real-world datasets.

Finally, we conduct experiments of reducing procedural variability to show the performance improvement capability of proposed models and report MSE results on the "simulated" real-world datasets: Boston, Concrete, and Energy. In each experiment, we use an 80%/20% random split for training data and test data in the original dataset, and then report MSE on its corresponding "simulated" dataset from our proposed PNC predictor and other baseline approaches with 10 experimental repetition times. These results are shown in Table 6.

Table 5: Confidence interval construction on "simulated" real-world benchmark datasets: Boston, Concrete, and Energy. $J = 40$. The CR results that attain the desired confidence level 95%/90% are in **bold**.

| | PNC-enhanced batching | | | PNC-enhanced cheap bootstrap | | | DropoutUQ | | |
|---|---|---|---|---|---|---|---|---|---|
| | 95%CI(CR/IW) | 90%CI(CR/IW) | MP | 95%CI(CR/IW) | 90%CI(CR/IW) | MP | 95%CI(CR/IW) | 90%CI(CR/IW) | MP |
| **Boston** | | | | | | | | | |
| Test 1 | **1.0**/1.172 | **0.975**/0.866 | 0.486 | 0.925/1.024 | 0.875/0.786 | 0.333 | 0.85/0.997 | 0.7/0.832 | 0.109 |
| Test 2 | **1.0**/0.873 | **0.95**/0.645 | 0.100 | **0.975**/0.869 | **0.925**/0.667 | 0.042 | 0.75/0.981 | 0.625/0.842 | 0.089 |
| Test 3 | **1.0**/0.896 | **1.0**/0.662 | -0.047 | **0.975**/1.473 | 0.85/1.131 | 0.0875 | 0.725/1.005 | 0.575/0.866 | -0.119 |
| Test 4 | **1.0**/0.915 | **1.0**/0.676 | 0.219 | 0.925/0.829 | 0.875/0.637 | 0.372 | 0.90/1.164 | 0.825/0.981 | 0.131 |
| Test 5 | **1.0**/0.587 | **1.0**/0.434 | 0.145 | **1.0**/0.531 | **0.975**/0.407 | 0.272 | 0.75/1.093 | 0.725/0.920 | 0.454 |
| **Concrete** | | | | | | | | | |
| Test 1 | **1.0**/1.378 | **1.0**/1.019 | 0.416 | **0.975**/1.013 | **0.975**/0.778 | 0.379 | 0.725/1.176 | 0.675/1.016 | 0.088 |
| Test 2 | **1.0**/1.583 | **1.0**/1.171 | 0.161 | **1.0**/0.874 | **1.0**/0.671 | 0.362 | 0.675/1.112 | 0.65/0.952 | 0.780 |
| Test 3 | **1.0**/1.673 | **1.0**/1.237 | 0.0455 | **1.0**/0.970 | **0.975**/0.745 | 0.516 | 0.875/1.061 | 0.75/0.887 | 0.597 |
| Test 4 | **1.0**/1.156 | **1.0**/0.855 | 0.484 | **1.0**/0.960 | **0.95**/0.737 | 0.466 | 0.875/1.101 | 0.825/0.949 | 0.552 |
| Test 5 | **1.0**/1.208 | **1.0**/0.893 | 0.216 | **1.0**/0.601 | **1.0**/0.462 | 0.309 | 0.825/1.027 | 0.8/0.880 | 0.684 |
| **Energy** | | | | | | | | | |
| Test 1 | **0.975**/0.566 | **0.975**/0.418 | 0.331 | **1.0**/0.596 | **1.0**/0.458 | 0.152 | 0.75/0.854 | 0.725/0.733 | 0.037 |
| Test 2 | **1.0**/0.422 | **1.0**/0.312 | 0.294 | **0.95**/0.570 | 0.875/0.437 | 0.191 | 0.80/0.611 | 0.725/0.504 | 0.231 |
| Test 3 | **1.0**/0.435 | **1.0**/0.321 | -0.734 | **1.0**/0.724 | **1.0**/0.556 | -0.766 | 0.75/0.749 | 0.675/0.630 | -0.908 |
| Test 4 | **1.0**/0.572 | **1.0**/0.423 | -0.723 | **1.0**/0.613 | **1.0**/0.470 | -0.792 | 0.825/0.651 | 0.775/0.549 | -0.847 |
| Test 5 | **0.975**/0.512 | **0.975**/0.379 | -0.679 | **0.975**/0.550 | **0.975**/0.422 | -0.702 | 0.825/0.644 | 0.80/0.550 | -0.807 |

Table 6: Reducing procedural variability to improve prediction on "simulated" real-world benchmark datasets: Boston, Concrete, and Energy. The best MSE results are in **bold**.

| MSE | One base network | PNC predictor | Deep ensemble (5 networks) | Deep ensemble (2 networks) |
|---|---|---|---|---|
| **Boston** | | | | |
| Test 1 | $(3.52 \pm 0.78) \times 10^{-1}$ | $\mathbf{(1.04 \pm 0.03) \times 10^{-1}}$ | $(1.56 \pm 0.28) \times 10^{-1}$ | $(2.24 \pm 0.51) \times 10^{-1}$ |
| Test 2 | $(3.15 \pm 0.59) \times 10^{-1}$ | $\mathbf{(1.01 \pm 0.04) \times 10^{-1}}$ | $(1.38 \pm 0.26) \times 10^{-1}$ | $(2.03 \pm 0.47) \times 10^{-1}$ |
| Test 3 | $(4.18 \pm 0.69) \times 10^{-1}$ | $\mathbf{(1.93 \pm 0.05) \times 10^{-1}}$ | $(2.28 \pm 0.33) \times 10^{-1}$ | $(2.68 \pm 0.37) \times 10^{-1}$ |
| Test 4 | $(3.33 \pm 0.81) \times 10^{-1}$ | $\mathbf{(1.17 \pm 0.06) \times 10^{-1}}$ | $(1.48 \pm 0.17) \times 10^{-1}$ | $(1.84 \pm 0.24) \times 10^{-1}$ |
| Test 5 | $(3.46 \pm 0.45) \times 10^{-1}$ | $\mathbf{(1.52 \pm 0.04) \times 10^{-1}}$ | $(1.99 \pm 0.31) \times 10^{-1}$ | $(2.45 \pm 0.32) \times 10^{-1}$ |
| **Concrete** | | | | |
| Test 1 | $(2.47 \pm 0.24) \times 10^{-1}$ | $\mathbf{(1.90 \pm 0.04) \times 10^{-1}}$ | $(2.02 \pm 0.07) \times 10^{-1}$ | $(2.21 \pm 0.14) \times 10^{-1}$ |
| Test 2 | $(3.00 \pm 0.29) \times 10^{-1}$ | $\mathbf{(2.18 \pm 0.04) \times 10^{-1}}$ | $(2.28 \pm 0.15) \times 10^{-1}$ | $(2.43 \pm 0.16) \times 10^{-1}$ |
| Test 3 | $(2.39 \pm 0.16) \times 10^{-1}$ | $\mathbf{(1.85 \pm 0.04) \times 10^{-1}}$ | $(2.01 \pm 0.11) \times 10^{-1}$ | $(2.18 \pm 0.08) \times 10^{-1}$ |
| Test 4 | $(2.85 \pm 0.55) \times 10^{-1}$ | $\mathbf{(1.84 \pm 0.02) \times 10^{-1}}$ | $(1.92 \pm 0.14) \times 10^{-1}$ | $(2.17 \pm 0.38) \times 10^{-1}$ |
| Test 5 | $(2.51 \pm 0.19) \times 10^{-1}$ | $\mathbf{(1.75 \pm 0.04) \times 10^{-1}}$ | $(1.86 \pm 0.06) \times 10^{-1}$ | $(2.16 \pm 0.16) \times 10^{-1}$ |
| **Energy** | | | | |
| Test 1 | $(1.09 \pm 0.14) \times 10^{-1}$ | $\mathbf{(7.73 \pm 0.08) \times 10^{-2}}$ | $(8.56 \pm 0.64) \times 10^{-2}$ | $(9.91 \pm 0.86) \times 10^{-2}$ |
| Test 2 | $(1.08 \pm 0.10) \times 10^{-1}$ | $\mathbf{(7.46 \pm 0.17) \times 10^{-2}}$ | $(8.29 \pm 0.44) \times 10^{-2}$ | $(9.57 \pm 0.84) \times 10^{-2}$ |
| Test 3 | $(1.23 \pm 0.10) \times 10^{-1}$ | $\mathbf{(7.51 \pm 0.12) \times 10^{-2}}$ | $(8.45 \pm 0.43) \times 10^{-2}$ | $(1.01 \pm 0.12) \times 10^{-1}$ |
| Test 4 | $(1.25 \pm 0.12) \times 10^{-1}$ | $\mathbf{(7.24 \pm 0.16) \times 10^{-2}}$ | $(8.31 \pm 0.91) \times 10^{-2}$ | $(1.07 \pm 0.19) \times 10^{-1}$ |
| Test 5 | $(1.12 \pm 0.09) \times 10^{-1}$ | $\mathbf{(6.81 \pm 0.23) \times 10^{-2}}$ | $(7.97 \pm 0.44) \times 10^{-2}$ | $(9.69 \pm 0.57) \times 10^{-2}$ |

## F.3 Confidence Intervals for Coverage Results

When evaluating the performance of confidence intervals, the CR value is computed based on a finite number of repetitions, which will incur a binomial error on the (population) coverage estimate. Therefore, in addition to reporting CR as a point estimate, we also report a confidence interval for the (population) coverage. Table 7 reports the Clopper–Pearson exact binomial confidence interval. Note that this interval can be computed straightforwardly by the point estimate (CR) and the number of repetitions ($J$). Therefore, we can refer to Table 7 to additionally obtain a confidence interval of coverage for our results, e.g., in Tables 1 and 3.

Table 7: Clopper–Pearson exact binomial confidence interval for coverage estimates at confidence level 95%. $J = 100$.

| CR value | Confidence interval of coverage | CR value | Confidence interval of coverage | CR value | Confidence interval of coverage |
|---|---|---|---|---|---|
| 1.00 | $[0.964, 1.000]$ | 0.99 | $[0.946, 1.000]$ | 0.98 | $[0.930, 0.998]$ |
| 0.97 | $[0.915, 0.994]$ | 0.96 | $[0.901, 0.989]$ | 0.95 | $[0.887, 0.984]$ |
| 0.94 | $[0.874, 0.978]$ | 0.93 | $[0.861, 0.971]$ | 0.92 | $[0.848, 0.965]$ |
| 0.91 | $[0.836, 0.958]$ | 0.90 | $[0.824, 0.951]$ | 0.89 | $[0.812, 0.944]$ |
| 0.88 | $[0.800, 0.936]$ | 0.87 | $[0.788, 0.929]$ | 0.86 | $[0.776, 0.921]$ |
| 0.85 | $[0.765, 0.914]$ | 0.84 | $[0.753, 0.906]$ | 0.83 | $[0.742, 0.898]$ |
| 0.82 | $[0.731, 0.890]$ | 0.81 | $[0.719, 0.882]$ | 0.80 | $[0.708, 0.873]$ |

