# OpenReview forum: "Efficient Uncertainty Quantification and Reduction for Over-Parameterized Neural Networks"
_NeurIPS.cc/2023/Conference — NeurIPS 2023 poster_

### Official Review · Reviewer_nEzW · 2023-06-28

**Soundness:** 3 good
**Presentation:** 3 good
**Contribution:** 3 good
**Rating:** 7
**Confidence:** 3

**Summary:**

This paper proposes an epistemic uncertainty assessment framework which comes with statistical coverage guarantees and low computation costs for over-parametrized neural networks.  This approach seeks to remove procedural uncertainty by using one auxiliary network. In addition, approaches are provided to construct confidence intervals using a small number of retrained networks on different problem settings.


**Strengths:**

The authors introduce a Procedural-Noise-Correcting (PNC) predictor, which uses one auxiliary network trained on the data but where all labels set to zero (instead of +-1).  This mimicks the variability coming from the training procedure leveraging asymptotic results from NTK theory. The idea is simple and intuitive and appears to work on some toy experiments.

**Weaknesses:**

Experiments are only provided on low-dimensional toy problems.

In addition, for the coverage studies, only 40 samples are used which results in relatively large binomial errors on the reported coverage properties. This issue isn't discussed at all. It would be better to both increase the number of samples and to report the results in a way that respects the uncertainty due to finite samples.

**Questions:**

How wide is wide enough that the NTK theory used is approximately valid?

Have you looked into studying more realistic problems (even if still only simulated)?


**Limitations:**

Some discussion should be provided on limitations. For example, how large does the NN need to be?

---

> ### Author Rebuttal · Authors · 2023-08-10
>
> We greatly appreciate the reviewer for the careful reading and valuable feedback. We address the reviewer's concerns below.
>
> Q1. (Toy problems) To fully evaluate our approaches, we conducted additional experiments on the real-world dataset. For additional experiments, please refer to our global response to all reviewers and our discussions there.
>
> Q2. (Selection on repeat times) The 40 here represents 40 experimental repetitions: Essentially, we regenerate the entire datasets 40 times and re-run our approaches on new datasets 40 times to generate 40 confidence intervals. We tried more repetition times, such as 80 times in the experiment: the plot of the coverage rate and interval width against the number of experimental repetitions is shown in the attached PDF file (in our global response to all reviewers). As shown, 40 times is sufficiently large to guarantee a robust evaluation of the coverage rate and interval width.
>
> Q3. (Width in NTK) There are some rough lower bounds of the width in the literature so that the NTK holds approximately; See, e.g.,  [*], [**].
>
> Essentially, they show that the width should be at least a polynomial function of the sample size as well as other constants that are intrinsic in the data. In the experiment, we choose a 32$\times$sample size (main paper) or 128$\times$sample size (Appendix) to reflect this dependence. Note that the bounds derived from theory are rough and typically too large to be implemented in practice, so we need to balance it with our actual computation units.
>
> Q4. (Experiments) For additional experiments on real-world datasets, please refer to our global response to all reviewers and our discussions there.
>
> [*]: Arora, Sanjeev, et al. "On exact computation with an infinitely wide neural net." Advances in neural information processing systems 32 (2019).
>
> [**]: Du, Simon, et al. "Gradient descent finds global minima of deep neural networks." International conference on machine learning. PMLR, 2019.

---

> > ### Comment · Reviewer_nEzW · 2023-08-13
> >
> > I think you missed the point about the coverage estimates. Any finite number of repetitions used to estimate the coverage will incur a binomial error on the coverage estimate. In the paper you ignore this and have examples that are 1 sigma from the desired asymptotic value that are labeled as undercoverage. I would suggest representing that better and doing more repetitions to reduce the uncertainty.
> >
> > Nevertheless based on the other replies I've raised by score by one.

---

> > > ### Author Response · Authors · 2023-08-14
> > >
> > > We greatly appreciate the reviewer for increasing our score, and we also apologize for not adequately addressing the issue on coverage estimation accuracy. We now see the point of the reviewer. In the final version, we will 1) increase the number of experimental repetitions from 40 to at least in the hundreds. 2) report the confidence interval of the coverage by using the binomial or sample proportion confidence interval. Both of these would serve to better represent the comparison results.

---

### Official Review · Reviewer_bPtq · 2023-06-29

**Soundness:** 3 good
**Presentation:** 2 fair
**Contribution:** 3 good
**Rating:** 6
**Confidence:** 3

**Summary:**

The paper focuses on uncertainty quantification, specifically on estimating the statistical range of predictions made by a deep neural network (DNN). This is achieved through the use of a novel DNN called Procedural Noise Correcting (PNC), which is capable of estimating the variability inherent in the training process, particularly the variability arising from different initializations of the DNN. In order to obtain a more accurate estimate of the idealized DNN, denoted as h^*, the authors employ the Neural Tangent Kernel (NTK) and train the PNC DNN. To construct the confidence interval for the PNC, they utilize either batching or bootstrap methods.

**Strengths:**

In this paper, the authors introduce a remarkable deep neural network (DNN) named PNC, which effectively addresses uncertainty quantification. The presentation of the paper is well written, although there are a few missing notations. However, the supplementary materials provided by the authors greatly enhance the overall completeness of the paper. The problem tackled by the authors is intriguing, and to the best of my knowledge, I have not come across any other papers that address this particular issue using a similar approach.


**Weaknesses:**

There are several instances of missing notations in the paper, which make it challenging to comprehend. I'm curious about the authors' decision to only consider the random initialization of network parameters when discussing procedural uncertainty. Why did they not take into account random batches or the non-deterministic backpropagation? Algorithm 1 is also difficult for me to understand. If only \bar(s) is predicted, how can h_n*(x) be obtained? Is it necessary to apply the NTK? I did not see it explicitly mentioned in algorithm 1.

Regarding the experimental protocol, I believe it could benefit from a more robust approach. It might be valuable to compare the proposed method to other approaches, such as:

[1] Blundell, Charles, et al. "Weight uncertainty in neural network." International conference on machine learning. PMLR, 2015.
[2] Maddox, W. J., Izmailov, P., Garipov, T., Vetrov, D. P., & Wilson, A. G. (2019). A simple baseline for bayesian uncertainty in deep learning. Advances in neural information processing systems, 32.

[3] He, Bobby, Balaji Lakshminarayanan, and Yee Whye Teh. "Bayesian deep ensembles via the neural tangent kernel." Advances in neural information processing systems 33 (2020): 1010-1022.

Additionally, I find it intriguing to include experiments on additional datasets such as MNIST, CIFAR-10, and CIFAR-100, not limited to regression tasks but also for classification purposes. It would provide valuable insights into how the proposed DNN performs in the presence of epistemic uncertainty. Why not evaluate its behavior by applying out-of-distribution (OOD) data?

The authors highlight that PNC involves minimal computation; however, they only apply their technique to relatively simple datasets. This raises questions about whether it truly qualifies as a low-budget technique.


**Questions:**

Please see Weaknesses

**Limitations:**

The authors' paper lacks emphasis on the societal impact of their work. Typically, I wouldn't expect authors in the field of Uncertainty Quantification to delve into this aspect. However, I do feel that the claims made in the paper are overly assertive. For instance, on line 29, the paper mentions general uncertainty quantification, yet it only focuses on regression tasks. Additionally, the claim of working with Overparametrized Neural Networks seems exaggerated, considering that the DNN employed in the study consists of just two fully connected layers. It would be advisable for the authors to exercise caution when making such claims.

---

> ### Author Rebuttal · Authors · 2023-08-10
>
> We greatly appreciate the reviewer for the careful reading and valuable feedback. We address the reviewer's concerns below.
>
> Q1. (NTK and algorithms) Most of the essential notations are introduced in the statement of Proposition 3.1 and at the beginning of Section 3.1, while we defer the details related to NTK in Appendix C due to the page limit. Regarding the missing notations, we would appreciate it if the reviewer could give more guidance.
>
> NTK is the essential component of our theory, which provides the theoretical foundation to establish the statistical properties of all our algorithms, including justifying Algorithm 1. Therefore, we need the NTK theory to derive the statistical guarantee of Algorithm 1.
>
> The major difference between our work and previous literature is that our UQ framework has simultaneous $\textit{statistical coverage guarantee}$ and $\textit{low computation cost}$. Some Bayesian approaches are fast-to-implement empirically but do not (or have not been shown to) enjoy the statistical coverage guarantee that ours does. The major theoretical difficulty in neural networks, compared with classical models such as linear regression, is that neural networks are non-convex and have many global minima, making standard statistical theory impossible to be applied. The recent NTK theory gives a statistical characterization of neural network training, which provides a possible foundation for theoretical UQ. With this, our theoretical framework can characterize the uncertainty from the random initialization, build connections between UQ for neural networks and UQ for kernel-based regression, and develop large-sample asymptotic theory, which takes the very first step in the UQ framework with statistical coverage guarantee, even only considering random initialization. It is more appealing to incorporate random batches or SGD in the framework, but then it becomes unclear how to obtain statistical coverage guarantees, which is left as our future research direction.
>
> $\bar{s}$ is not a “prediction”; it is the average of the initial network (without using data or training). $\bar{s}$ is used to generate the procedural-shifted label in Step 2, which is then used to train the second network. $\hat{h}^*_{n}(x)$ is the output of Algorithm 1, which is the subtraction of two neural networks from Steps 1 and 2, respectively.
>
> Q2. (Compare with Bayesian approaches) We cited these works and discussed them in Section 1: “While powerful, these approaches nonetheless possess inference error that could be hard to quantify, and ultimately finding rigorous guarantees on the performance of these approximate posteriors remains open to our best knowledge.” The major difference between our work and these works is that our UQ framework has a statistical coverage guarantee. Some Bayesian approaches are fast-to-implement empirically but do not (or have not been shown to) enjoy the statistical coverage guarantee that ours does. For the illustration in our experiments, DropoutUQ is probably the most cited (well-known) work in deep-learning-based UQ, and could be a representative of these Bayesian methods without statistical guarantees.
>
> Q3. (Classification) NTK is the essential component of our theory, providing the theoretical foundation to establish the statistical properties of our algorithms. In previous literature, NTK is generally employed for mean-square-error (MSE) loss, e.g.,
>
> Du, Simon, et al. "Gradient descent finds global minima of deep neural networks." International conference on machine learning. PMLR, 2019.
>
> Lee, Jaehoon, et al. "Wide neural networks of any depth evolve as linear models under gradient descent." Advances in neural information processing systems 32 (2019).
>
> as well as other references for NTK in our paper. The major technical discrepancy related to NTK between the regression tasks and the classification tasks is that the softmax operation in the network final layer and the cross-entropy-type loss used in the classification does not have an explicit statistical characterization of neural network training. Therefore, most NTK-based theories consider MSE loss. It would be an interesting but different direction to study the statistical guarantee for classification neural networks, which we recognize as our future research direction.
>
> Q4. (Computation) First of all, we have added more experiments on “simulated” real-world datasets to validate our performances; please refer to our global response to all reviewers and our discussions there. We also want to emphasize that our rigorous theory holds for any $m’\ge 2$ in PNC-enhanced batching and $R \ge 1$ in PNC-enhanced cheap bootstrap, meaning that our approaches can construct asymptotically exact-coverage confidence intervals using as low as two PNC procedures (four network training in total) without additional overheads. In terms of running times, it is as low as 4 times the standard network (point prediction) training time.
>
> Q5. (Terminology) We understand the reviewer’s concern and would definitely remove or clarify the overclaims in the paper (e.g., “general” uncertainty quantification, etc). Nonetheless, we should point out that “Overparametrized” used in deep learning theory, by convention, means the width is sufficiently large, while the depth can be as small as 2, e.g.,
>
> Li, Yuanzhi, Tengyu Ma, and Hongyang R. Zhang. "Learning over-parametrized two-layer neural networks beyond NTK." Conference on learning theory. PMLR, 2020.
>
> Li, Yuanzhi, and Yingyu Liang. "Learning overparameterized neural networks via stochastic gradient descent on structured data." Advances in neural information processing systems 31 (2018).
>
> NTK theory and our theory are applied to neural networks with any depth. In experiments, we use a two-layer neural network for illustration. So, this terminology appears fine to us, but we would clarify it in the paper as suggested by the reviewer.

---

> > ### Comment · Reviewer_bPtq · 2023-08-15
> >
> > Dear Authors,
> >
> > I extend my gratitude for your response. After revisiting the paper alongside your answer, I find that the content has become notably clearer.
> > Regarding your responses to **points 1-4**, I find myself in agreement with the explanations provided by the authors.
> >
> > As for **point 5**, I hold a differing perspective from what the authors refer to as a 'Deep' Neural network. While I may not entirely align with the terminology used, acknowledging the variation in nomenclature within the field, I am open to conceding this point.
> >
> > In light of the clarifications and persuasive arguments presented, **I am pleased to convey my decision to revise my initial review.**

---

> > > ### Author Response · Authors · 2023-08-16
> > >
> > > We greatly appreciate the reviewer for your support and for increasing our score. We are also glad to hear that our response helped make the contents notably clearer. Regarding the terminology, we will make sure to clarify it in the final version to avoid overclaims in the paper as kindly suggested by the reviewer.

---

### Official Review · Reviewer_cgMr · 2023-06-30

**Soundness:** 3 good
**Presentation:** 3 good
**Contribution:** 3 good
**Rating:** 6
**Confidence:** 3

**Summary:**

The authors focus on the task of uncertainty quantification for neural networks.
They contribute (i) a procedure to remove procedural uncertainty, an uncertainty that arises due to
randomness in the training procedure, and (ii) an approach to cheaply construct confidence intervals with asymptotic coverage guarantees. The proposal is evaluated on two small synthetic experiments.


**Strengths:**

- The proposed approach offers a cheap (compared to ensemble models) and principled approach to remove procedural uncertainty and construct confidence intervals.
- The paper is well written.


**Weaknesses:**

The main weakness of the paper is its empirical evaluation, which consists of a single simple data set, a sum of sine functions, evaluated with small data sets (up to 1024) in very low dimensional spaces (up to eight dimensions). Any kind of scalability guarantee is not given/evaluated, which is concerning, especially since the theory assumes access to the full training data (l118).
A second synthetic function (sum of exponentials and squared covariates) is evaluated in the appendix, again without too many details.


### Minor
- Throughout the paper, the term _extremely_ is used repeatedly (extremely fast, extremely time-consuming, extremely few) instead of providing proper quantitative numbers (if only in the order of magnitudes, or relative to other approaches).
- Prop 3.4 $\pi_n$ is only defined in the appendix
- When submitting to NeurIPS, please follow the NeurIPS style guide, i.e., place your captions above tables, no vertical lines in tables, etc.
- The regularizing $\lambda$ is fixed to $\lambda \equiv 0.1^{10}$, i.e., essentially zero? Why keep it at all?


**Questions:**

- The evaluation is limited to a single function in a very low-dimensional setting. Can the authors provide a further experimental evaluation with respect to the scalability of the proposed approach?
- Wrt the paragraph around l174-l193. I am not sure I understand this. $\bar s \equiv 0.2$, or $\bar s \equiv 0$, gives us a constant target, that is independent of the input, i.e., $\hat \phi_{n, \theta^b}'(\cdot)$ should learn to be constant, giving us $\bar \phi_{n,\theta^b} \equiv 0.2$, i.e., step 2 of Algorithm 1 becomes irrelevant. Which part am I misunderstanding?


**Limitations:**

Discussion on limitations and broader impact are missing.

---

> ### Author Rebuttal · Authors · 2023-08-10
>
> We greatly appreciate the reviewer for the careful reading and valuable feedback. We address the reviewer's concerns below.
>
> Q1. (“extremely”) Thanks for pointing this out. We agree that “extremely” should be clarified. In the revised version, we will avoid using “extremely” but provide more concrete numbers to elaborate our point. For instance, we will change,
>
> “Given these computational bottleneck, we will utilize light-computation resampling alternatives, including batching and the so-called cheap bootstrap method, which allows valid confidence interval construction using extremely few model retrainings.”
>
> to
>
> “Given these computational bottleneck, we will utilize light-computation resampling alternatives, including batching and the so-called cheap bootstrap method, which allows valid confidence interval construction using as few as one additional model retraining.”
>
> Q2. ($\pi_n$) We will add this definition to the main body.
>
> Q3. (Format) We will revise the format of our tables accordingly.
>
> Q4. (Regularization) Our theory applies to any regularization greater than zero, regardless of how small it is. We introduce it in our theory so that the eigenvalue of the NTK Gram matrix will be bounded away from zero and thus the inversion of the regularized NTK Gram matrix can be computed stably. In practice, the regularization is typically very small or simply zero, e.g., in
>
> Lee, Jaehoon, et al. "Wide neural networks of any depth evolve as linear models under gradient descent." Advances in neural information processing systems 32 (2019).
>
> We maintain a small but nonzero regularization in the experiment to match our theoretical results.
>
> Q5. (Experiments) We understand this concern. We have carried out additional experiments on more sophisticated problems using “simulated” real-world data. Please refer to our global response to all reviewers.
>
> Q6. (Constant label) Even if every training data has label 0, the output network is not a zero-constant network: The major difference between neural networks and classical models such as linear regression, is that neural networks are non-convex and have many local/global minima. Starting from an initialization parameter $\theta^b$, gradient descent will find a global minimum that is very close to $\theta^b$ (but not $0$ in general although $0$ is one of the global minima). This phenomenon has been characterized by the NTK theory or lazy training, e.g.,
>
> Du, Simon, et al. "Gradient descent finds global minima of deep neural networks." International conference on machine learning. PMLR, 2019.
>
> Chizat, Lenaic, Edouard Oyallon, and Francis Bach. "On lazy training in differentiable programming." Advances in neural information processing systems 32 (2019).
>
> Another way to see this is that in Proposition 3.1, if we plug in $\textbf{y}=0$ (zero-label on all training data), we will not get a zero-constant network. Instead, the output network depends on the initialization network. This phenomenon also highlights the need to consider random initialization as an important uncertainty component in the UQ task.

---

> > ### Comment · Reviewer_cgMr · 2023-08-16
> >
> > Thank you for your clarifications. I keep my score.

---

> > > ### Author Response · Authors · 2023-08-16
> > >
> > > Thank you very much. We appreciate the reviewer for your reading and reply.

---

### Official Review · Reviewer_VEbh · 2023-07-03

**Soundness:** 3 good
**Presentation:** 2 fair
**Contribution:** 3 good
**Rating:** 6
**Confidence:** 3

**Summary:**

In the paper, authors present a new approach to quantify and mitigate a specific aspect of epistemic uncertainty in model predictions. They identify and quantify "procedural variability," a type of epistemic uncertainty that arises from noise in the training process. Based on the Neural Tangent Kernel (NTK) theory, the authors introduce a "procedure-noise-correction" method. They offer theoretical support and experimental evidence to justify their approach.

**Strengths:**

Generally, the majority of research papers propose methods that quantify the overall epistemic uncertainty, but a few discuss the estimation and evaluation of its different components. The process of splitting epistemic uncertainty into different parts, while not a novel concept, remains significantly under-studied. Therefore, the problem addressed by this paper is promising and brings potential value for further studies.

The strengths of this paper are:

1) The authors built a mathematical framework for their proposed method.
2) The authors' approach to quantify and remove procedural variability is indeed novel and opens new perspectives for future works.
3) Combined with the PNC-Enhanced batching, the presented approach seems practical, and practitioners in the field may find it beneficial.
4) The supplementary material is verbose and resembles a textbook, providing a comprehensive guide to the details of derivations, proofs, and certain theoretical aspects like the NTK.

**Weaknesses:**

There are some aspects of this paper I consider as weaknesses:

1. Narrative of the Paper: Despite its overall good organization and clarity, Section 3 in the main document is challenging to understand. Reorganizing this section could enhance the paper's readability. Some propositions could be moved to the Appendix, freeing up space for the experimental section (see Weakness 2). Adding illustrative elements, such as a schematic diagram of the dataset construction and overall architecture, could further improve the readers' understanding.
2. Experiments: The set of experiments presented appear overly simplified and unrepresentative. Even though the multi-layer perceptron (MLP) models with two layers and varying unit numbers technically qualify as over-parameterized for d=1,2,4,8, the example is overly simplistic. The only experiment is with a sinusoidal signal form, and a y-label variance of 0.001 isn't explained or justified. The paper could be enriched by the inclusion of generally accepted benchmarks, such as regression benchmarks containing UCI datasets[1]. Moreover, it's clearly seen from your results that an ensemble of 5 models substantially outperforms an ensemble of 2 models, and training these ensembles isn't overly computationally demanding considering the complexity of data and models. Hence, comparing with larger ensembles, say 50 or 100 members, could provide more insightful results.
3. The initial part of the paper seems to promote the enhancement of **uncertainty quantification** by disentangling epistemic uncertainty into different components, but the actual evaluation of this uncertainty is lacking. The experiment appears to improve the **optimization process** rather than the quantification of uncertainty. An experimental suggestion would be to select a measure for epistemic uncertainty and apply it to both in-distribution (validation split) and out-of-distribution data. Computing the uncertainty measure for each data point across independently trained models several times and comparing the resulting ROC-AUC with the one obtained through your method could demonstrate whether the elimination of procedural variability truly improves the estimates of epistemic uncertainty.
4. There is no code available, so I can not check the reproduction of the experiments.

[1] Gal Y., Ghahramani Z. Dropout as a bayesian approximation: Representing model uncertainty in deep learning //international conference on machine learning. – PMLR, 2016. – С. 1050-1059.

**Questions:**

Can the method be somehow adapted for classification?



Edit: I would like to thank the authors for their answers during rebuttal period.

**Limitations:**

As I can see, this paper has no potential negative societal impact.

---

> ### Author Rebuttal · Authors · 2023-08-10
>
> We greatly appreciate the reviewer for the careful reading and valuable feedback. We address the reviewer's concerns below.
>
> Q1. (Experiments) For additional experiments on real-world datasets, please refer to our global response to all reviewers and our discussions there. We hope this would alleviate your concern that our data set is overly simplistic (and not motivated as well). For larger ensembles, the emphasis of our paper is that our PNC-predictor only requires 2 networks training. Therefore, a fair comparison with the same running time should be 2 networks in the deep ensemble approach, and our approach clearly outperforms it. Deep ensemble with any networks of more than 3 (5 or 50) does not change our conclusion even if they perform similarly to ours, since they require much more running times than ours.
>
> Q2. (Evaluation) Our work focuses on the quantification of $\textit{epistemic uncertainty}$, which refers to the errors coming from the inadequacy of the model or training data noises. This is different from $\textit{aleatoric uncertainty}$ (the randomness in the conditional data distribution). We provided a discussion on the difference between them in Appendix A. In particular, epistemic uncertainty is typically used on in-distribution data, while detecting out-of-distribution data generally requires aleatoric uncertainty or predictive uncertainty that is different from our study scope, even though this is certainly an interesting future direction for us to explore.
>
> When speaking of the “improve the optimization process” experiment, our understanding is that the reviewer probably means Table 2. However, we also have Table 1 for evaluation of our epistemic uncertainty quantifier. We conduct the evaluation of epistemic uncertainty statistically via the task of constructing confidence intervals, which is our task at the beginning of Section 4 (Table 1). More precisely, we aim to construct an interval that can cover the ground-truth value $h^*(x)$ with high probability with respect to the training randomness. To evaluate the confidence interval performance, we repeat our experiments 40 times to obtain 40 confidence intervals and check their coverage rate. For 95% confidence level, the coverage rate of 40 confidence intervals should be around or larger than 95%. Our performance appears competitive in this task as well.
>
> Q3. (Code.) We will definitely make our code publicly available. Also, and at the time being, Section E.1 Experimental Details provides our experimental details and configurations.
>
> Q4. (Classification) The theoretical developments needed for regression tasks and classification tasks appear different and not easily adaptable. NTK is the essential component of our theory, providing the theoretical foundation to establish the statistical properties of our algorithms. In previous literature, NTK is generally employed for mean-square-error (MSE) loss, such as
>
> Du, Simon, et al. "Gradient descent finds global minima of deep neural networks." International conference on machine learning. PMLR, 2019.
>
> Lee, Jaehoon, et al. "Wide neural networks of any depth evolve as linear models under gradient descent." Advances in neural information processing systems 32 (2019).
>
> He, Bobby, Balaji Lakshminarayanan, and Yee Whye Teh. "Bayesian deep ensembles via the neural tangent kernel." Advances in neural information processing systems 33 (2020): 1010-1022.
>
> as well as other references for NTK in our paper. The major technical discrepancy related to NTK between the regression tasks and the classification tasks is that the softmax operation in the network final layer and the cross-entropy-type loss used in the classification does not have an explicit statistical characterization of neural network training. Therefore, most NTK-based theories consider MSE loss. It would be an interesting but different direction to study the statistical guarantee for classification neural networks, which we recognize as our future research direction.

---

> > ### Comment · Reviewer_VEbh · 2023-08-15
> >
> > I would like to thank the authors for their detailed answers!
> >
> > I am quite confused with the answer to Q2.
> >
> > In particular:
> > ``epistemic uncertainty is typically used on in-distribution data, ... detecting out-of-distribution data generally requires aleatoric uncertainty''.
> > Can you please elaborate on what precisely you mean?
> >
> > If aleatoric uncertainty is captured by $\pi_{Y \mid X}$, does not it mean that aleatoric uncertainty makes sense only for that $X$ which came from the in-distribution? Otherwise not clear what should be the distribution over $Y$ for out-of-distribution $X$.

---

> > > ### Author Response · Authors · 2023-08-16
> > >
> > > We appreciate the reviewer for your reading and reply. We apologize for not adequately explaining the comment on the out-of-distribution detection. Let us elaborate on this point a bit more. There are two different scenarios for detecting out-of-distribution data, as follows:
> > >
> > > 1) The goal is to check whether the joint distributions are identical, i.e., whether $(X_{test}, Y_{test})$ is from the same distribution as $(X_{train}, Y_{train})$. To this end, we can estimate the aleatoric uncertainty $\pi_{Y \mid X}$ and check whether $Y_{test}$ falls into the highest density region of $\pi_{Y \mid X_{test}}$ [1]. If the answer is no, then with high probability, $Y_{test}|X_{test}$ is out-of-distribution, and thus $(X_{test}, Y_{test})$ is also out-of-distribution. This approach considers the conditional distributions of Y given X.
> > >
> > > This is the scenario we referred to in the rebuttal, which is indeed related to our problem setting. In particular, our problem aims to quantify the uncertainty in the obtained model $\hat{h}(x)$ against the target best model $h^*(x)$ (with a fixed $x$). We do so by providing a confidence interval around $h^*(X_{test})$ (not $Y_{test}$), because we focus on the quantification of epistemic uncertainty. On the other hand, $Y_{test}$ typically contains noise in regression tasks (i.e., the aleatoric uncertainty is not zero), and to carry out the out-of-distribution detection scheme, the construction of the highest density region needs to consider the aleatoric uncertainty.
> > >
> > > 2) The reviewer probably means the second scenario that has been widely studied in classification: The goal is to check whether the marginal distributions are identical, i.e., whether $X_{test}$ is from the same distribution as $X_{train}$ (typically given some conditions, such as for a certain class). To our knowledge, this task generally requires estimating the distribution of $X_{train}$ or its features (extracted from the neural network), and then checking whether $X_{test}$ fits. For instance, [2,3] use information from the likelihood function based on the estimated density function of $X_{train}$ from generative models, [4,5] check the energy score or softmax score from the distribution of the final layer's logits, [6] aggregates the p-values from the distribution of intermediate layers’ features. These works are related to quantifying the distribution of $X_{train}$ or its features (logits can also be viewed as condensed features), while our work has a different focus: The epistemic uncertainty is regarding the response $\hat{h}(x)$ while $x$ is a fixed test point (not random), and the randomness is from the training data and training procedure (different in-distribution training data and training procedures lead to different responses $\hat{h}(x)$).
> > >
> > > References:
> > >
> > > [1] Hyndman, Rob J. "Computing and graphing highest density regions." The American Statistician 50.2 (1996): 120-126.
> > >
> > > [2] Nalisnick, Eric, et al. "Do Deep Generative Models Know What They Don't Know?." International Conference on Learning Representations. 2019.
> > >
> > > [3] Grathwohl, Will, et al. "Your classifier is secretly an energy based model and you should treat it like one." International Conference on Learning Representations. 2020.
> > >
> > > [4] Liu, Weitang, et al. "Energy-based out-of-distribution detection." Advances in neural information processing systems 33 (2020): 21464-21475.
> > >
> > > [5] Hendrycks, Dan, and Kevin Gimpel. "A Baseline for Detecting Misclassified and Out-of-Distribution Examples in Neural Networks." International Conference on Learning Representations. 2017.
> > >
> > > [6] Haroush, Matan, et al. "A Statistical Framework for Efficient Out of Distribution Detection in Deep Neural Networks." International Conference on Learning Representations. 2022.

---

> > > > ### Comment · Reviewer_VEbh · 2023-08-18
> > > >
> > > > Dear authors,
> > > >
> > > > Thank you for your answers.
> > > >
> > > >
> > > > I think you resolved my confusion. Let me explain. I have some experience in uncertainty quantification, especially epistemic uncertainty quantification, and out-of-distribution detection in classification. In the problems I was addressing, we typically have some unrelated objects as out-of-distribution samples. Imagine that the source task was to classify cats and dogs, while during the test there appeared an image of a person's face.
> > > >
> > > > This is why I asked how $\pi(Y\mid X)$ can even be related for out-of-distribution samples (in my previous example, what should be the distribution over 'cat' and 'dog' labels given an image of a person's face)? Now I realize that you rather address ``outliers'', but not totally unrelated objects as out-of-distribution.
> > > >
> > > > I think some clarification of this in the main text may help people (with similar previous experiences as mine) to understand the paper better.
> > > >
> > > >
> > > > Based on the discussion, I would like to raise my evaluation score.

---

> > > > > ### Author Response · Authors · 2023-08-19
> > > > >
> > > > > We greatly appreciate the reviewer for your reply, including the kind sharing of your experience, and also for increasing our score. The discussion would also help us phrase our paper better. We would make sure to clarify our relation with out-of-distribution detection in the main body of the final version of this paper, as kindly suggested by the reviewer.

---

### Author Rebuttal · Authors · 2023-08-10

We greatly appreciate the reviewers for their careful reading and valuable feedback.

In this global response, we will show some additional experiments to showcase our approach to more challenging problems than in the paper via “simulated” real-world datasets. In the following, we will first discuss why synthetic datasets are employed in our experiments and then provide our experimental results and rationale.

The major reason for using synthetic datasets in our experiments is that precise evaluation of a confidence interval requires the following two critical components, which nevertheless are not available in real-world data: 1) We need to know the ground-truth regression function, while the label in real-world data typically contains some aleatoric noise. 2) To estimate the coverage rate of the CI, we need to repeat the experiments multiple times (40 times in our paper) by keep regenerating new independent datasets (from the same data distribution). In practice, we cannot regenerate new real-world datasets.

Thus, to address the reviewer’s concern about experiments on larger, more realistic and preferably real-world data, we provide a way to mimic an evaluation of the CI on real-world datasets as follows.

Step 1. Select a subset of the real-world data $(x_i, y_i)$ ($i \in \mathcal{I}$) and a test data point $(x_{k}, y_{k})$ ($k \notin \mathcal{I}$).

Step 2. For each experimental repetition j, add some “artificial” independent noise on the label to obtain a “simulated” real-world dataset $(x_i, y_i+\epsilon_{i,j})$ ($i \in \mathcal{I}$) where $\epsilon_{i,j}$ are all independent variables. Construct CI based on this “regenerating” training data $(x_i, y_i+\epsilon_{i,j})$ ($i \in \mathcal{I}$) and evaluate the coverage on $(x_{k}, y_{k})$.

In the above setting, $y_{k}$ becomes the “true” mean response of $x_{k}$ without aleatoric noise, and $\epsilon_{i,j}$ represents the only data variability. Again, as we discussed above, precise evaluation is impossible for real-world data; the above procedure, although not precise, is the best we can do to provide a resembling evaluation.

Using this setting, we conduct experiments on real-world benchmark regression datasets from UCI datasets: Boston, Concrete, and Energy. These results are shown in the Table in our attached PDF file.

From the results, we can see that our approaches also work well for these “simulated” real-world datasets. They can provide accurate confidence intervals that satisfy the coverage requirement of the confidence level. In contrast, DropoutUQ does not have such statistical guarantees. Therefore, our approaches not only work for synthetic datasets but also is scalable well to be applied in benchmark real-world datasets.

---

### Decision · Program_Chairs · 2023-09-21

**Decision:**

Accept (poster)

**Comment:**

Before rebuttal the main critic of the reviewers was the limmited experimental analysis on synthetic data. During rebuttal authors added additional experimental results which convinced all reviewers to vote for acceptance. The novel results should definitely be added to a revised version of the paper.